

# CIrrMap250: Annual maps of China's irrigated cropland from 2000 to 2020 developed through multisource data integration

Ling Zhang[1]*, Yanhua Xie[2], Xiufang Zhu[3], Qimin Ma[4], Luca Brocca[5]

[1]Key Laboratory of Remote Sensing of Gansu Province, Heihe Remote Sensing Experimental Research Station, Northwest Institute of Eco-Environment and Resources, Chinese Academy of Sciences, Lanzhou 730000, China

[2]Department of Geography & Environmental Sustainability, The University of Oklahoma, 100 East Boyd St, Norman, OK 73019, USA

[3]State Key Laboratory of Remote Sensing Science, Beijing Normal University, Beijing 100875, China

[4]College of Resources and Environment, Chengdu University of Information Technology, Chengdu 610225, China

[5]Research Institute for Geo-Hydrological Protection, National Research Council, Perugia 06128, Italy

*Correspondence to*: Ling Zhang (zhanglingky@lzb.ac.cn)

**Abstract.** Accurate maps of irrigation extent and dynamics are important to study food security and its far-reaching impacts on Earth systems and the environment. While several efforts have been made to map irrigated areas in China, few of them have provided multi-year maps, incorporated national land surveys, addressed data discrepancies, and considered the fraction coverage of irrigated cropland (i.e., the mixed pixel issue). In this study, we addressed these important gaps and developed new annual maps of China's irrigated cropland from 2000 to 2020, named as CIrrMap250. We harmonized irrigated area statistics and land surveys and reconciled them with remote sensing data. The refined estimates of irrigated area were then integrated with multiple remote sensing data (i.e., vegetation indices, hybrid cropland product, and paddy field maps) and irrigation suitability map through a semi-automatic training approach. We then evaluated our CIrrMap250 maps using independently interpreted 20,000 reference locations, high-resolution irrigation water withdrawal data, and existing local to nationwide maps. Our evaluation results showed that CIrrMap250 agreed well with the reference points, with an overall accuracy of 0.79-0.88 for years 2000, 2010, and 2020, respectively. The CIrrMap250-estimated irrigated area can explain 50-60% of the variance in irrigation water withdrawals across China. Our CIrrMap250 product showed superior performance than currently available ones (i.e., IrriMap_CN, IAAA, and GFSAD). CIrrMap250 revealed that China's irrigated area has increased by about 180,000 km2 (or 25%) from 2000 to 2020, with the majority (61%) being water-unsustainable and occurring in regions facing high to severe water stress. Moreover, our product unveiled a noticeable northward shift of China's irrigated area, attributed to substantial expansion in irrigated cropland across Northeast and Northwest China. The accurate representation irrigation area in CIrrMap250 will greatly support hydrologic, agricultural, and climate studies in China for improved water and land resources management.



## 1 Introduction

Irrigation is increasingly important as an adaption strategy to climate change (Zaveri and B. Lobell, 2019; Bhattarai et al., 2023) and plays a vital role in ensuring food security by reducing both water and heat stresses of crops (Zhu and Burney, 2022; Zhu et al., 2022). With 20% of spatial coverage in global croplands and providing 40% global food production (Wwap, 2019),
irrigated agriculture is a critical component of land and water resource management (Mcdermid et al., 2023). Globally, agricultural irrigation accounts for 60-70% of total freshwater withdrawals and 80-90% of consumptive water uses (Wu et al., 2022; Qin et al., 2022). Large volumes of irrigation water use intensify water management and drive myriad Earth system and environmental impacts (Mcdermid et al., 2021; Mcdermid et al., 2023). These impacts include changes in hydroclimatic and biogeochemical cycling (Yang et al., 2023; Guo and Zhou, 2022; Kang and Eltahir, 2018; Mishra et al., 2020; Thiery et al.,
2020), depletion of aquifers and surface water bodies (Cheng et al., 2014; Noori et al., 2021), freshwater salinization (Thorslund et al., 2021), and landsides (Lacroix et al., 2020). Given the vital importance of irrigation, it is essential to know the exact location and its' dynamics, which, however, are challenging, due to the hidden nature of irrigation signals and the frequent confusion between irrigated and rainfed fields (Ozdogan and Gutman, 2008; Chen et al., 2023; Zhang et al., 2022d).

Remote sensing provides significant opportunities for cost-effective and spatially explicit mapping of land surfaces
(Potapov et al., 2021). While numerous land use/cover and thematic cropland products have been made available to the public, they often lack information on irrigation status (Mpakairi et al., 2023; Xie and Lark, 2021). Over the past decade, there has been a growing interest in using satellite Earth observations to map irrigation extents (Massari et al., 2021). Currently, methods for mapping irrigated areas based on satellite data can be broadly categorized into vegetation-based, soil moisture-based, and vegetation-soil moisture integrated approaches. Various vegetation indices derived from optical sensors, such as the
normalized difference vegetation index (NDVI) (Rouse et al., 1974), green index (GI) (Gitelson, 2005), and normalized difference water index (NDWI) (Gao, 1996), have been employed to detect irrigated areas using threshold splitting methods (Zhu et al., 2014; Ozdogan et al., 2010; Wang et al., 2023), spectral matching techniques (Ozdogan and Gutman, 2008; Lu et al., 2021), decision trees (Ozdogan and Gutman, 2008; Shahriar Pervez et al., 2014; Ambika et al., 2016; Xiong et al., 2017), and supervised classification algorithms (Deines et al., 2019; Deines et al., 2017; Xie et al., 2019). The underlying principle
of the vegetation-based approach is that irrigated fields typically exhibit higher productivity, greenness, and moisture content compared to adjacent rainfed areas, especially under drought conditions. Moreover, remotely sensed soil moisture from microwave and optical sensors has also been applied to detected irrigate areas by using threshold splitting methods (Yao et al., 2022), supervised/unsupervised classification algorithms (Dari et al., 2021; Gao et al., 2018), and remote sensing-modeling comparison approaches (Zohaib and Choi, 2020; Zaussinger et al., 2019). The rationale behind the soil moisture-based method
is that irrigation alters soil moisture and leads to distinct spatiotemporal dynamics compared to adjacent rainfed areas. Additionally, the vegetation-soil moisture integrated approach, which combines vegetation indices with soil moisture for irrigated area detection, has also gained attention and achieved success in recent years (Zuo et al., 2023; Longo-Minnolo et al., 2022; Pun et al., 2017; Elwan et al., 2022).



Despite significant advancements in remote sensing technique for irrigation mapping, identifying irrigated areas at
large spatial scales (e.g., national and global levels) remains a grand challenge due to substantial variations in irrigation
practices, geographical and climatic characteristics (Salmon et al., 2015; Zhang et al., 2022d). This challenge is further
compounded by the lack of sufficient reference data (Xie and Lark, 2021; Xie et al., 2019). Consequently, high-precision
irrigated area maps are still lacking globally and in most countries (Mpakairi et al., 2023; Chen et al., 2023). In recent years,
researchers have sought to address the challenges of large-scale irrigation mapping by integrating remote sensing data with
agricultural statistics and other relevant datasets, such as irrigation suitability and existing irrigated area maps (Xie et al., 2021;
Meier et al., 2018; Zhang et al., 2022a; Zhang et al., 2022d). They have successfully generated new irrigation maps at the
global or national scale, featuring higher spatiotemporal resolution and mapping accuracy compared to previous products.
These efforts underscore the great potential of multisource data fusion techniques for large-scale irrigation mapping.

China is a big agricultural country with the largest irrigated area in the world. With only 8% of the world's arable
land, China feeds 20% of the global population and has a tight connection with the food supply chain of other nations.
Therefore, the development of reliable maps of irrigated cropland is particularly important for China. Despite this, less attention
has been devoted to mapping areas of irrigated cropland in China than in other countries with extensive irrigation, such as the
United States and India (Zhang et al., 2022d; Zhu et al., 2014). It is only in recent years that several maps of irrigated cropland
specifically tailored for China have emerged, facilitated by the integration of multisource data including remote sensing,
statistics, existing irrigation maps, and irrigation suitability maps (Zhang et al., 2022c; Bai et al., 2022; Xiang et al., 2020;
Zhang et al., 2022b; Zhang et al., 2022d).

While these previous studies have considerably improved our understanding of the spatial distribution of irrigated
cropland in China, limitations remain. First, few studies provide annual maps of irrigated cropland, hindering a spatiotemporal
analysis of irrigated areas in China. As a result, it remains unclear where the expansion of irrigated area is water-sustainable
(i.e., irrigated area expanded in paces without experiencing water stress) (Mehta et al., 2022). Second, irrigated area data from
official statistical bureaus, which were collected through field-sampling surveys in conjunction with bottom-up aggregation,
have been extensively utilized to constrain the overall extent of irrigated cropland in previous studies. Besides statistical data,
the National Land Surveys conducted by the State Council of China actually also offer accurate and reliable information on
irrigated cropland areas. The National Land Surveys involve a great number of investigators and relies on state-of-the-art
satellite remote sensing imagery and advanced survey techniques (Chen et al., 2022). The harmonization of irrigated area
statistics with the National Land Surveys might help to reduce biases and uncertainties associated with irrigated area (Yu et
al., 2021),  but this has rarely been taken into account. Third, the majority of farms in China are small and fragmented, with
the average farmland size being less than a hectare (Teluguntla et al., 2018). This leads to widespread presence of mixed pixels
in which both cropland other land use/cover types are present. However, most previous studies described irrigated cropland in
a Boolean fashion, where each pixel is entirely occupied by either irrigated cropland or non-irrigated cropland. This may lead
to overestimation or underestimation of irrigated cropland, depending on the proportion of cropland within the grid cell.
Finally, it is worth noting that, apart from the study conducted by Zhang et al. (2022a), many other studies assessed their maps



with a relatively limited number of reference samples, potentially compromising the reliability of their evaluation results. Obtaining a sufficient number of reference points is crucial for a robust evaluation of national-scale irrigated cropland maps, 100 a task that is, however, challenging due to the substantial cost and time involved.

Building on our previous work (Zhang et al., 2022d; Zhang et al., 2023a), this study aims to bridge the important gaps mentioned above by integrating remotely sensed data (i.e., vegetation indices, hybrid cropland products, and paddy field maps), irrigated area statistics and surveys, and irrigation suitability to create new annual maps of irrigated cropland in China (2000-2020). The newly developed irrigated cropland maps (named as CIrrMap250) have a spatial resolution of 250 meters and 105 describe irrigated cropland distribution through fractional coverage. Our specific objectives are: (i) to assess the accuracy of CIrrMap250 using a sufficient number of referencing points and high-resolution data on irrigation water withdrawals; (ii) to compare the performance of CIrrMap250 with three existing large-scale irrigation maps that cover the entire China, including IrriMap_CN (Zhang et al., 2022a), IAAA (Siddiqui et al., 2016), and GFSAD (Thenkabail et al., 2016), as well as a field scale map, i.e., OPTRAM30 (Yao et al., 2022); (iii) to investigate the spatiotemporal dynamics of China's irrigated cropland and 110 quantify the water sustainability of changes in irrigated area..

## 2 Data acquisition and processing

### 2.1 Remote sensing data

We collected the Terra Moderate Resolution Imaging Spectroradiometer (MODIS) vegetation indices, i.e., NDVI and Enhanced Vegetation Index (EVI) (Huete et al., 1997), from the NASA's Earth Science Data Systems 115 (https://www.earthdata.nasa.gov/). These indices were generated every 16 days with a spatial resolution of 250 meters. Meanwhile, the surface spectral reflectance of MODIS band 4 from the MOD09A1 product was resampled from the original 500 meters to 250 meters through the nearest neighbor interpolation (Debeurs and Townsend, 2008). These resampled data, in conjunction with the 250-meter and 8-day surface reflectance of band 1 from the MOD09Q1 product, were used to derive the Greenness Index (GI) (Supplementary Table S1). All MODIS data were quality screened against quality and usefulness 120 indicators, and only pixels free from clouds and snow/ice that meeting the highest quality criteria were deemed reliable (Hilker et al., 2012). The data for unreliable pixels were reconstructed using a straightforward nearest neighbor interpolation method.

We created a new high-resolution (30-m) hybrid cropland product for China (CCropLand30) by fusing state-of-the-art remote sensing land use and land cover products with the latest national land survey (Zhang et al., 2023a). CCropLand30 was generated at a 5-year interval from 2000 to 2020 and it exhibited a higher accuracy compared to existing products. Building 125 upon CCropLand30, we developed 250-m cropland layers for the years 2000, 2005, 2010, 2015, and 2020, which describe cropland distribution using the fractional coverage method, i.e., estimating the proportion of cropland in each 250-meter grid. These layers serve as the foundation for mapping irrigated cropland. Additionally, we extracted paddy fields from China's Land-use/cover dataset (CLUD) for the years 2000, 2005, 2010, 2015 and 2020 (Liu et al., 2014; Xu et al., 2018). Paddy fields



include cultivated land where rice and lotus roots are grown and supported by water and irrigation facilities, and they could be
considered as part of irrigated cropland with high confidence (Zhang et al., 2022c).

### 2.2 Irrigated area statistics and surveys

### 2.2.1 Harmonization of irrigated area statistics and surveys

We collected annual data on irrigated area (2000-2020) from diverse statistical yearbooks provided by the National Bureau of
Statistics of China and local statistical bureaus. These yearbooks encompass the Provincial Statistical Yearbook, the Rural
Statistical Yearbook, the China Statistical Yearbook for Regional Economy, and the China Water Statistical Yearbook. The
primary data source for these datasets is the China Economic and Social Big Data Research Platform (https://data.cnki.net/).
We compiled high-resolution (i.e., county-level) irrigated area data for more than 80% of  provinces in China and prefecture-
level data for the remaining provinces for each year from 2000 to 2020  (Zhang et al., 2022d), which provide substantially
more detailed information on the distribution of irrigated cropland than earlier studies (Xiang et al., 2020; Zhang et al., 2022b;
Zhu et al., 2014).

In addition to statistical data, land survey also provides accurate and reliable information on irrigated areas. China
has currently conducted three rounds of National Land Surveys in 1980s, 2010 and 2020.  The National Land Surveys engaged
a significant number of surveyors nationwide and utilized high-resolution satellite remote sensing imagery, along with
advanced survey techniques such as mobile internet, cloud computing, and drones (Chen et al., 2022). The results and maps
from these land surveys were not made public until recently due to the national security concerns. The Ministry of Natural
Resource of the People's Republic of China has released the county-level survey results (including cropland and its subtypes,
i.e., dryland, irrigated land, and paddy field) of the second and third National Land Surveys (https://www.mnr.gov.cn/). The
surveyed area of irrigated land and paddy field reflects the extent of irrigated cropland, and covers the periods 2009-2016 and
2019-2022. During years with available survey data, irrigated area statistics were harmonized with the surveyed irrigated area
at the county scale using Eq. 1. This process operated under the assumption that (1) the maximum value between statistical
and surveyed irrigated area should be more reliable, and (2) irrigated area should be less than the total cropland area. The first
assumption was made due to the tendency of both statistical and surveyed data to underestimate irrigated area, owing to
insufficient and representative field sampling (Zhang et al., 2022a) and the prevalence of fragmented and small croplands
(Teluguntla et al., 2018). We also tested alternative harmonization methods (e.g., mean and minimum), but they demonstrated
inferior performance compared to the maximum harmonization approach. In years lacking survey data, the harmonized
irrigated area was determined using Eq. 2, assuming that the relative changes in statistical irrigated area are reliable.

$$A_{harm}^{ts} = min(max(A_{stat}^{ts}, A_{surv}^{ts}), CA_{surv}^{ts}) \tag{1}$$

$$A_{harm}^{t2} = min(A_{harm}^{ts} \times \frac{A_{stat}^{t2} - A_{stat}^{ts}}{A_{stat}^{ts}}, CA_{surv}^{ts}) \tag{2}$$



where $A_{harm}$, $A_{stas}$ and $A_{surv}$ represent the county-level areas of harmonized, statistical and surveyed irrigated cropland, respectively; $CA$ is the surveyed area of cropland; and $ts$ and $t2$ indicate the year with and without land surveys, respectively.

### 2.2.2 Reconciliation between statistical/surveyed data and remote sensing data

Cropland area statistics and survey data are inherently incompatible with remote sensing data due to differences in
measurement techniques. The former measures the net area of cropland, while the latter represents the gross area of cropland that includes subpixel non-cropland features such as field ridges, linear elements, and scattered features (e.g., roads, ponds, and houses) (Zhang et al., 2023a). Consequently, statistical and surveyed cropland areas exhibit a negative and systematic bias compared to those derived from remote sensing data (Zhang et al., 2021; Zhang et al., 2022d). Irrigated cropland is a part of cropland, and its statistics and surveys also indicate the net area of irrigated area. Consequently, a gap exists between the
irrigated area from statistics/surveys and that derived from remote sensing data. Direct use of the statistical/surveyed irrigated acreage to constrain remote sensing-based irrigated cropland extent likely leads to underestimation of irrigated croplands (Schepaschenko et al., 2015). To fill this gap, we adjusted the harmonized irrigated area data (Section 2.2.1) to reconcile the statistical/surveyed data with remote sensing data, as seen in Eq.3. This adjustment was implemented under the assumption that the irrigation proportion remains consistent in both the statistical/surveyed data and the remote sensing-derived maps. For
example, if the statistical/survey data indicates a 99% irrigation proportion in the croplands of a given county, the remote sensing-derived irrigation proportion should also be as high as 99%.

$$A_{recon}^t = A_{harm}^t \times \frac{CA_{RS}^t}{CA_{surv}^t} \qquad (3)$$

where $A_{recon}^t$ and $A_{harm}^t$ are the reconciled and harmonized irrigated area, respectively, for the year $t$; $CA_{RS}^t$ is remote sensing-derived cropland area that was estimated from our hybrid cropland product (Zhang et al., 2023a); $CA_{surv}^t$ is the surveyed
cropland area; $CA_{RS}^t/CA_{surv}^t$ indicate the bias ratio of remote sensing-derived cropland area relative to surveys. This ratio was estimated for each county and constrained to the median value of all counties in its agricultural zones (Zhang et al., 2022c) to exclude extreme bias ratios and to ensure a conservative adjustment. In years lacking survey data, the bias ratio was estimated using a straightforward nearest-neighbor interpolation method.

### 2.3 Auxiliary data

This study utilized various auxiliary datasets, including meteorological and environmental variables, irrigation water withdrawal, water scarcity index, and administrative boundaries. Daily meteorological observations, including precipitation, relative humility, air temperature and pressure, at approximately 2400 meteorological stations were collected from the National Meteorological Information Center (NMIC, http://data.cma.cn/). These datasets were used in combination with the MCD43A3

albedo product for the computation of potential evapotranspiration (PET) (Priestley and Taylor, 1972) and aridity index (i.e., the ration of precipitation to PET). Environmental data consists of elevation, slope, crop intensity, soil type, and distance to water bodies. Elevation data originated from the Shuttle Radar Topography Mission digital elevation model (SRTM DEM), and the slope map was derived using the slope function in ArcGIS software based on SRTM DEM data. Distance to water bodies was determined using the Euclidean distance tool in ArcGIS, employing spatial distribution data of water bodies,

including rivers, lakes, reservoirs, canals, and ponds. The above auxiliary data for this study were sourced partly from the National Tibetan Plateau (https://data.tpdc.ac.cn/) and partly from the Resource and Environment Science and Data Center (https://www.resdc.cn/Default.aspx).

Moreover, data on irrigation water withdrawals at medium-sized administrative units known as prefectures were compiled for two distinct time frames (specifically, 2009-2011 and 2018-2020) from Water Resources Departments of the 31

provinces and the local statistical bureaus. The prefecture-level data on Water Scarcity Index (WSI) spanning the period from 2010 to 2020 were extracted from our earlier study (Zhang et al., 2023c). The WSI was computed as the ratio of total water usage (irrigation, industry, domestic water use, and other water use for forestry, livestock, fishery, and ecology) to water availability (i.e., total surface water and groundwater generated by precipitation).

## 3 Methodology

In this study, we create CIrrMap250 by integrating multisource data through a semi-automatic training approach (Zhang et al., 2022d; Xie et al., 2019). Following the acquisition and processing of data, our methodology began with the creation of training samples, as depicted in Figure 1. This step involves three major processes that include:(i) generating intermediate irrigation maps through a threshold-calibration method; (ii) establishing a training pool (i.e., potential training data) via overlay analysis of the intermediate maps; and (iii) generating training samples through random sampling from the training pool. Building upon

these training samples, we classify irrigated and rainfed cropland in each county on an annual basis using the random forest algorithm. The mapping outcomes were then then mosaicked and post-processed to obtain annual maps of irrigated cropland in China, denoted as CIrrMap250. Afterwards, we evaluated the accuracy of CIrrMap250, and conducted performance and visualization comparisons with existing products. Lastly, we examined the spatiotemporal changes in irrigated croplands and quantified the water sustainability of irrigation area expansion by comparing it with water stress areas.

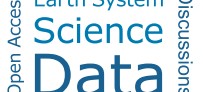

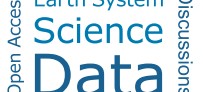

**Figure 1. Workflow of this study**

### 3.1 Generation of training samples

A threshold-calibration method was applied to automatically generate the training pool for irrigated and rainfed cropland, following the previous studies (Zhang et al., 2022d; Xie et al., 2019). We first calculated the peak vegetation index and adjusted it by irrigation suitability. The growth-period peak values of NDVI, EVI, and GI were determined for cropland grids in each year. A static irrigation suitability map were constructed based on the elevation, slope, and aridity index of cropland (Zhang et al., 2022d). As demonstrated by Liu et al. (2022), these factors are pivotal in influencing the spatial distribution of irrigated cropland in China. Cropland with lower elevation, gentler slope and higher aridity index was hypothesized to have higher



irrigation suitability and potential. Specifically, the cropland irrigation suitability map was derived by combining the irrigation suitability values of elevation, slope, and aridity index, as in Eq. 3.

$$S_{i,j,k} = \frac{1}{4} w_{1,k} SElev_{i,j} + \frac{1}{4} w_{2,k} SSlope_{i,j} + \frac{1}{10} w_{3,k} SArid_{i,j} \qquad (3)$$

where $S_{i,j,k}$ is the irrigation suitability for cropland cell $i$ in county $j$ of province $k$; $w$ is the weight of the influencing factors, which was determined by a trial-and-error procedure; $SElev$, $SSlope$, and $SArid$ are the irrigation suitability values of elevation, slope, and aridity index, respectively (Supplementary Table S2). The peak vegetation index was subsequently adjusted by irrigation suitability (Eq. 4), which the assumption that irrigated cropland is not only greener and more productive but also more suitable for irrigation than rainfed cropland.

$$SVI_{i,j,k}^{t} = S_{i,j,k} \times Peak\ (VI_{i,j,k}^{g,t}) \qquad (4)$$

where $SVI$ denotes the irrigation suitability-adjusted peak vegetation index; $VI$ denotes the value of vegetation index, $g$ and $t$ represent the growth period and year, respectively.

We then generated three intermediate irrigation maps for each year from 2000 to 2020 utilizing the $SVI$ (i.e., irrigation suitability-adjusted peak NDVI, EVI, and GI) and the paddy field maps. This was accomplished through a threshold splitting method (Pervez and Brown, 2010; Zhu et al., 2014; Meier et al., 2018). Specifically, the $SVI$ values for all croplands were ranked in descending order within each county, and the cumulative irrigated area was sequentially estimated. Subsequently, the accumulated area was compared with the reconciled irrigated area. The SVI value corresponding to the grid where the cumulative irrigated area closely matched the reconciled irrigated area was determined as the threshold value. Notably, for paddy fields, the SVI value was set to the maximum SVI of the croplands in a given county, prioritizing it as irrigated areas. The croplands were finally classified into "irrigated" and "rainfed" categories using Eq. 5.

$$cropland_{i,j,k} = \begin{cases} irrigated_{i,j,k}^{t} & SVI_{i,j,k}^{t} \geq threshold_{j,k}^{t} \\ rainfed_{i,j,k}^{t} & SVI_{i,j,k}^{t} < threshold_{j,k}^{t} \end{cases} \qquad (5)$$

These intermediate irrigation maps were overlaid and intersected; and pixels consistently identified by these maps as irrigated or rainfed cropland were designated as potential training samples, constituting the training pool for a given year and county. We randomly selected 200 rainfed cropland grids and 200 irrigated cropland grids from the training pool for each county and each year, which ensures a balance between the requirement for sufficient samples and computational efficiency of the classification algorithm (Xie et al., 2019; Zhang et al., 2022d).

**3.2 Classification of irrigated cropland using random forest**

We employed the random forest algorithm (Breiman, 2001) to classify irrigated and rain-fed cropland using the random samples extracted from the training pool. The implementation of the random forest algorithm was performed using the MATLAB TreeBagger function. The hyperparameters of the random forest model were optimized through a trial-and-error

process. These parameters include the number of trees, the minimum number of observations per node, and the number of variables randomly sampled at each decision split (Supplementary Table S3). The chosen predictors encompass both time-varying variables (i.e., vegetation indices, precipitation, temperature, PET, and aridity index) and time-invariant environmental variables (i.e., latitude, longitude, crop intensity, elevation, distance to water bodies, slope, and soil type). The classification of irrigated and rainfed cropland was conducted independently in each county for each year from 2000 to 2020. After

classification, we merged the annual and county-level mapping results to generate the maps of irrigated cropland in China. To enhance the accuracy of these maps, a spatial filter (a 7×7 window) was applied to eliminate isolated pixels (constituting <5% of the window area) and identify missed irrigated croplands (comprising >95% of the window area).

### 3.3 Accuracy assessment and inter-comparison

The accuracy of CIrrMap250 was assessed from three distinct perspectives. First, pixel-scale accuracy was evaluated using over 20,000 reference points collected from existing literatures and land-use maps of the National Land Survey in China. Furthermore, the performance of CIrrMap250 was indirectly assessed by comparing its irrigated area estimates with high-resolution data on irrigation water withdrawal. In addition, we compared CIrrMap250 with three currently available large-scale irrigation maps, i.e., IrriMap_CN (Zhang et al., 2022a), IAAA (Siddiqui et al., 2016), and GFSAD (Thenkabail et al.,

2016), as well as a field scale (30-m resolution) map in the Hexi Corridor of Northwest China (Yao et al., 2022).

### 3.3.1 Assessment with reference points

We assessed the accuracy of CIrrMap250 using three independent datasets of validation samples (Figure 2). The validation samples for the year 2000 were obtained from Zhu et al. (2014), which were primarily derived from the crop growth and soil

moisture dataset provided by the China Meteorological Data Sharing Service System (https://data.cma.cn/). The validation samples for the year 2020 were acquire from Chen et al. (2023), who mapped the center pivot irrigation systems (CPIS) in global arid regions. The CPIS are characterized by a circular irrigation pattern centered on pivots, which creates a distinct circular pattern on the crop (Figure 2c). This characteristic enables a reliable identification of the CPIS from remote sensing images. We extracted the CPIS polygons distributed throughout China and converted them into validation points (i.e., the

center of each CPIS polygon), which are mainly located in Northern China. In addition, we retrieved the validation samples for the year circa 2010 from the provincial land-use maps of the second National Land Survey in China (https://www.mnr.gov.cn/). Due to the lack of georeferencing information, we georeferenced these land use maps using the georeferencing tool in ArcGIS in conjunction with high-resolution images. The irrigated samples were taken from the patches of irrigated lands and paddy fields in the georeferenced land-use maps, while non-irrigated samples were taken from dryland

patches. Note that the surveyed land-use maps of the third National Land Survey are not available currently. In total, we obtained a more than 20,000 reference samples, enabling a robust assessment of the irrigation maps.

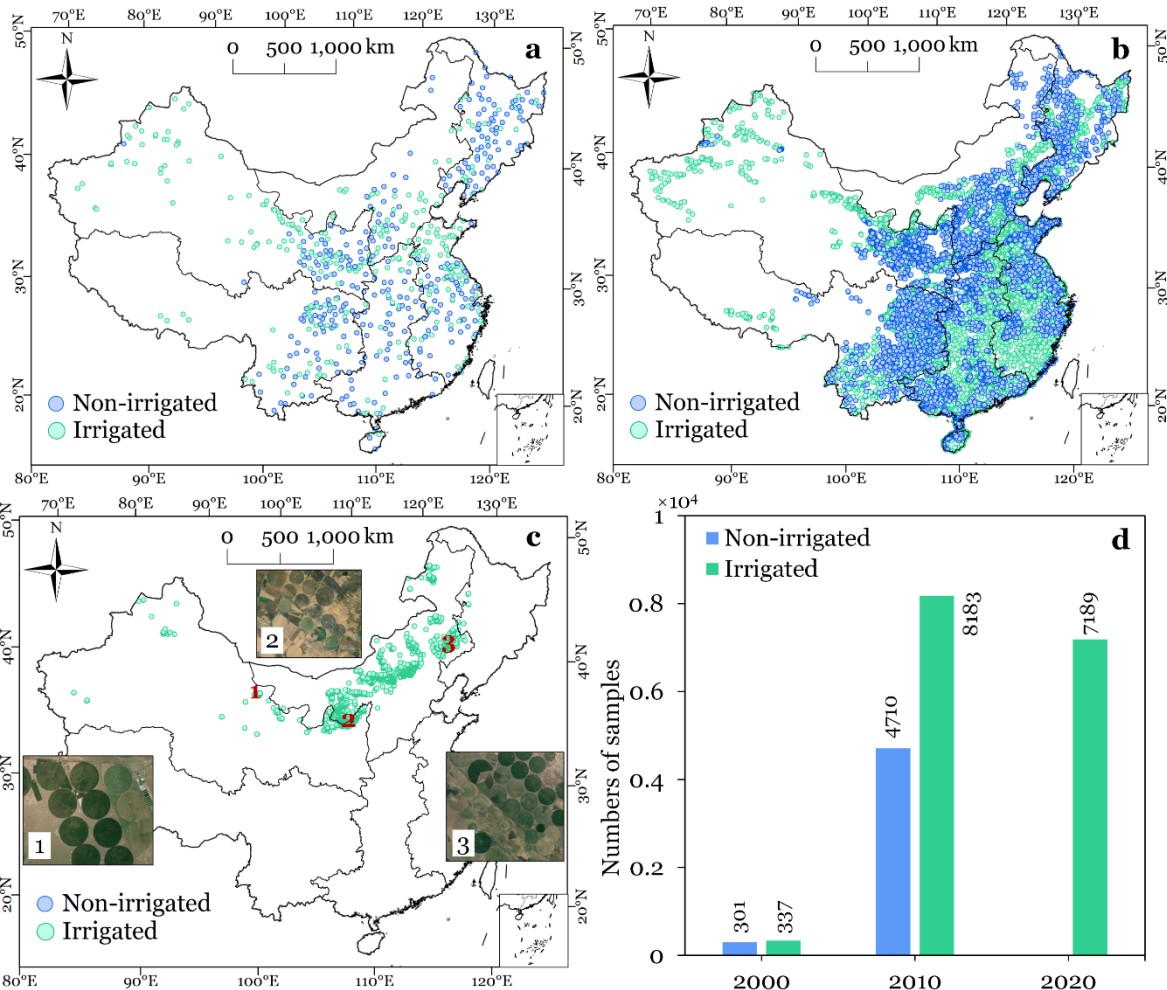

**Figure 2. Spatial distribution of validation samples. a** and **b**, Spatial distribution of the third-party samples in 2000 and 2020, respectively. **c**, Spatial distribution of the samples for the year 2010 retrieved from provincial land-use maps of the second national land survey in China. **d**, Numbers of irrigated and non-irrigated samples for different years.

The performance of CIrrMap250 was evaluated quantitively using the overall accuracy (OA), F1-score, producer's accuracy (PA), and user's accuracy (UA) (Supplementary Table S4). CIrrMap250 describes irrigated cropland distribution through fractional coverage rather than in a binary manner. The pixel values in CIrrMap250 indicate the percentage of irrigated cropland within each grid cell. It's noteworthy that this percentage represents the proportion of cropland within the 250-meter grid cell (estimated from the 30-meter hybrid cropland product), not the proportion of irrigated cropland to total cropland. Essentially, the cropland area within each 250-meter grid cell is categorized as either "irrigated" or "non-irrigated". Hence, for pixel-scale accuracy evaluation, CIrrMap250 was converted into binary maps, whereby pixels with values greater than 0 were coded as 1, representing irrigated cropland, while other pixels were coded as 0, representing non-irrigated area.





### 3.3.2 Assessment with irrigation water withdrawal data


We further assessed the performance of CIrrMap250 by comparing its irrigated area estimates with high-resolution (prefecture-level) data on irrigation water withdrawals for the years circa 2010 and 2020. Irrigated area is a major driver to irrigation water withdrawal (Puy et al., 2021; Lamb et al., 2021). Therefore, irrigation water withdrawal can indirectly validate the accuracy of irrigation maps (Zhang et al., 2022a). A more accurate irrigated cropland map is expected to exhibit a more robust correlation

between its irrigated area estimates and actual irrigation water withdrawals, in contrast to maps with lower accuracy. The strength of this correlation was gauged using the coefficient of determination ($R^2$) in a linear regression model, which quantifies the extent to which the variance in irrigation water withdrawals can be explained by changes in irrigated area.

### 3.3.3 Comparison with existing products

We compared CIrrMap250 with three existing large-scale irrigation maps including IrriMap_CN (Zhang et al., 2022a), IAAA (Siddiqui et al., 2016), and GFSAD (Thenkabail et al., 2016). IrriMap_CN are annual irrigated cropland maps across China at a 500-meter resolution spanning from 2000 to 2019. It was recently developed using MODIS data and machine learning method based on the training samples generated from the existing irrigation maps downscaled from the statistical data (Zhang et al., 2022a). IAAA are irrigated area maps at a 500-m resolution for the years 2000 and 2020, covering Asia and Africa. These

maps were created by leveraging seasonal variations captured in multi-seasonal satellite images (Siddiqui et al., 2016). GFSAD is a global irrigated cropland map at a 1000-m resolution for the year 2010. It was generated by overlaying the five dominant crops of the world with the remote sensing-derived irrigated and rainfed cropland area map (Thenkabail et al., 2016).

In addition, we obtained a field-scale remote sensing irrigation cropland map, denoted as OPTRAM30, developed by Yao et al. (2022). OPTRAM30 was specifically created for the Hexi Corridor in Northwest China using the soil moisture

change detection method with the optical trapezoid model. This map has a high resolution of 30 meters and demonstrates an accuracy approaching 100% when validated against in situ datasets. Given the high accuracy and spatial resolution of OPTRAM30, it can serve as a valuable reference for the evaluation of large-scale irrigation maps. Hence, we additionally made a comparison of CIrrMap250, IrriMap_CN, IAAA, and GFSAD with OPTRAM30 in the Hexi Corridor.

### 3.4 Changes in irrigated area and comparison with water stress areas


We examined the trends in irrigated areas in a spatially explicit manner using 21 years of data. The trends were quantified by calculating the slope of the regression line fitted to the time-series data of irrigated areas at the pixel scale using the least squares method. Furthermore, we adopted the concept of "center of gravity" to track the spatial dynamics of irrigated areas (Zeng and Ren, 2022). The gravity center of irrigated area ($X$, $Y$) is represented as:

$$X^t = \frac{\sum_{i=1}^{n} IrrArea_i^t \times x_i}{IrrArea_i^t} \qquad (6)$$



$$Y^t = \frac{\sum_{i=1}^{n} IrrArea_i^t \times y_i}{rrArea_i^t} \quad\quad\quad (7)$$

where $IrrArea_i^t$ denotes the irrigated area in grid $i$; $x_i$ and $y_i$ are the longitude and latitude of grid $i$, respectively; $n$ is the number of irrigated cropland grids; and $t$ is year.

In addition, we quantified the water sustainability of changes in irrigation areas. The expansion and decline in irrigated areas between 2000 and 2020 were first identified at the pixel scale. To better visualize the results, we aggregated changes in irrigated area to a 5-km resolution, following previous studies (Deines et al., 2019; Xie and Lark, 2021). Subsequently, we compared these changes with a prefecture-level water stress map derived from the mean values of WSI over the period 2010-2020. The WSI denotes the fraction of available water resources appropriated by humans and is employed to categorize water stress across different prefectures into four levels: low (WSI≤0.2), moderate (0.2<WSI≤0.4), high (0.4 < WSI≤1.0), and severe (WSI>1) (Zhang et al., 2023c). Expansions of irrigated areas under severe to extreme water stress were designated as "unsustainable" due to their potential to exacerbate the depletion of surface water and groundwater resources (Mehta et al., 2022). Conversely, expansion of irrigated aeras under low to moderate water stress or reductions in irrigated areas under severe to extreme stress were deemed "sustainable".

## 4 Results

### 4.1 Accuracy assessment of irrigated cropland maps

#### 4.1.1 Pixel-scale assessment

As depicted in Figure 3 and Supplementary Table S5, CIrrMap250 attains an OA and F1-score of 0.79 and 0.78, respectively, for the year 2000, surpassing the performance of IrriMap_CN and IAAA. In the year 2010, CIrrMap250 achieves a high OA of 0.79 and a F1-score of 0.71, whereas the existing maps attain OA values below 0.66 and F1 scores under 0.63. For the year 2020, CIrrMap250 detects 88% of the fields with center pivot irrigation systems, while IrriMap_CN identifies only 20% (Figure 3c and Supplementary Figure S1). For irrigated samples, CIrrMap250 has significantly higher producer's accuracy compared to the existing products. CIrrMap250 and IrriMap_CN performs similarly in user's accuracy. For non-irrigated samples, the producer's accuracy of CIrrMap250 is relatively lower than that of IrriMap_CN, but the user's accuracy is significantly higher than that of IrriMap_CN. In terms of producer's and user's accuracy, both CIrrMap250 and IrriMap_CN obviously outperform IAAA and GFSAD.





**Figure 3. Performance of CIrrMap250 and the existing maps (IrriMap_CN, IAAA, GFSAD).** Panels **a**, **b** and **c** show the results for 2000, 2010, and 2020, respectively. OA, PU, and UA are overall accuracy, producer's accuracy, and user's accuracy, respectively. Irr and NIrr indicate irrigated and non-irrigated samples, respectively.




### 4.1.2 Nationwide and regional comparison with existing products

Figure 4 compares the spatial distribution of irrigated cropland in CIrrMap250 with the existing maps. At the national scale, CIrrMap250 and IrriMap_CN, specifically developed for China, can capture similar irrigation patterns. Irrigation hotspots (e.g, North China Plain and Northwest China) and well-known irrigation districts like Hetao, Baojixia, Dujiangyan, Qingtongxia, and Fenhe are consistently identified by these maps. The irrigated croplands depicted by CIrrMap250 are more widely distributed than those portrayed by IrriMap_CN across the majority of China (Supplementary Figure S2). CIrrMap250 yields irrigation ratios (i.e., the ratio of irrigated area to the total cropland area) of 0.58, 0.70, and 0.96, respectively, for China, Northern China, and Xinjiang Uygur Autonomous Region. These values align more closely with the reality and the official report (https://gtdc.mnr.gov.cn/), in comparison to those derived from IrriMap_CN, which are only 0.47, 0.37, 0.61, respectively (Supplementary Figure S2). However, CIrrMap250 tends to yield lower estimates of irrigation area in Northeast China (NEC) when compared to IrriMap_CN. In contrast to CIrrMap250 and IrriMap_CN, IAAA notably underestimates irrigated croplands in Northwest China (NWC) and North China (NC), but overestimates them in NEC and Southwest China (SWC). GFSAD shows overestimations of irrigated area in the Dujiangyan district and the North China Plain, but exhibits evident omission errors in sparsely distributed irrigation regions like NWC and the southern part of South China (SC).

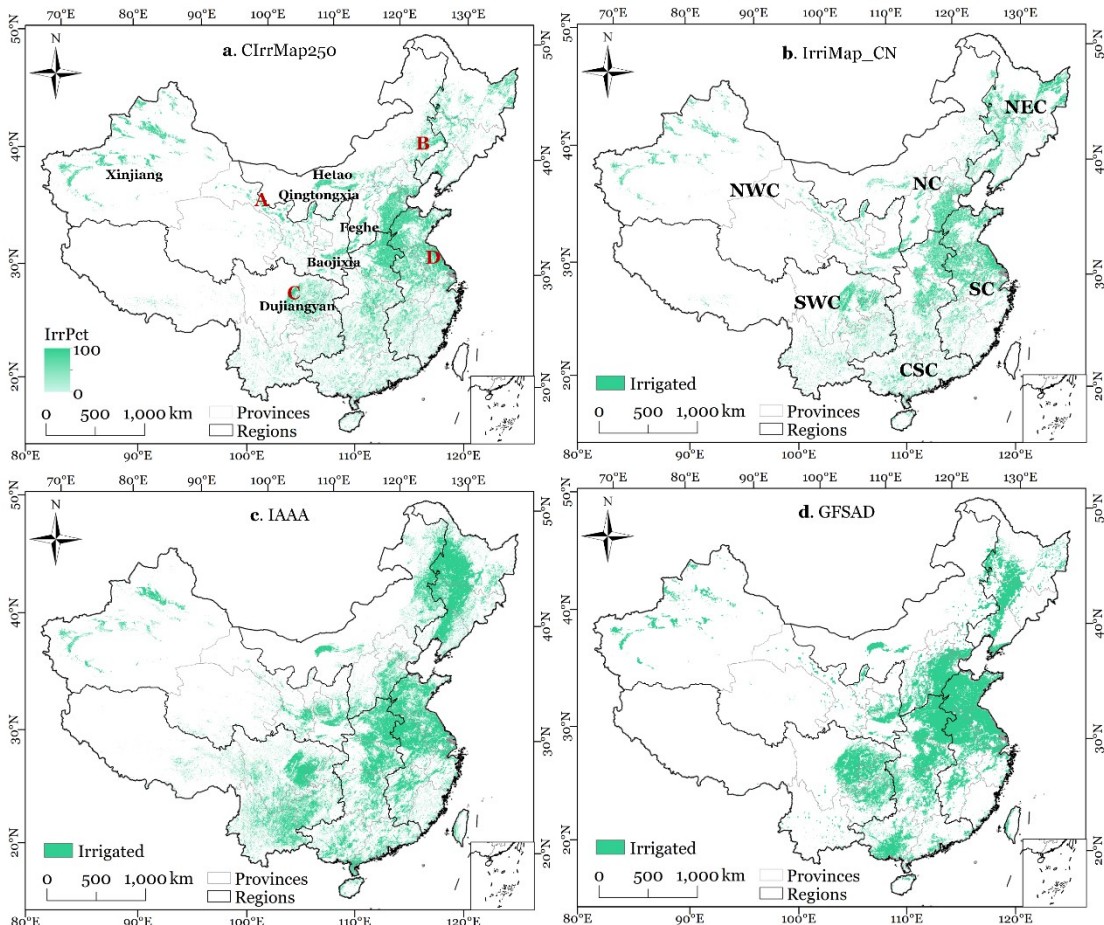


**Figure 4. Spatial distribution of irrigated cropland in different irrigation maps for the year 2010.** NEC, NC, NWC, SWC, SC and CSC are the abbreviations of Northeast China, North China, Northwest China, Southwest China, South China, and Central South China. IrrPct represents the proportion of irrigated cropland relative to the total area of a grid cell.

We further compared CIrrMap250 with the existing maps in four heavily irrigated zones (A-D locations are shown
in Figure 4a). Zones A and B are situated in arid regions where crop growth depends greatly on irrigation, while Zones C and D are located in humid regions where paddy rice is extensively cultivated and relies heavily on supplemental irrigation. As depicted in Figure 5, CIrrMap250 accurately portrays the actual distribution of irrigated cropland in these zones. In contrast, IrriMap_CN underestimates irrigation extent in zones A and B and lacks detailed information on irrigated cropland in zones C and D. IAAA significantly underestimates the irrigated area in zone A, incorrectly identifies irrigated cropland in zone B,
and overestimates irrigated cropland in region C. The GFSAD, with a coarse resolution of 1 kilometer, has the lowest agreement with the distribution of actual irrigated cropland among the four maps.


**Figure 5. Visual comparison of CIrrMap250 with the existing maps.** The five rows from top to bottom correspond to the Google map, CIrrMap250, IrriMap_CN, IAAA

and GFSAD, respectively. Locations of the four selected zones are presented in Figure 4a.

Figure 6 provides an additional comparison of the aforementioned large-scale irrigation maps with the field-scale remote sensing irrigation map (OPTRAM30) in the Hexi Corridor of Northwest China. CIrrMap250 exhibits a robust agreement with OPTRAM30 in mapping irrigated cropland. While IrriMap_CN captures the general pattern of irrigated croplands, it tends to underestimate the extent of irrigated cropland in this region. In contrast, IAAA struggles to identify

irrigated cropland in this area, displaying significant omission and commission errors. Similarly, GFSAD has a limited ability to accurately depict irrigated areas in the Hexi Corridor.

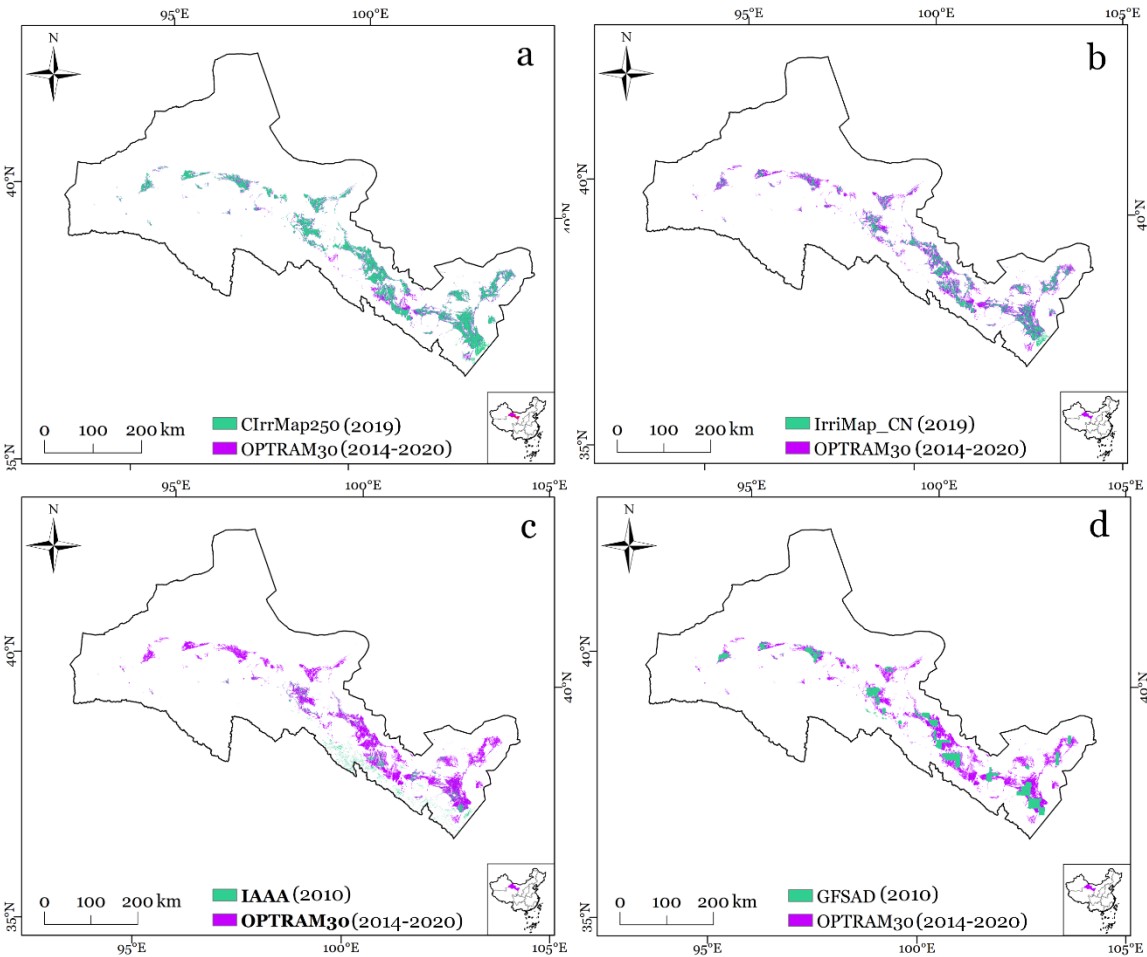

**Figure 6. Comparison of large-scale irrigation maps (CIrrMap250, IrriMap_CN, IAAA, GFSAD) with the field-scale remote sensing irrigation map (OPTRAM30) in the Hexi Corridor of Northwest China.**




### 4.1.3 Comparison irrigated area with high-resolution irrigation water use data

As illustrated in Figure 7, there is a good correlation between the CIrrMap250-estimated irrigated area and the irrigation water withdrawal. Changes in irrigated area determined by CIrrMap250 account for approximately 50% and 60% of the variance in irrigation water withdrawals for the years circa 2010 and 2020, respectively. In contrast, changes in irrigated areas derived

from IrriMap_CN can only explain 40% and 48% of the variance in irrigation water withdrawals for the same periods, namely 2010 and 2020. The estimates of irrigated areas from the other two maps, namely IAAA and GFSAD, are able to explain only a small proportion of the variances in irrigation water withdrawals (i.e., 0.12 and 0.20), suggesting a relatively low performance of these maps in China. These results indirectly imply the better performance of CIrrMap250 over the existing irrigation maps.

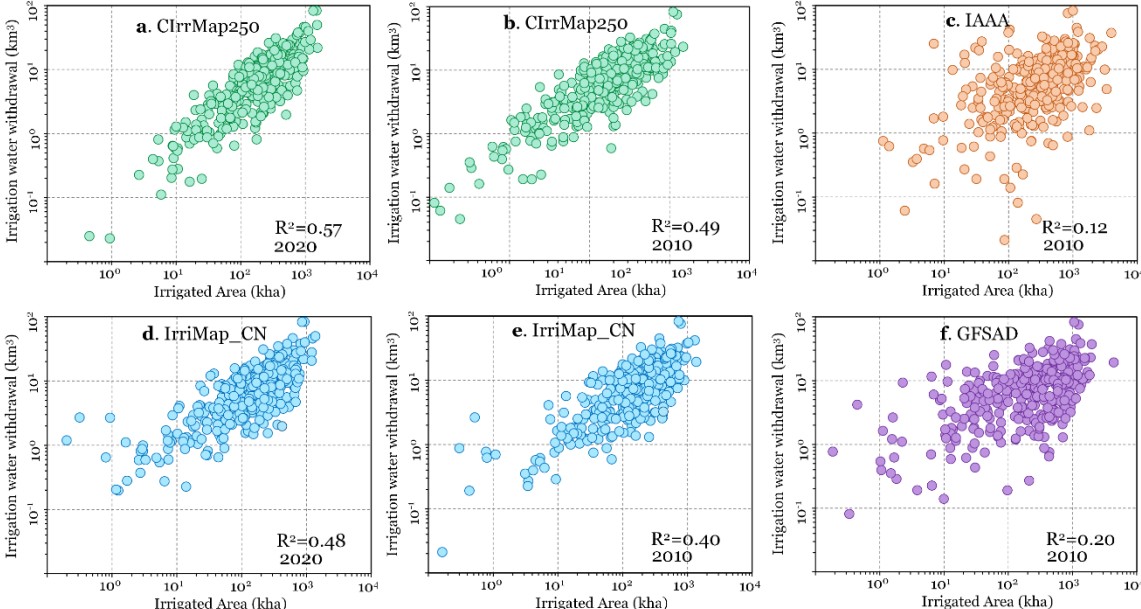

**Figure 7. Scatterplots of irrigated area estimates against irrigation water withdrawals for the years circa 2010 and 2020.** The data are presented in logarithmic units to reflect both small and large values.

### 4.2 Spatiotemporal changes of irrigated croplands

As depicted in Figure 8, irrigated area expands significantly in NEC and NWC from 2000 to 2020. Conversely, it reduced
notably in the northern parts of SC and CSC, the northeastern part of SWC, and the southern parts of CSC and NC. The decline in irrigated areas tends to be concentrated in populous areas, which can be attributed to the rapid urban expansion on large areas of cropland (Zhang et al., 2023a). The center of gravity for irrigated area is situated on the border of NC and CSC, and exhibits a noticeable northward shift from 2000 to 2020. This northward spatial trend in irrigated area is likely to exacerbate



the water crisis in Northern China (Li et al., 2023), which has only 20% of China's water resources but supports more than
half of its population.

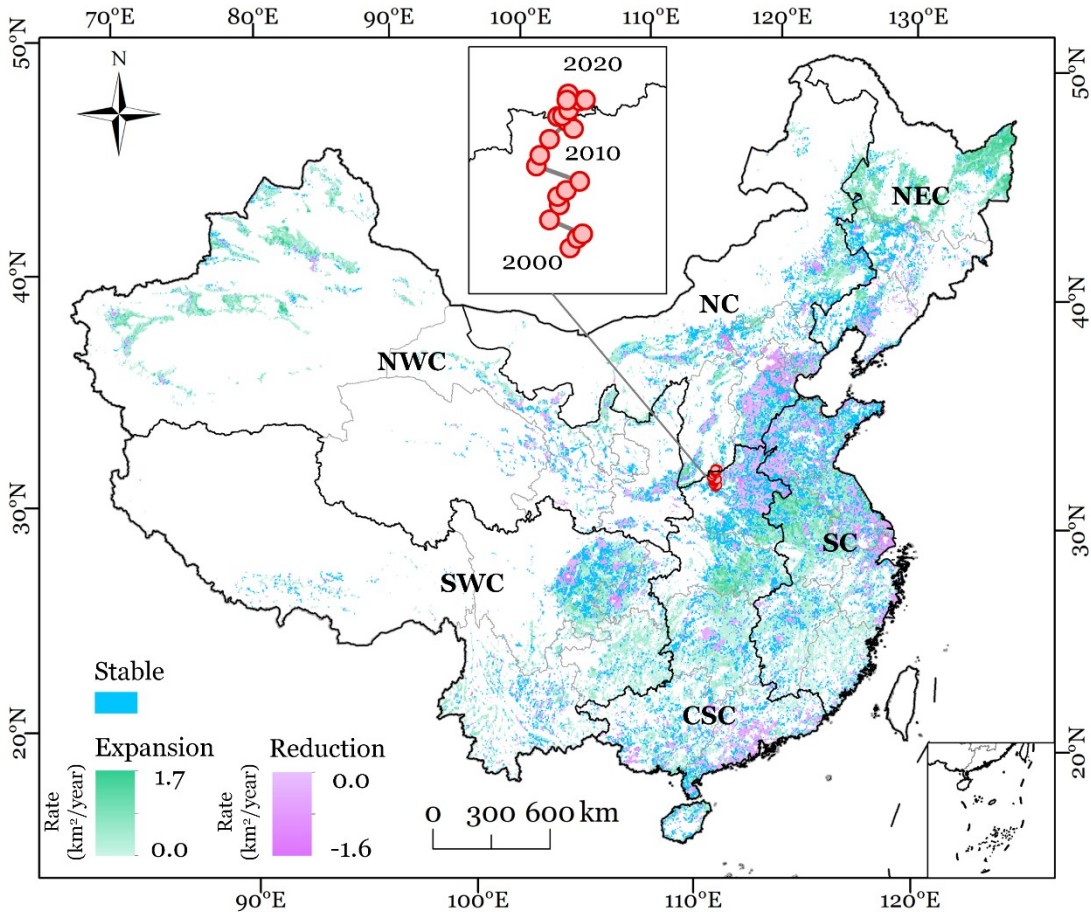

**Figure 8. Spatiotemporal changes in irrigated area from 2000 to 2020.** Pixels with significant increasing or decreasing trend (p<0.05) marked as "expansion" or "reduction", while those with insignificant changes as "stable". Pixels with < 5% irrigated croplands were excluded from the map. Inserted panel on the top of the figure depicts the center-of-gravity movement of irrigated area.

As shown in Figure 9, all subregions exhibit an increasing trend in irrigated area from 2000 to 2020, with NEC expanding significantly faster than the other subregions. The irrigated area of China increases from 750,000 to 950,000 km$^2$ at the rate of about 10,000 km$^2$/year (or 1.29%/year). Notably, NEC and NWC contribute to about half of this expansion. Despite the consistent upward trend in irrigated area, the relative changes in the proportion of irrigated areas, in relation to China's total irrigated area, are inconsistent across different subregions. The proportion of irrigated area in NEC and NWC shows an upward trend, whereas that in SCS, SC, and NC displays a downward trend. SC has the largest proportion of irrigated cropland (26%-30%), followed by CSC (22%-24%), NC (16%-17%), NWC (12%-14%), SWC (11%), and NEC (7%-11%).

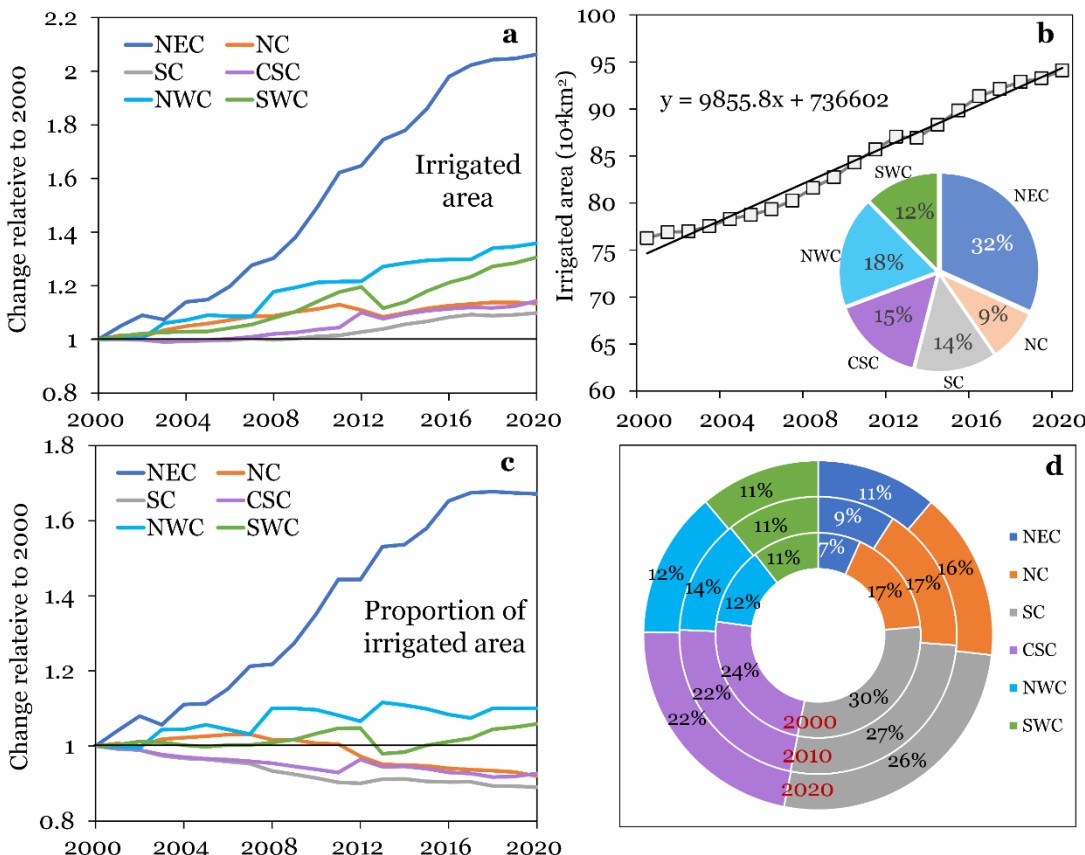

**Figure 9. Changes in irrigated area of the six subregions of China during 2000-2020. a**, Relative changes in irrigated area. **b**, Changes in China's total irrigated area, with the contribution of different subregions depicted in the inserted pie chart. **c**, Relative changes in the proportion of irrigated area. **d**, Proportion of irrigated area for the years 2000, 2010 and 2020.

### 4.3. Irrigated cropland changes under different water stress levels

Figure 10 shows the changes in irrigated cropland under different levels of water stress. We find a gross expansion of irrigated area by about 250,000 $km^2$ in China from 2000 to 2020, of which 64% is unsustainable from the perspective of water resources and has been in regions with high to severe water stress. The expansion of irrigated area is mainly situated in NWC, NEC, NC, and the northern parts of CSC and SC. The gross reduction in irrigated area is about 70,000 $km^2$, of which 72% has been sustainable and located in regions with high to severe water stress. These sustainable reduction in irrigated area, primarily located in NC, CSC and SC, mitigates the unstainable irrigated cropland expansion. The net expansion of irrigated area is about 180,000 $km^2$, of which 61% is water unsustainable. The subregions NEC and NWC have a larger proportion of unsustainably expanded irrigated area compared to other subregions, accounting for about 70% of China's net unsustainable



irrigation expansion. In contrast, the subregions CSC and SWC have a greater proportion of sustainably expanded irrigated area than in other subregions due to the abundance of water resources and lower water stress.

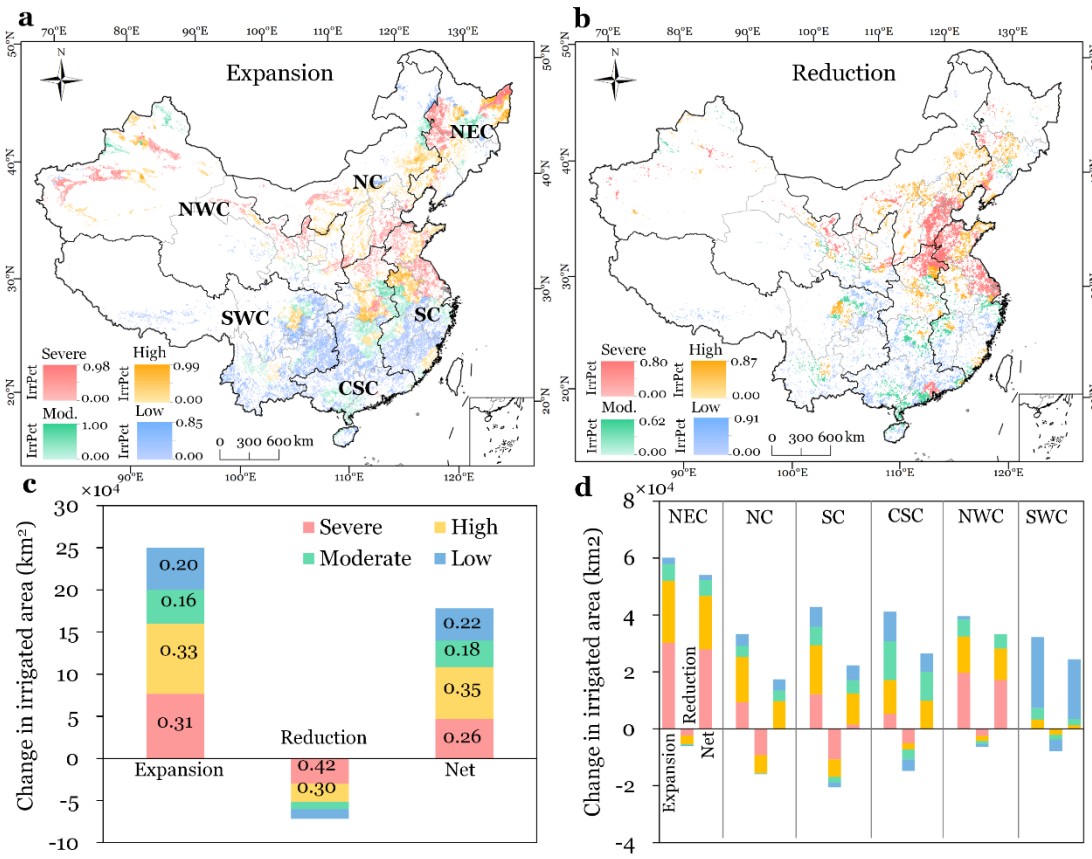

**Figure 10. Changes in irrigated area between 2000 and 2020 under different water stress levels**. Panels **a** and **b** show the spatial distribution of gross expansion and reduction of irrigated areas, respectively, under low to severe water stress. Panel **c** and **d** show the gross and net changes in irrigated area by water stress category for China and the six subregions.

# 5 Discussion

## 5.1 Improvement of CIrrMap250 over existing products

Our CIrrMap250 product provides annual maps of China's irrigated cropland from 2000 to 2020, exhibiting higher accuracy compared to existing products. The improved performance of CIrrMap250 can be attributed to several key factors. First, CIrrMap250 has digested unprecedentedly detailed irrigated area statistics and reliable national land surveys, and meanwhile, has considered discrepancy been statistical/survey data and remote sensing data. We compiled county-level statistical data for





over 80% of provinces in China, along with prefecture-level data for the remaining provinces. These datasets, for the first time, were harmonized with the national land surveys, greatly reducing the errors and uncertainties in irrigated area statistics. The harmonized irrigated area data were further adjusted to reconcile the statistical/surveyed data with remote sensing data. The reconciliation was necessary because statistical and surveyed irrigated area represents the net extent of irrigated cropland, whereas remote sensing-derived irrigated area indicates the gross extent. Without adjusting the original irrigated area statistics,

the irrigation extent would be significantly underestimated, leading to a decrease in irrigation mapping accuracy by 8%-26% (Supplementary Figure S4).

Furthermore, CIrrMap250 describes irrigated cropland distribution through fractional coverage, rather than the binary approach adopted in most existing products. The majority of farms in China are small and fragmented. For instance, in the year 2020, we observed that about 37% of the 250-m cropland grids were occupied by less than half of croplands in China,

while less than 40% of cropland grids were occupied by more than 90% of croplands. Therefore, it becomes crucial to consider the fraction coverage of cropland in cropland masks for the purpose of mapping irrigated areas. We conducted an additional irrigation mapping experiment, in which the 250-m cropland maps were described in a binary manner and resampled from the 30-m hybrid cropland product. As depicted in Supplementary Figure S5, a substantial portion of irrigated cropland would be overlooked if the fractional coverage of cropland is not taken into account, particularly in South China. The accuracy of the

irrigated cropland map would decrease by approximately 5%-6% (Supplementary Figure S6).

Lastly, CIrrMap250 has incorporated an irrigation suitability map, derived by combining irrigation suitability values of three influential factors-elevation, slope, and aridity index—using a weighted average method. To demonstrate the importance of integrating irrigation suitability into irrigation mapping process, we randomly generated 250 sets of weights (assigned to the influencing factors) for all provinces in China, resulting in 250 distinct irrigation suitability maps. Based on

these maps, we then created 250 different irrigated cropland maps for the year 2010 using the proposed method of this study. As shown in Supplementary Figure S7, regardless of the choice of irrigation suitability maps, these irrigation maps consistently outperform the baseline irrigation map, which was created using the method in this study but excluded irrigation suitability during the mapping process. Furthermore, there is a narrow range (0.75-0.77) in the overall accuracy of these irrigation maps, implying the robustness (low sensitivity) of the mapping method to the use of different irrigation suitability maps.


### 5.2 Uncertainties and limitations of CIrrMap250

Despite the advancements made in CIrrMap250 compared to existing products, we acknowledge several limitations associated with the product. Firstly, the accuracy of CIrrMap250 is intricately tied to irrigated area statistics. Despite our efforts to harmonize irrigation statistics with national land surveys, inherent biases and uncertainties persist due to technical and political

factors, such as variations in statistical methods and administrative divisions(Thenkabail et al., 2009; Meier et al., 2018). These biases and uncertainties are inevitably reflected in CIrrMap250, since our training samples were derived from the statistics-constrained irrigation maps. Furthermore, CIrrMap250 has a relatively coarse spatial resolution of 250 meters. While the



spatial resolution of CIrrMap250 is higher than many existing large-scale irrigation maps, it may still not be applicable to smaller spatial scales (e.g., field or irrigation district scales). In addition, the mix-pixel problems could bring uncertainties to our mapping results. Despite the consideration of fractional average of cropland, CIrrMap250 cannot differentiate irrigated and rain-fed croplands at the subpixel scales. There are many small and fragmented croplands in the mountainous regions of South China with complex terrain and diverse vegetation types. CIrrMap250 should be used with caution in these regions due to the wide existence of the mixed pixels. The mix-pixel problems could not only significantly affect the precision of cropland masks (Zhang et al., 2023a), but also the difference in vegetation indices between irrigated and rainfed cropland. Despite these limitations, our CIrrMap250 makes a valuable contribution to the field of irrigation mapping and will greatly support hydrologic, agricultural, and climate studies in China. Efforts to overcome the above limitations and explore avenues for potential enhancements will undoubtedly improve the accuracy and utility of our irrigation maps in the future.

## 6 Data availability

The annual maps of China's irrigated cropland from 2000 to 2020 (named as CIrrMap250) can be accessed at: https://doi.org/10.6084/m9.figshare.24814293.v1 (Zhang et al., 2023b). All maps are presented in the GeoTIFF format, with geographic coordinates using the WGS84 reference system. Pixel size is 0.00225 × 0.00225 degree (~250 m ×250 m at Equator).

## 7 Conclusions

This study outlines the development of annual maps of irrigated cropland in China from 2000 to 2020, denoted as CIrrMap250. The new product was developed by integrating multisource data, including remote sensing data (vegetation indices, hybrid cropland product, and paddy field maps), irrigated area statistics and surveys, and irrigation suitability map. The integration of these data was achieved through a semi-automatic training approach, which first generated training samples using a threshold-calibration method and subsequently employed the random forest algorithm for classifying irrigated and rainfed cropland. We evaluated the accuracy of CIrrMap250 using over 20,000 reference collected from existing literatures and land-use maps of the National Land Survey in China. Furthermore, an indirect assessment of CIrrMap250 was carried out using higher-resolution data on irrigation water withdrawals. Our CIrrMap250 product was compared to three available large-scale irrigation maps (i.e., IrriMap_CN, IAAA, and GFSAD) as well as a field scale map (i.e., OPTRAM30).

Results indicated that CIrrMap250 attained an overall accuracy of 0.79-0.88 for the years 2000, 2010 and 2020, surpassing the precision of the existing products. Furthermore, the CIrrMap250-estimated irrigated area can explain 50-60% of the variance in prefecture-level irrigation water withdrawals, and showed a stronger correlation with irrigation water

withdrawals than the existing products. The visual comparison further confirmed the better performance of CIrrMap250 over the existing products. Leveraging 21 years of data, we found a consistent upward trend in the irrigated area across all subregions of China from 2000 to 2020. Notably, the growth rate in Northeast and Northwest China surpasses that of the remaining subregions. Consequently, the center of gravity of China's irrigated cropland shifted significantly northward, potentially exacerbating the water crisis in North China. Over the period from 2000 to 2020, we observed a net increase of about 180,000 km$^2$ (or 25%) in China's irrigated area. However, a significant portion (61%) of this expansion is deemed unsustainable from a water resources perspective and have been in regions facing high to severe water stress.

The performance improvement of CIrrMap250 over existing products can be attributed to the digestion of detailed irrigated area statistics and reliable national land surveys, the consideration of discrepancy been statistical/survey data and remote sensing data, the description of irrigation cropland distribution through fractional coverage, and the incorporation of irrigation suitability. We anticipate that our CIrrMap250 product will greatly support hydrologic, agricultural, and climate studies in China for improved water and land resources management.

## Author contribution

LZ conceived the research, carried out the experiments, analysed the results, and prepared the manuscript with contributions from all co-authors. YX analysed the results, provided the technical support, reviewed and edited the manuscript. XZ and QM collected the validation dataset. LB reviewed and edited the manuscript, and supervised the work.

## Competing interests

The authors declare that they have no conflict of interest.

## Acknowledgements

This study is supported by the National Natural Science Foundation of China (42271286 and 41901045), and the Youth Innovation Promotion Association of Chinese Academy of Sciences (2023454). We greatly appreciate the Ministry of Natural Resource of the People's Republic of China for the data provision.

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
