# Peer review of "CIrrMap250: Annual maps of China's irrigated cropland from 2000 to 2020 developed through multisource data integration"

_Earth System Science Data, 2024_

## Referee Comment (RC2)

**Reviewer's Report for ESSD-2024-2**

**CIrrMap250: Annual maps of China's irrigated cropland from 2000 to 2020 developed through multisource data integration**

Author: Zhang et al.

**General summary:**

This study presents the development of a multi-year (2000-2021) irrigated cropland map for China, named CIrrMap250. The authors employ a semi-automatic training approach integrating remote sensing data (vegetation indices, hybrid cropland products, and paddy field maps), county-level irrigation statistics and surveys, and an irrigation suitability map. Utilizing a threshold-calibration method and the random forest algorithm, the CIrrMap250 map is evaluated against reference sites and other large-scale irrigation maps, demonstrating superior accuracy. The study reveals a consistent net expansion of irrigated croplands in Northeast and Northwest China, with over 60% deemed unsustainable due to severe water stress. The CIrrMap250 map holds significant application potential for water resource management and food security. I have some comments below and please address them before this article can be published.

**Major comments:**

1 L74-81: China's vast agricultural landscape comprises diverse cropping systems and associated irrigation methods, such as rice paddies in the South and Northeast, and corn/wheat rotations in the North China Plain and Northwest. The study does not adequately address this diversity. It would be beneficial for the CIrrMap250 to provide detailed mappings for irrigation methods for associated crop types, if possible. Moreover, the integration of county-level yearbook data on irrigated crop types and rotations could enhance the map's specificity and utility. Clarifying how different cropping systems and crop types are distinguished would significantly improve the comprehensiveness of the methodology. For example, L123 - Mapping 30-m CCropLand30 cropland layer (available every 5-year) - does this dataset also tell you which crop type is associated with each pixel?

2 Although the CIrrMap250 is purportedly an annual dataset, the primary analyses are based on three specific years (2000, 2010 and 2020). Although Figure 9 presents 20-year timeseries of irrigated croplands in different regions, it is crucial to present the interannual variability of irrigation areas. Analyzing annual data across the entire 20-year period can reveal the influence of climatic factors, such as temperature and precipitation, on irrigation trends. Additionally, showcasing irrigation transitions in various regions, beyond the highlighted area between CSC and NC, would provide a more comprehensive view of national trends.

3. Since this data product is a fusion of data from multiple sources - would it be good and necessary to quantify uncertainties of different sources? Such as the assumption outlined in 2.2.1, L151-155, and also in 2.2.2 L164-184. Section 5.2 L486-502 discussed the uncertainties

and limitations, what are the limitations associated with integrating multiple sources? For example, for the same region, how are remote-sensing indices, statistics and survey data differ from each other (or not)? In which regions does each index perform better? Please give some discussion as it may be useful to evaluate CIrrMap250 product and future user's information.

4. The study employs numerous data products and indices, yet lacks clear definitions and descriptions. A detailed table in the main text or supplementary material, listing all data products used, indices defined, could enhance clarity. Including equations for calculating indices such as PET, aridity index, Water Scarcity Index (WSI), NDVI, EVI, and GI is essential for transparency. Defining WSI and its components, including whether groundwater pumping in North China is considered, would further elucidate the methodology. Providing detailed and comprehensive information, similar to the content in Tables S1-S3, would greatly benefit readers.

5. 3.1 L219 - can you also give a brief description of the threshold-calibration method, instead of stating "following the previous studies" - let the readers of this paper understand your method is important. Particularly Equation 5 - how do you determine threshold? Since you have 20-year data, is this threshold constant? Or it changes year-by-year, please elaborate.

6. L223-226, please elaborate on this statement - "Cropland with lower elevation, gentler slope and higher aridity index was hypothesized to have higher irrigation suitability and potential" - is this statement a hypothesis? Or has already been demonstrated in Liu et al. It is not clear by now.

7. L275-280 the description of Figure 2c and in the figure caption don't match - Figure 2c shows 2010, but the texts indicate 2020. Please check your texts and figure captions.
Also, since you can identify the center pivotal system, can you also provide an irrigation method map, distinguishing between sprinkler irrigation (mostly in North China) and flood irrigation (more common in South China)?

8. L337-340, please define WSI. What are the major water resources used in WSI? How about considering pumping groundwater for irrigation in North China, is it a part of WSI calculation and evaluation of the irrigation map?

9. Figure 5, please give a scale legend for region A, B, C, and D. i.e. how big are these four regions?

10. Figure 8. The transition only shows three years ,2000, 2010, and 2020. What about the interannual variability, since you have 20-year annual data. It would be good to show an interannual timeseries of irrigation transitions across China.
Figure 8 highlights an area between CSC and NC, how about other regions in China? What are the transitions across regions over the 20-year period?

11. Please also discuss the potential use of CIrrMap250, who will be interested in using this data? Science communities, Hydrologic models? Climate models?, or water resource managers?

**Minor comments:**
1: L17-20: "...reconciled them with remote sensing data … integrated with multiple remote sensing data …" These two sentences seem redundant.
2: L94: "both cropland and other land use"
3: 231 "with the assumption that"
4: 3.3 L265-270 and 3.3.3 L310-323, these texts seem similar and redundant.

---

## Author Comment (AC1)

**Responses to the comments of Referee #1**

**Article ID:** essd-2024-2
**Title:** CIrrMap250: Annual maps of China's irrigated cropland from 2000 to 2020 developed through multisource data integration
**Authors:** Ling Zhang, Yanhua Xie, Xiufang Zhu, Qimin Ma, Luca Brocca

Dear Reviewer,

Thank you very much for the great efforts on our manuscript. Inspired by your valuable comments, we have made a major revision to our manuscript. The key revisions include:

(1) The data description has been carefully rewritten to avoid any potential misinterpretations by users.
(2) Additional experiments have been conducted to provide further explanation of our methodology.
(3) Additional information and discussion regarding our results have been incorporated.
(4) Many paragraphs, sentences, and figures have been revised to improve readability, conciseness, and clarity.

The detailed point-to-point responses are as follows. Texts in red are the reviewer's comments; **those in black** are our responses to the reviewer's comments; and *those in blue and italics* are the revised texts appeared in the revised manuscript.

We will finalize the revised manuscript once we have received comments from the second reviewer. At that stage, we will attach a clean version (essd-2024-2_Manuscript_Clean_Version.docx) as well as a tracking enabled version (essd-2024-2_Manuscript_Marked_Version.docx) with editing marks for your reference.

The manuscript "CIrrMap250: Annual maps of China's irrigated cropland from 2000 to 2020 developed through multisource data integration" applies a random forest algorithm to classify and produce a new irrigation map product (CIrrMap250) over China at 250m resolution. The authors evaluate the new maps quantitatively and qualitatively (using reference data, withdrawal data and other existing irrigation products) over the 2000-2020 period. Generally, the paper is properly structured. It is well suited for this journal. However, the manuscript and supporting document appear rushed with several inconsistencies and mistakes. Some remarks:

• Check (and re-check) all the reported details. I.e., the performance metrics and other variables in the figures, tables and elsewhere in the manuscript (and the supplementary document). Please correct all inconsistencies. More below.

**Response:** Thanks for your detailed and valuable comments. We sincerely apologize for the mistakes we made in the original manuscript. We have thus carefully read through the revised manuscript and supplementary file, including figures, equations, tables, and text. please refer to our point-by-point response below for further details.

• What is the definition of 'irrigated cropland' as used in this study? At first I was rather intrigued when the authors mentioned in the initial sections that their product gives the irrigated cropland (which I interpreted as the fraction of vegetation cover that is actually irrigated). On further reading, however, it seemed the authors were only labeling the pixels as either irrigated [1] or not [0] and then presenting the total fraction vegetation cover (FVC) of [1] as the 'irrigated cropland' …is my understanding correct? If this is the case, what differentiates this product from a binary [1,0] irrigation map that is combined with the many (readily available) FVC products. Actually, one would argue that the latter method is better as it is not prone to misinterpretation by the user. Users are likely to misinterpret the produced CIrrMap250 irrigation maps to mean the ACTUAL irrigated pixel proportion and not the total vegetation cover. Also, how do you address pixels that have possibly been assigned an FVC of ~0 (e.g. at early growth stages) but have an [actual] irrigated area/extent larger than 0?

**Response:** Thanks for the valuable comments. Your understanding is correct. In this study, each 250-meter pixel was categorized as either irrigated or non-irrigated. No further classification was conducted to distinguish between irrigated and non-irrigated cropland at the subpixel level. Therefore, if a pixel was classified as "irrigated", it was assumed that all cropland within that pixel was irrigated.

The binary irrigation maps were spatially filtered and finally multiplied by the corresponding cropland mask layers to produce the annual maps of irrigated cropland in China (i.e., CIrrMap250). As a result, the pixel value of our product indicates the

percentage of a 250-meter resolution pixel covered by irrigated croplands (i.e., irrigated area / pixel area ×100). This post-processing step was implemented to consider the fractional coverage of croplands within moderate-resolution pixels, thereby enhancing the accuracy of irrigated area estimates in China where farms are typically small and fragmented. For instance, in a binary irrigation map, if 10 grids in a county are classified as "irrigated", the calculated irrigated area would be 250×250×10 = 625,000 m² without considering fractional coverage of cropland. However, if the cropland coverage within each grid in the county is only 50%, then the actual irrigated area should be halved, amounting to 312,500 m².

To mitigate any misinterpretations, we have explicitly clarified our product in the introduction and methodology sections. Additionally, we have removed phrases such as "irrigation cropland proportion", "fraction coverage of irrigated cropland", and "the mixed pixel issue" from our dataset descriptions.

*The newly developed irrigated cropland maps (CIrrMap250) feature a spatial resolution of 250 meters and an annual temporal resolution, spanning the period from 2000 to 2020. These maps show the percentage of each 250 m by 250 m pixel that is covered by irrigated cropland (i.e., irrigated area / pixel area ×100).*

*Finally, the binary, spatially filtered irrigation maps were multiplied by the corresponding cropland mask layers to produce the annual maps of irrigated cropland in China (i.e., CIrrMap250). As a result, the pixel value of our product indicates the percentage of a 250-meter resolution pixel covered by irrigated croplands (i.e., irrigated area / pixel area ×100). This post-processing step was implemented to consider the fractional coverage of croplands within moderate-resolution pixels, thereby enhancing the accuracy of irrigated area estimates in China where farms are typically small and fragmented.*

Consistent with prior researches (Zhu et al., 2014; Meier et al., 2018; Zhang et al., 2022a; Wu et al., 2023), irrigated cropland in our study is defined as cropland that is subject to irrigation. Consequently, a crucial step of mapping irrigated cropland involved selecting or generating suitable cropland mask layers. The classification of irrigated and non-irrigated cropland was exclusively conducted at the cropland grids (i.e., irrigated cropland was restricted to cropland areas). Thus, each irrigation map corresponds to a specific cropland mask. For example, CIrrMap250 utilized the cropland mask from the high-resolution (30-meter) hybrid cropland product (CCropLand30) (Zhang et al., 2024), while IrriMap_CN employed the cropland mask from the National Land Cover Dataset (NLCD) (Zhang et al., 2022a). Consequently, binary irrigation maps cannot be merged with other cropland masks due to significant disparities in cropland identification by different cropland datasets. For instance, a pixel classified as irrigated cropland in the irrigation map based on cropland mask A may

become non-irrigated if merged with another cropland mask B, as it may be classified as non-cropland in cropland mask B.

The Fraction of Vegetation Cover (FVC) typically represents the percentage of ground covered by green vegetation, ranging from 0% to 100%. However, in our study, fraction coverage of cropland denotes the proportion of cropland area within the 250-meter grids, which was derived from our high-resolution hybrid cropland product. For example, a pixel value of 0.2 in the cropland mask layer indicates that 20% of the 250-meter grid is covered by cropland. The classification of irrigated and non-irrigated cropland was exclusively performed on the cropland grids identified by cropland masks. Cropland proportion in each pixel was assumed to remain unchanged throughout the year in our study and other similar studies (Zhu et al., 2014; Meier et al., 2018; Zhang et al., 2022a; Wu et al., 2023).

- The CIrrMap250 product is limited to China. Have the authors considered applying a similar methodology to other regions, e.g. extend it globally? Obviously, training and test datasets from other global sites would be required, but would it be viable to apply your RF classifier/model (as-is) to other regions beyond China? What would be the limitations?

**Response:** In this study, we developed CIrrMap250 by integrating multisource data through a semi-automatic training approach (Zhang et al., 2022d; Xie et al., 2019). While our irrigation mapping method is applicable to other regions worldwide, we acknowledge that its effectiveness largely depends on the availability and reliability of multisource datasets, particularly those related to irrigation area statistics and surveys. This dependency stems from our methodology's framework, which uses a threshold-calibration method to generate training samples for each county in China based on remote sensing data, irrigation area statistics/surveys, and irrigation suitability maps. Consequently, the random forest models trained in this study were customized for China and may not be directly transferable to other regions due to significant variations in irrigation practices, and geographical and climatic characteristics (Salmon et al., 2015; Zhang et al., 2022d).

Specific comments :

L16: "… and considered the fraction coverage of irrigated cropland (i.e., the mixed pixel issue). In this study, we addressed these important gaps …" - This is somewhat misleading as the mixed pixel issue is not addressed in this manuscript. I was expecting that the authors were referring to 'mixed pixel' in terms of irrigation, i.e. proportion of

the fraction vegetation cover (FVC) that is irrigated or not. If not mistaken, the only consideration here is the total FVC, which is provided within most RS products anyway, and can thus be similarly combined (rather straightforwardly) with any available binary/boolean [1,0] irrigation maps. Also see your comment in L495 : "CIrrMap250 cannot differentiate irrigated and rain-fed croplands at the subpixel scales. There are many small and fragmented croplands in … with complex terrain and diverse vegetation types. CIrrMap250 should be used with caution in these regions due to the wide existence of the mixed pixels"

**Response:** Yes, we agree that this is confusing as the issue of mixed pixels has not been explicitly addressed in our work. We have revised the sentence to mitigate any confusion.

*Accurate maps of irrigation extent and dynamics are important to study food security and its far-reaching impacts on Earth systems and the environment. While several efforts have been made to map irrigated areas in China, few of them have provided multi-year maps, incorporated national land surveys, addressed data discrepancies, or considered the fractional coverage of cropland within moderate-resolution pixels.*

Actually, here, we intend to highlight a gap in previous studies, wherein binary cropland masks were utilized for irrigation mapping. In such masks, each pixel is classified eighter as cropland or non-cropland, disregarding the fractional coverage of cropland within the moderate-resolution pixels. This may lead to overestimations or underestimations of the extent of irrigated cropland due to the following two reasons. First, many studies generated the cropland mask layers by resampling the original 30-meter cropland data to moderate resolution (e.g., 1 km or 500 m). This resampling process could overlook cropland that covers a relatively small proportion of the moderate-resolution grid, while overestimating cropland in grids that are not totally covered by cropland. Secondly, the threshold-splitting method used in this study was commonly used in conjunction with irrigated area statistics to depict the spatial distribution of irrigated cropland; and this method relies on the assumption that the spatially allocated irrigated area should be equal to the statistics. If it is assumed that each grid cell is fully covered by cropland, the extent of irrigated cropland may be significantly underestimated. For instance, if the statistical irrigated area of a county is 625,000 $m^2$, and 10 grids (pixel area:250×250 = 62,500 $m^2$) would be classified as irrigated cropland in a binary cropland mask. However, if the cropland proportion within each grid in the county is only 50%, then in reality, 20 grids should be classified as irrigated cropland.

L17: "… named as CIrrMap250 …" – consider describing all abbreviations such as CIrr before use.

**Response:** The abbreviation "CIrrMap250" has been explained.

*In this study, we addressed these important gaps and developed new annual maps of China's irrigated cropland from 2000 to 2020, named as CIrrMap250 (China's irrigation map, with a spatial resolution of 250 m).*

L23: "… accuracy of 0.79-0.88 for years 2000, 2010, and 2020, respectively" - only for years 2000, 2010, 2020? What about the other years in between? Is it because the evaluation data were only available for those 3 years? If so, make it a bit clear here.

**Response:** Yes, the evaluation was conducted only for years 2000, 2010, and 2020, because the reference data were only available for the 3 years.

*Our evaluation results showed that CIrrMap250 agreed well with the available reference points for the years 2000, 2010, and 2020, attaining an overall accuracy of 0.79-0.88.*

L42: its' >> its

**Response:** Sorry for our carelessness. It has been revised.

*Given the vital importance of irrigation, it is essential to know its precise location and dynamics.*

L45-46: "While numerous land use/cover and thematic cropland products have been made available to the public, they often lack information on irrigation status …" - Why would it be important to provide land use land cover (LULC) maps with irrigation status information? Should rain/precipitation or evapotranspiration information be provided within LULC maps/products as well?

**Response:** We agree with you. This sentence has been removed in the revised manuscript.

L51: "…normalized difference water index (NDWI)…" - Note that there is another index that goes by the same name but used to detect floods/open water bodies (NDWI, McFeeters (1996)) – so it could ideally be used to map areas that employ flood irrigation

(rice paddies, for example).

**Response:** Thank you for this reminder. The Normalized Difference Water Index (NDWI) proposed by Gao (1996) is known for its sensitivity to both soil and plant water content, making it a valuable tool for monitoring rice paddy fields (Dong et al., 2016; Singha et al., 2019). Consequently, it was utilized in this study as well as many other studies (Deines et al., 2017; Deines et al., 2019; Xiang et al., 2020; Zhang et al., 2022), for mapping irrigated areas. Regarding the NDWI proposed by McFeeters (1996), we acknowledge its potential utility for mapping areas employing flood irrigation, such as rice paddies. We have incorporated this reference into the revised manuscript.

L57: "…been applied to detected irrigate areas…" >> …to detect irrigated areas

**Response:** Sorry for our carelessness. It has been revised.

*Moreover, remotely sensed soil moisture from microwave and optical sensors has also been applied to detect irrigated areas by using threshold splitting methods (Yao et al., 2022), supervised/unsupervised classification algorithms (Dari et al., 2021; Gao et al., 2018), and remote sensing-modelling comparison approaches (Zohaib and Choi, 2020; Zaussinger et al., 2019)*

L74: "China is a big agricultural country with the \*largest irrigated area in the world …" – any reference for this?

**Response:** Yes, we have added the reference.

*China is a big agricultural country with the largest irrigated area in the world (International Commission on Irrigation and Drainage, 2018)*

*International Commission on Irrigation and Drainage: World Irrigated Area-2018, https://www.icid.org/world-irrigated-area.pdf, 1-6, 2018.*

L85: "… in paces …" – do you mean places?

**Response:** Yes, it's a type error. It has been revised.

*As a result, it remains unclear where the expansion of irrigated area is water-sustainable (i.e., irrigated area expanded in places without experiencing water stress) (Mehta et al., 2024).*

L97: "many other studies " – which studies? Add some reference[s] here

**Response:** We have added the related references.

*Finally, it is worth noting that, apart from the study by Zhang et al. (2022a), many other studies assessed their irrigation maps with a relatively limited number of reference samples, potentially compromising the reliability of their evaluation results (Zhu et al., 2014; Zhang et al., 2022d; Xiang et al., 2020; Bai et al., 2022).*

L104-105: "CIrrMap250) have a spatial resolution of 250 meters and describe irrigated cropland distribution through fractional coverage" – what of the temporal resolution? Also, as already mentioned above, this statement is misleading as one could assume you are providing the fraction of total FVC that is under irrigation.

**Response:** The newly developed irrigated cropland maps (CIrrMap250) have an annual temporal resolution. The phrase "describe irrigated cropland distribution through fractional coverage" has been removed to prevent potential confusion. We have rewritten the data descriptions in the introduction and methodology sections. Please refer to our response to your first comment.

*The newly developed irrigated cropland maps (CIrrMap250) feature a spatial resolution of 250 meters and an annual temporal resolution, spanning the period from 2000 to 2020. These maps show the percentage of each 250 m by 250 m pixel that is covered by irrigated cropland (i.e., irrigated area / pixel area ×100).*

*Finally, the binary, spatially filtered irrigation maps were multiplied by the corresponding cropland mask layers to produce the annual maps of irrigated cropland in China (i.e., CIrrMap250). As a result, the pixel value of our product indicates the percentage of a 250-meter resolution pixel covered by irrigated croplands (i.e., irrigated area / pixel area ×100). This post-processing step was implemented to consider the fractional coverage of croplands within moderate-resolution pixels, thereby enhancing the accuracy of irrigated area estimates in China where farms are typically small and fragmented.*

L113-116: "These indices were generated every 16 days with a spatial resolution of 250 meters…" – to be consistent with other descriptions in the section, provide the product number of the vegetation indices product; is it MOD13Q1? " …band 4 …band1" – consider adding the spectral ranges here as well

**Response:** Yes, the product number is MOD13Q1. We have added the product number

as well as the spectral ranges for the bands.

*We collected the Terra Moderate Resolution Imaging Spectroradiometer (MODIS) vegetation indices, i.e., NDVI and Enhanced Vegetation Index (EVI) (Huete et al., 1997), from the NASA's Earth Science Data Systems (https://www.earthdata.nasa.gov/). These vegetation indices (MOD13Q1) were generated every 16 days with a spatial resolution of 250 meters. Meanwhile, the surface spectral reflectance of MODIS band 4 (545-565 nm) from the MOD09A1 product was resampled from the original 500 meters to 250 meters using simple nearest-neighbor interpolation (Debeurs and Townsend, 2008). These resampled data were used alongside with the 250-meter and 8-day surface reflectance of band 1 (620-670 nm) from the MOD09Q1 product to derive the Greenness Index (GI) (Supplementary Table S1).*

L119: "Greenness Index (GI) (Supplementary Table S1)" – in Table.S1 (supplementary document) under GI, you write the 'formula' as GI=NIR/green, and 'MODIS bands' as 'Bands 01, 04'. The sub-caption however reads: 'Red: band 01' and 'Green: band 04' …which is which? Please correct.

**Response:** Apologies for our carelessness. In calculating the GI index, we utilized 'Bands 02, 04' instead of 'Bands 01, 04'. We have revised the table accordingly, as listed below for your reference.

*Table S1. Summary of the MODIS-derived vegetation indices used in this study*

| Vegetation indices | Formula | MODIS bands | Resolution |
|---|---|---|---|
| NDVI | (NIR - Red) / (NIR + Red) | Band 01 (Red)
 Band 02 (NIR) | 250 m/16 day |
| EVI | 2.5*(NIR-Red) / (NIR+ 6*Red–7.5*Blue+1) | Band 01 (Red)
 Band 02 (NIR)
 Band 03 (Blue) | 250 m/16 day |
| GI | NIR/Green | Band 02 (NIR)
 Band 04 (Green) | 250 m/8 day |

*where NIR is the near-infrared band (841-876 nm), and Red (620 – 670 nm), Blue (459-479 nm) and Green (545-565 nm) are the are the visible red band, visible blue band, and visible green band, respectively.*

L121: "… The data for unreliable pixels were reconstructed using a straightforward nearest neighbor interpolation method…" - is this the right way to go about it? For example, for an overcast pixel (which is maybe vegetated), why would you take the remotely sensed spectral signal of the next/closest cloud-free pixel (which is maybe urban/built-up)? Meaning you may end up missing vegetated pixels under irrigation or vice versa. Why not just drop such pixels from your analysis (i.e. at that particular time)?

**Response:** We completely understand your concern. Indeed, directly applying the interpolation method to all MODIS data in China could significantly impact the results. As you pointed out, if the neighboring pixel with reliable data is located in an urban or built-up area, the reconstructed pixel is likely to be erroneously excluded from irrigated cropland due to the low value of vegetation index. However, in this study, we actually extracted MODIS data only for cropland pixels in China. For cropland pixels with unreliable data, their values were interpolated from the nearest neighboring cropland pixels with reliable data. This approach helps to avoid interpolating data for cropland pixels from areas covered by other land use types, such as urban and forest. We have provided a more detailed explanation in the revised manuscript.

*We extracted MODIS data for all cropland pixels in China. In cases where cropland pixels had unreliable data, their values were interpolated from the nearest neighboring cropland pixels with reliable data.*

We chose not to exclude pixels with unreliable data at a particular time from our analysis because our mapping process relies heavily on the peak values of MODIS vegetation indices during the growth period. Omitting pixels with unreliable data for a specific time could potentially result in unreliable peak values of vegetation indices, thereby affecting our mapping results.

L157-159: "In years lacking survey data, the harmonized irrigated area was determined using Eq. 2, assuming that the relative changes in statistical irrigated area are reliable" - could you explain the rationale behind Equation (2)? How to interpret it? to me it appears that a year without survey data could end up having a lower assigned/harmonized irrigated area despite having a larger irrigated [statistical] area without land survey (Astatt2). For instance, if we assume: Aharmts=20, Astatts=20, Astatt2=30, CAsurvts=40 ; then Aharmt2 becomes min(20*(30-20)/20,40)=10?
···the harmonized value (Aharmt2) even becomes negative if we consider Astatt2 to be less than Astatts. What am I missing? Please clarify.

**Response:** We apologize for the typographical error in Equation 2, where the relative changes of the statistical irrigated area should plus one before being multiplied with

$A_{harm}^{ts}$. The correct equation should be:

$$A_{harm}^{t2} = min\,(A_{harm}^{ts} \times (1 + \frac{A_{stat}^{t2} - A_{stat}^{ts}}{A_{stat}^{ts}}), CA_{surv}^{ts}) \qquad (2)$$

For your example, Aharmts=20, Astatts=20, Astatt2=30, CAsurvts=40; then Aharmt2 becomes min (20*(1+(30-20)/20),40) =30. In this case, the relative change of the statical irrigated area is (30-20)/20*100=50%. Consequently, the harmonized data in the survey year should be adjusted by increasing 50%, i.e., 20*(1+0.5) =30. This ensures that the relative changes between Aharmt and Aharmt2, and between Astatts and Astatts2 remain consistent. This process preserves the interannual changes observed in the statistical irrigated area while enhancing data consistency across years. For instance, in a span of five years lacking survey data, the recorded statistical irrigated areas are 11, 12, 13, 14, and 15 hectares respectively, whereas the reconciled irrigated areas in adjacent years with survey data might amount to 101, 102, 103, 104, and 105 hectares. Without the aforementioned adjustment, notable data inconsistencies would arise. In the revised manuscript, we corrected Equation 2, and meanwhile, double-checked all other equations to ensure their correct formulation.

*In years lacking survey data, the irrigated area was determined by adjusting the harmonized data in adjacent survey year using relative change information derived from irrigated area statistics (Eq. 2). This method preserved the interannual changes observed in statistical irrigated area while enhancing data consistency across years.*

$$A_{harm}^{t2} = min\,(A_{harm}^{ts} \times (1 + \frac{A_{stat}^{t2} - A_{stat}^{ts}}{A_{stat}^{ts}}), CA_{surv}^{ts}) \qquad (2)$$

*where $A_{harm}$, $A_{stas}$ and $A_{surv}$ represent harmonized, statistical and surveyed areas of irrigated cropland, respectively; $CA$ is surveyed area of cropland; and $ts$ and $t2$ indicate the year with and without land surveys, respectively.*

L189: "… used in combination with the MCD43A3 albedo product" - this is a daily product. Did the authors calculate the daily PET? how did you reconcile this with the other 8/16-day products?

**Response:** Yes, the MCD43A3 albedo is a daily product, so we computed the daily PET accordingly. These daily PET values were summed to annual values spanning from 2000 to 2020. In the revised manuscript, we have clarified this point.

*These datasets were combined with the MCD43A3 albedo product to compute daily potential evapotranspiration (PET) using the Priestley-Taylor method (Priestley and Taylor, 1972). The daily PET values were summed to obtain annual values for the period from 2000 to 2020. These annual PET values were then used to derive the aridity index,*

*defined as the ratio of precipitation to PET.*

In terms of the other 8/16-day products, such as NDVI, EVI, and GI, we utilized their annual peak values during the growth period in this study, rather than directly employing their original values. Consequently, both the estimated PET and other MODIS products were utilized at annual scales.

L211: "… were then then" >> were then

**Response:** Revised.

*The mapping outcomes were then mosaicked and post-processed to obtain annual maps of irrigated cropland in China, denoted as CIrrMap250.*

L222: "*A static irrigation suitability map *were constructed based on …, and aridity index of cropland" - was this one map or several ('*A static' then '*were'). If one, why was the temporal variation of the aridity index not considered?

**Response:** In this study, we utilized a single and static irrigation suitability map. The concept of integrating irrigation suitability into the mapping process was inspired by previous researches assessing land potential for irrigation (Worqlul et al., 2015; Worqlul et al., 2017; Li and Chen, 2020; Zhang et al., 2022b). The aridity index served as a metric reflecting climate suitability for irrigation, where lower values indicate a higher deficit of available water for crops, thereby suggesting a greater need for irrigation. However, aridity is typically measured by comparing long-term average water supply (precipitation) to long-term average water demand (evapotranspiration) (Zomer et al., 2022), rather than through the lens of a single year's data. Consequently, in our study, we utilized a 21-year average aridity index (covering the period 2000-2020) in conjunction with elevation and slope data to produce a static irrigation map for China.

Regarding your concerns, we conducted an additional experiment to evaluate the influence of temporal variations in the aridity index on irrigation mapping results. Specifically, we computed the aridity index for each year and applied it to derive annual irrigation suitability maps spanning from 2000 to 2020. The resulting 21 suitability maps were then applied to each corresponding year to generate irrigated cropland maps using the methodology outlined in our study. We kept other factors the same to ensure that any disparities in the mapping results from our original ones are solely attributed to the use of annual irrigation suitability maps. As shown in Figure R1, the incorporation of annual irrigation suitability maps has a negligible impact on the accuracy of irrigation maps.

[Figure]

Figure R1. Accuracy of irrigation maps derived from the experiments with static and varying irrigation suitability maps, respectively. The static irrigation map was derived from elevation, slope, and the 21-year (i.e., 2000-2020) averaged aridity index, while the varying irrigation suitability maps from elevation, slope, and annual aridity index.

L230: "… (Supplementary Table S2)" – in Table S2 (supplementary document), why do you have the same 'Suitability value' for the lowest suitability classes S3 and S4. I.e., for the 'elevation' and 'slope' irrigation suitability factors, S3=2 and S4=2.

**Response:** It appears to be an unintentional error in the original table. For the irrigation suitability factors 'elevation' and 'slope', the correct values should be S3=2 and S4=1. The suitability values for each factor should follow a monotonic pattern, with higher values indicating greater suitability for irrigation. We have corrected this unintentional error and have also double-checked all the other tables in our manuscript to avoid similar typo errors.

*Table S2. Suitability values for the influencing factors of irrigation suitability*

| Influencing factors | Reclassification | Suitability value |
|---|---|---|
| elevation | S1: < min+100 | S1=4 |
| | S2: [min+100, min+300] | S2=3 |
| | S3: [min+300, min+500] | S3=2 |
| | S4: > min+500 | S4=1 |
| slope | S1: <2% | S1=4 |
| | S2: [2%, 4%] | S2=3 |
| | S3: [4%, 8%] | S3=2 |
| | S4: > 8% | S4=1 |
| aridity index | S1: <0.1 | S1=10 |
| | S2: [0.1, 0.2] | S2=9 |

| | |
|---|---|
| *S3: [0.2, 0.3]* | *S3=8* |
| *S4: [0.3, 0.4]* | *S4=7* |
| *S5: [0.4, 0.5]* | *S5=6* |
| *S6: [0.5, 0.6]* | *S6=5* |
| *S7: [0.6, 0.7]* | *S7=4* |
| *S8: [0.7, 0.8]* | *S8=3* |
| *S9: [0.8, 0.9]* | *S9=2* |
| *S10: >0.9* | *S10=1* |

*Note: min is minimum elevation of the mapping unit*

L258: "and time-invariant environmental variables (i.e., latitude, longitude, crop intensity" – why is the crop intensity considered time-invariant?

**Response:** In this study, we utilized a time-invariant crop intensity dataset (Xu, 2017) due to the lack of publicly available annual crop intensity dataset when our work was conducted. While an annual dynamic global cropping intensity dataset is available for the years covering 2001-2019 (Liu et al., 2021), employing this dynamic dataset did not yield improvements in mapping accuracy; in fact, it resulted in a slightly decreased accuracy, potentially attributable to its relatively lower precision in China. The test results, as depicted in Figure R2, are provided for your reference. Consequently, we opted to utilize the available time-invariant crop intensity data for this study.

[Figure]

Figure R2. Comparison of irrigation mapping accuracy for the year 2000, 2010, and 2020 in the experiments of using time-invariant and dynamic crop intensity datasets, respectively.

L261: "To enhance the accuracy of these maps, a spatial filter (a 7x7 window)…" - clarify what you mean by this. Why 7x7? …'constituting <5% of the window area' is ambiguous. Is the 250 m resolution retained after this?

**Response:** After classification, we merged the annual, county-level mapping results to generate preliminary maps of irrigated cropland in China. Finally, a spatial filtering was applied to improve the accuracy of these maps. Specifically, for each target irrigated pixel, we calculated the ratio of the number of irrigation grids to the total number of grids within a moving window. If the calculated ratio fell below 5%, we assigned all cropland grids within the moving window as "non-irrigated". Conversely, if the ratio exceeded 95%, we assumed all cropland grids (i.e., cropland proportion > 0) within the moving window to be irrigated. This post-processing step preserved the original spatial resolution of the maps (250 meters), removed isolated irrigation pixels, and also identified potentially omitted irrigated croplands. The size of the moving window was determined to be 7×7 pixels through a trial-and-error process. We tested three different window sizes (5×5, 7×7, and 9×9) in the post-processing step and found that the 7×7 window size yielded the highest mapping accuracy.

In the revised manuscript, we have clarified the spatial filtering process.

*A spatial filtering technique was then employed to enhance the accuracy of these maps. Specifically, for each target irrigated pixel, we calculated the ratio of the number of irrigation grids to the total number of grids within a moving window. The size of the moving window was determined to be 7×7 pixels through a trial-and-error process. If the calculated ratio fell below 5%, we then assigned all cropland grids (i.e., cropland proportion>0) within the moving window as "non-irrigated". Conversely, if the ratio exceeded 95%, we assumed all cropland grids within the moving window to be irrigated. The spatial filtering operation preserved the original spatial resolution of the maps (250 m) and removed isolated irrigation pixels, while also identifying potentially omitted irrigated croplands.*

L276: "were acquire from …" >> were acquired

**Response:** Revised.

*The validation samples for the year 2020 were acquired from Chen et al. (2023), who mapped the center pivot irrigation systems (CPIS) in global arid regions.*

L282-: "Due to the lack of georeferencing information, we georeferenced these land use maps using the georeferencing tool in ArcGIS in conjunction with high-resolution

images " – the authors do not talk about the data that were used to serve as ground control points for the georeferencing (e.g. How many GCPs, their spatial distribution, …?)

**Response:** In total, we selected 234 control points nationwide, primarily distributed along provincial boundaries. In the revised manuscript, we added the information on the georeferencing points, and provided the spatial distribution map of these points (see below) in the supplementary file.

*Due to the lack of georeferencing information, we georeferenced these land use maps using the georeferencing tool in ArcGIS in conjunction with high-resolution images. A total of 234 control points were selected nationwide (Supplementary Figure S1), primarily situated along provincial boundaries, to facilitate the georeferencing process. The irrigated samples were taken from the patches of irrigated lands and paddy fields in the georeferenced land-use maps, while non-irrigated samples were taken from dryland patches.*

[Figure]

*Figure S1. Spatial distribution of the identified reference points used for georeferencing the provincial land-use maps of the second National Land Survey in China*

L294-: "It's noteworthy that this percentage represents the proportion of cropland within the 250 …, not the proportion of irrigated cropland to total cropland" ; L362: "irrigated cropland in CIrrMap250". As already mentioned, giving the irrigated cropland as a percentage is very likely to mislead users into assuming that your

irrigation product provides the proportion of the total fraction of vegetation cover (FVC/cropland) that is irrigated. If feasible, wouldn't it be more useful to have both products, i.e. the total fraction cover product and the proportion of that that is deemed irrigated? The authors also acknowledge in L492 that "…cirrmap250 has a relatively coarse resolution". You may still argue that at the relatively higher spatial resolution of 250m, one could assume the whole cropland (total FVC) to be equivalent to the irrigated area. This might be true but still needs validation to avoid being misleading.

**Response:** To prevent misinterpretation, we have clarified our product in in the introduction and methodology sections.

*The newly developed irrigated cropland maps (CIrrMap250) feature a spatial resolution of 250 meters and an annual temporal resolution, spanning the period from 2000 to 2020. These maps show the percentage of each 250 m by 250 m pixel that is covered by irrigated cropland (i.e., irrigated area / pixel area ×100).*

*Finally, the binary, spatially filtered irrigation maps were multiplied by the corresponding cropland mask layers to produce the annual maps of irrigated cropland in China (i.e., CIrrMap250). As a result, the pixel value of our product indicates the percentage of a 250-meter resolution pixel covered by irrigated croplands (i.e., irrigated area / pixel area ×100). This post-processing step was implemented to consider the fractional coverage of croplands within moderate-resolution pixels, thereby enhancing the accuracy of irrigated area estimates in China where farms are typically small and fragmented.*

As you pointed out, one potential approach to avoid the misinterpretation is to provide both binary irrigated cropland maps and cropland mask layers (representing fractional coverage of cropland) to users. While we acknowledge the merits of this method, we opted not to implement it for the following reasons. Firstly, data users may misinterpret that pixels with value equals to 1 are fully irrigated, and may directly utilize the binary irrigation maps for their research, such as estimating irrigation water use or assessing the hydroclimatic impact of irrigation. However, relying solely on these maps could bring significant biases into their results. Secondly, users would need to combine the binary irrigation maps with the cropland mask layers, a process that may introduce errors and increase the risk of generating irrigation maps divergent from those we have released. Lastly, providing only irrigation maps, instead of both binary irrigation maps and cropland mask layers, aligns with the practices of other similar studies (Zhu et al., 2014; Meier et al., 2018; Xie and Lark, 2021; Zhang et al., 2022a; Wu et al., 2023).

L340: "… under severe to extreme…" - In the previous sentence (L339), only low, moderate, high and severe WSI ranges are described. What is the extreme WSI range? Is extreme synonymous to severe here?

**Response:** The levels of water stress should be categorized as: low (WSI ≤ 0.2), moderate (0.2<WSI≤0.4), severe (0.4 < WSI≤1.0), and extreme (WSI>1), in line with our previous study (Zhang et al., 2023). It has been revised in the new manuscript.

*The WSI denotes the fraction of available water resources appropriated by humans and is employed to categorize water stress into four levels: low (WSI≤0.2), moderate (0.2<WSI≤0.4), severe (0.4 < WSI≤1.0), and extreme (WSI>1) (Zhang et al., 2023b). Expansions of irrigated areas under severe to extreme water stress were designated as "unsustainable" due to their potential to exacerbate the depletion of surface water and groundwater resources (Mehta et al., 2024). Conversely, expansions of irrigated areas under low to moderate water stress or reductions in irrigated areas under severe to extreme stress were deemed "sustainable".*

L352: "CIrrMap250 and IrriMap_CN performs similarly in user's accuracy…" – TableS5 (supplementary document) shows a user accuracy (UA) of 1 (error of commission=0). Can this perfect UA be explained? From Fig3c/TableS2 (year 2020), IrriMap_CN has a producer accuracy (PA) of 0.2, why this huge discrepancy between the [perfect] irrigated.UA (1) and the [rather poor] irrigated.PA (0.2)?

**Response:** In 2010, the reference points were extracted from the Center Pivot Irrigation Systems (CPIS) map developed by Chen et al. (2023). All of these reference points represent irrigated samples, as shown in the newly added confusion matrix (see below). Consequently, both CIrrMap250 and IrriMap_CN achieved a perfect user's accuracy. However, IrriMap_CN exhibited a low producer's accuracy of 0.2, as only 20% of the irrigated samples were correctly identified. We have clarified it in the revised manuscript.

*For the year 2020, CIrrMap250 detects 88% of the fields with center pivot irrigation systems, while IrriMap_CN identifies only 20% (Figure 3c and Supplementary Figure S2). Note that both CIrrMap250 and IrriMap_CN achieves a perfect user's accuracy in this year mainly because all the reference points are irrigated samples (Section 3.31 and Supplementary Table S6).*

Since you use ~20,000 samples in your classification exercise (into irrigated and non-irrigated), could you provide (in supplementary doc) the CIrrMap/IrriMap confusion matrices for 2000, 2010, 2020 to aid with interpretation (i.e. how many of the reference samples are irrigated or not? How do you split these into training and test sets? …more details on how the RF classifier used in CIrrMap250 performs, …)

**Response:** Thanks for your suggestion. As per your advice, we have incorporated the Confusion Matrix for CIrrMap250 and the existing maps (IrriMap_CN, IAAA, GFSAD) in 2000, 2010, and 2020, respectively. The Confusion Matrix presents the numbers of correctly and erroneously classified irrigated and non-irrigated samples by different products, thereby facilitating a more comprehensive understanding of our results. It's important to note that, in this study, the training samples were generated using a threshold-calibration method (refer to Section 3.1), rather than obtained from the reference points. All reference samples were independent of the training data and were utilized for the performance evaluation of irrigation maps.

*Table S6. Confusion matrix for CIrrMap250 and the existing maps (IrriMap_CN, IAAA, GFSAD) in 2000, 2010, and 2020, respectively*

| | Products | Classified | Reference | |
| --- | --- | --- | --- | --- |
| | | | Irrigated | Non-irrigated |
| 2000 | CIrrMap250 | Irrigated | 271 | 75 |
| | | Non-irrigated | 66 | 246 |
| | IrriMap_CN | Irrigated | 172 | 43 |
| | | Non-irrigated | 165 | 278 |
| | IAAA | Irrigated | 221 | 177 |
| | | Non-irrigated | 116 | 144 |

| | Products | Classified | Reference | |
| --- | --- | --- | --- | --- |
| | | | Irrigated | Non-irrigated |
| 2010 | CIrrMap250 | Irrigated | 6818 | 1385 |
| | | Non-irrigated | 1365 | 3325 |
| | IrriMap_CN | Irrigated | 5003 | 1167 |
| | | Non-irrigated | 3180 | 3543 |
| | IAAA | Irrigated | 5274 | 2183 |
| | | Non-irrigated | 2909 | 2527 |
| | GFSAD | Irrigated | 4939 | 1995 |
| | | Non-irrigated | 3244 | 2715 |

| | Products | Classified | Reference | |
| --- | --- | --- | --- | --- |
| | | | Irrigated | Non-irrigated |
| 2020 | CIrrMap250 | Irrigated | 6340 | 0 |
| | | Non-irrigated | 849 | 0 |
| | IrriMap_CN | Irrigated | 1426 | 0 |
| | | Non-irrigated | 5763 | 0 |

L366-367: "CIrrMap250 yields irrigation ratios (i.e., the ratio of irrigated area to the total cropland area) of…" – this sentence contradicts L294 (i.e., "… this percentage represents the proportion of cropland within the 250 …, not the proportion of irrigated cropland to total cropland"), and many other statements in this report (e.g. L495 "cirrmap250 cannot differentiate irrigated and rainfed croplads at the subpixel scales"). Such inconsistencies make it somewhat difficult to follow and interpret your results/analyses.

**Response:** In this stud, we classified each 250-meter grid cell as either irrigated or non-irrigated. The binary irrigation maps were finally multiplied by the corresponding cropland mask layers to produce the annual maps of irrigated cropland in China (i.e., CIrrMap250). As a result, the pixel value of our product indicates the percentage of a 250-meter resolution pixel covered by irrigated croplands (i.e., irrigated area / pixel area ×100). While our product does not provide the proportion of irrigated cropland area to total cropland area at the pixel scale, it can be utilized to determine irrigation ratio at the regional scale. Specifically, for a target region, we first calculate the irrigated cropland area and the total cropland area, respectively. Then, the area of irrigated cropland is divided by the total area of cropland to estimate the irrigation ratio for this region. We have revised the related sentence in the new manuscript.

*IrriMap_CN estimates the irrigation ratios (i.e., the ratio of irrigated cropland area to total cropland area) to be 0.47, 0.37, and 0.61, respectively, for China, Northern China, and Xinjiang Uygur Autonomous Region (Supplementary Figure S4). In comparison, the values derived from CIrrMap250 are 0.58, 0.70, and 0.96, respectively, which align more closely with reality and official reports (https://gtdc.mnr.gov.cn/).*

L370-374: "However, CirrMap250 tends … southern part of South China (SC)" – why? Could you discuss this section a little bit more. Readers may not go back to the literature on the other products to find out by themselves. Also, what does '*southern part of *South China' mean?

**Response:** Thanks for the suggestion. We have provided more explanation and discussion on those results. Additionally, we have revised the phrase "south part of South China" to simply "South China" to prevent any potential confusion.

*Nevertheless, CIrrMap250 tends to yield lower estimates of irrigation area in Northeast China (NEC) when compared to IrriMap_CN, possibly due to inaccurate statistical and survey data in this region. In contrast to CIrrMap250 and IrriMap_CN, IAAA notably underestimates irrigated croplands in Northwest China (NWC) and North China (NC), but overestimates them in NEC and Southwest China (SWC). This could*

*be explained by the fact that IAAA was developed using an unsupervised classification algorithm based mainly on vegetation dynamics (Siddiqui et al., 2016), limiting its ability to accurately depict the extent and spatial heterogeneity of irrigation in China (Tian et al., 2024). GFSAD shows overestimations of irrigated area in the Dujiangyan district and the North China Plain, but exhibits evident omission errors in sparsely distributed irrigation regions like NWC and South China (SC). The large bias of GFSAD is understandable, as it is not an irrigation-specific product and only considers five major crop types (Xie et al., 2021; Thenkabail et al., 2016).*

L388: Fig5 – This figure needs improvement. How come no irrigated pixels in zone B are detected by the 1Km GFSAD product?

**Response:** In the revised manuscript, we have improved the figure by thickening the borders of the subplots. Meanwhile, we have provided the legend for different products (as shown below). Regarding your comment on lacking irrigated pixels of GFSAD in Zone B, we have carefully reviewed the map and indeed found that it did not identify any irrigated cropland. This is because GFSAD notably underestimates irrigated cropland in Zone B and its surrounding regions. Furthermore, we cross-checked the comparison results of the irrigated cropland map with GFSAD in the study by Zhang et al. (2022a), they also reported significant underestimation of irrigated cropland by GFSAD, particularly in Southern China. The underestimation of irrigated cropland by GFSAD may be attributed to the fact that it is not an irrigation-specific product and only considers five major crop types (Thenkabail et al., 2016; Xie et al., 2021).

[Figure]

*Figure 5. Visual comparison of CIrrMap250 with the existing maps. The five rows from top to bottom correspond to the Google map, CIrrMap250, IrriMap_CN, IAAA and GFSAD, respectively. Locations of the four selected zones are presented in Figure 4a.*

L391: "Figure 6 …CIrrMap250 exhibits a robust agreement with OPTRAM3" - This is not clear from the figure. Qualitatively, Figure 6a may even be interpreted differently unless the authors have overlain CIrrMap250 over OPTRAM30. If that is the case, please find a better way to illustrate/present the map inter-comparisons.

**Response:** Thanks for the comment. In the previous version, Figure 6a presented overlays of CIrrMap250 on the OPTRAM30 map. We agree it is not easy to interpret. We have thus revised the figure. Specifically, we have depicted the irrigated cropland distribution in CIrrMap250, IAAA, IrriMap_CN, GFSAD, and OPTRAM30, respectively. Additionally, we have further compared CIrrMap250 and IrriMap_CN

with OPTRAM30 in two local zones to better illustrate the differences between CIrrMap250 and IrriMap_CN. We present the revised figure alongside the related descriptions below for your reference.

*Figure 6 provides an additional comparison of the aforementioned large-scale irrigation maps with the field-scale remote sensing irrigation map (OPTRAM30) in the Hexi Corridor of Northwest China. CIrrMap250 exhibits a robust agreement with OPTRAM30 in the distribution of irrigated cropland. While IrriMap_CN captures the general pattern of irrigated croplands in this region, it tends to underestimate irrigation extent, as demonstrated evidently in the two selected local zones (Figure 6d). The IAAA product struggles to identify irrigated cropland in this area, displaying significant omission and commission errors. Similarly, GFSAD has a limited ability to accurately depict irrigated areas in the Hexi Corridor.*

[Figure]

*Figure 6. Comparison of large-scale irrigation maps with the field-scale remote sensing irrigation map (OPTRAM30) in the Hexi Corridor of Northwest China. Panels a, b, c, e, and f depict the distribution of irrigated cropland in OPTRAM30, CIrrMap250, IAAA, IrriMap_CN, and GFSAD, respectively. Panel d presents the comparisons of CIrrMap250 and IrriMap_CN with OPTRAM30 in two local zones.*

Figure6 – the [0-100] color scale as provided in Figure4 is missing. In supplementary, Figure S2 (b, c, d) – the magenta color scale (for IrriMap, IAAA, GFSAD) is missing. Additionally, why was year 2019 selected in Figure6a,b (CIrrMap250/IrriMap_CN) for the comparisons with the 2014-2020 OPTRAM30 product?

**Response:** The color scale has been provided in the revised figure 6 (see above). The OPTRAM30 product was derived by counting the detected irrigation events over an

extended period (2014-2020) to complement for any missed detections. We opted to compare CIrrMap250 and IrriMap_CN with OPTRAM30 using data from the year 2019 for two reasons. Firstly, the authors of OPTRAM30 have utilized images from 2019 and 2020 to assess the spatial pattern of their irrigation maps. Second, IrriMap_CN covers the period from 2000 to 2019, while our product CIrrMap250 spans from 2000 to 2020. To maintain consistency between IrriMap_CN and CIrrMap250, we selected the year 2019 for data comparison.

Figure S2 becomes Figure S3 now, and it has been replotted in the new manuscript. We presented it here for your reference.

[Figure]

*Figure S3. Comparison of the distributions of irrigated cropland in CIrrMap250 with the existing products (IrriMap_CN, IAAA, GFSAD). Panel a displays the distribution of irrigated cropland in CIrrMap250, while panels b, c, and d overlay the irrigation maps IrriMap_CN, IAAA, and GFSAD on CIrrMap250, respectively.*

L406-407: "…, namely 2010 and 2020. The estimates of irrigated areas from the other two maps, namely IAAA and GFSAD, are able to explain only a small proportion of the variances in irrigation water withdrawals (i.e., 0.12 and 0.20) …" – Please clarify. According to Figure 7c,e, these (IAAS and GFSAD) metrics only apply to year 2010 NOT 2020.

**Response:** Yes, these metrics of IAAA and GFSAD only apply to the year 2010. We have clarified it in the revised manuscript.

*As shown in Figures 7c and f, the irrigated area estimates from the other two maps (i.e., IAAA and GFSAD) demonstrate limited explanatory power, explaining only 12% and 20% of the variation in irrigation withdrawals for the year 2010.*

L410: "…irrigated area estimates against irrigation water withdrawals…" – maybe you mean 'irrigated water withdrawals against irrigation area estimates…'? Y against X.

**Response:** We agree, and it has been revised.

*Figure 7. Scatterplots of irrigation water withdrawals against irrigated area estimates from different products for the years circa 2010 and 2020. The data are presented in logarithmic units to reflect both small and large values.*

L426: "As shown in Figure 9, all subregions exhibit an increasing trend in irrigated area from 2000 to 2020" - is this conclusion based on CIrrMap250 or some other [reference] data?

**Response:** Yes, the conclusion is based on CIrrMap250. We have revised the sentence.

*As shown in Figure 9, our annual irrigation maps (i.e., CIrrMap250) indicated that all subregions exhibited an increasing trend in irrigated area from 2000 to 2020, with NEC expanding significantly faster than the other subregions.*

L435-: Figure 9d - some of the percentage entries in the concentric pie charts are likely incorrect. I.e. percentages for years 2000 and 2020 add up to 101% (11+7+17+30+24+12) and 98% (11+11+16+26+22+12), respectively.

**Response:** Thanks for the reminder. We have checked the results carefully and revised the figure accordingly. We present the revised figure here for your reference.

[Figure]

*Figure 9. Changes in irrigated area across the six subregions of China during 2000-2020. a, Relative changes in irrigated area. b, Changes in China's total irrigated area, with the contribution of different subregions depicted in the inserted pie chart. c, Relative changes in the proportion of irrigated area. d, Proportion of irrigated area for the years 2000, 2010 and 2020.*

L445: "… The net expansion of irrigated area is about 180,000 …" but L427 reads "The irrigated area of China increases from 750,000 to 950,000…", which is ~200,000. Both for the 2000-2020 period. Please be consistent with the presented numbers.

**Response:** Thank you for the comment. We have double-checked our results and have confirmed that the irrigated area of China has increased from about 760,000 to 940,000 km$^2$ over 2000-2020, with a net increase of about 180,000 km$^2$. We have revised this sentence.

*The irrigated area of China increased from about 760,000 to 940,000 km2 at the rate of about 10,000 km2/year (or 1.29%/year).*

L465: "…leading to a decrease in irrigation mapping accuracy by 8%-26% (Supplementary Figure S4)." – do these numbers refer to supplementary Figures S3? They do not appear in Figure S4.

**Response:** Apologies for our carelessness. We have placed the figure in the wrong place. This sentence should be referred to Figure S5 in the revised manuscript. We present the figure below for your reference. Furthermore, we have carefully reviewed the figures in Supplementary file to avoid similar mistakes.

[Figure]

*Figure S5. Comparison of the performance of irrigated cropland maps constrained by different irrigated area data. "without adjustment" means the use of the original irrigated area statistics, while "with adjustment" indicates the use of the harmonized and reconciled irrigated areas (this study).*

The caption of Figure S3 reads "Comparison of irrigated ratio estimates of CIrrMap250 and IrrMap_CN in China, Northern China, Xinjiang Uygur Autonomous Region" … does this mean that this conclusion only applies to that specific part of China?

**Response:** Figure S3 has been erroneously presented in the original manuscript. We present the correct figure below for your reference. The sentence in the main texts that refers to the figure are as follows (see Section 4.1.2):

*IrriMap_CN estimates the irrigation ratios (i.e., the ratio of irrigated cropland area to total cropland area) for China, Northern China, and Xinjiang Uygur Autonomous Region as only 0.47, 0.37, and 0.61, respectively (Supplementary Figure S4). In comparison, the values derived from CIrrMap250 are 0.58, 0.70, and 0.96, respectively, which align more closely with reality and official reports (https://gtdc.mnr.gov.cn/).*

[Figure]

*Figure S4. Comparison of irrigated ratio estimates of CIrrMap250 and IrriMap_CN in China, Northern China, Xinjiang Uygur Autonomous Region*

L474-475: "... The accuracy of the irrigated cropland map would decrease by approximately 5%-6% (Supplementary Figure S6) " - Figure S6 (in the supplementary document) contradicts this statement. From the figure, it appears that "considering FC of cropland" (blue bars according to the plot legend, and "this study" according to the caption) yields worse overall accuracies (OA) than "Neglecting FC of cropland" (green bars). This is the case for all three (2000, 2010, 2020) years. Is the plot legend correct?

**Response:** Thanks for the kind reminder. We have mistakenly presented the legend of the figure. As depicted in the revised figure (see below), the accuracy of the irrigated cropland map would decrease by approximately 5%-6% if we disregard the fractional coverage of cropland. Yes, this decrease in mapping accuracy can be observed across the three years (i.e., 2000, 2010, and 2020).

[Figure]

*Figure S7. Comparison of performance of irrigated area maps in the scenarios of considering fractional coverage (FC) of irrigated cropland (this study) and neglecting FC of irrigated cropland*

**References**

Chen, F., Zhao, H., Roberts, D., Van de Voorde, T., Batelaan, O., Fan, T., Xu, W., 2023. Mapping center pivot irrigation systems in global arid regions using instance segmentation and analyzing their spatial relationship with freshwater resources. Remote Sensing of Environment, 297: 113760.

Debeurs, K., Townsend, P., 2008. Estimating the effect of gypsy moth defoliation using MODIS. Remote Sensing of Environment, 112(10): 3983-3990.

Dong, J., Xiao, X., Menarguez, M.A., Zhang, G., Qin, Y., Thau, D., Biradar, C., Moore, B., 2016. Mapping paddy rice planting area in northeastern Asia with Landsat 8 images, phenology-based algorithm and Google Earth Engine. Remote Sensing of Environment, 185: 142-154.

Gao, B.-c., 1996. NDWI—A normalized difference water index for remote sensing of vegetation liquid water from space. Remote Sensing of Environment, 58(3): 257-266.

Huete, A.R., Liu, H.Q., Batchily, K., van Leeuwen, W., 1997. A comparison of vegetation indices over a global set of TM images for EOS-MODIS. Remote Sensing of Environment, 59(3): 440-451.

Li, H., Chen, Y., 2020. Assessing potential land suitable for surface irrigation using groundwater data and multi-criteria evaluation in Xinjiang inland river basin.

Computers and Electronics in Agriculture, 168: 105079.

Liu, X., Zheng, J., Yu, L., Hao, P., Chen, B., Xin, Q., Fu, H., Gong, P., 2021. Annual dynamic dataset of global cropping intensity from 2001 to 2019. Scientific Data, 8(1): 283.

McFeeters, S.K., 1996. The use of the Normalized Difference Water Index (NDWI) in the delineation of open water features. International Journal of Remote Sensing, 17(7): 1425-1432.

Mehta, P., Siebert, S., Kummu, M., Deng, Q., Ali, T., Marston, L., Xie, W., Davis, K.F., 2022. Majority of 21stcentury global irrigation expansion has beenin water stressed regions (preprint). https://doi.org/10.31223/X5C932.

Meier, J., Zabel, F., Mauser, W., 2018. A global approach to estimate irrigated areas – a comparison between different data and statistics. Hydrology and Earth System Sciences, 22(2): 1119-1133.

Priestley, C.H.B., Taylor, R.J., 1972. On the Assessment of Surface Heat Flux and Evaporation Using Large-Scale Parameters. Monthly Weather Review, 100(2): 81-92.

Siddiqui, S., Cai, X., Chandrasekharan, K., 2016. Irrigated Area Map Asia and Africa. International Water Management Institute. https://waterdata.iwmi.org/applications/irri_area/.

Singha, M., Dong, J., Zhang, G., Xiao, X., 2019. High resolution paddy rice maps in cloud-prone Bangladesh and Northeast India using Sentinel-1 data. Scientific Data, 6(1): 26.

Thenkabail, P., Knox, J., Ozdogan, M., Gumma, M., Congalton, R., Wu, Z., Milesi, C., Finkral, A., Marshall, M., Mariotto, I., You, S., Giri, C., Nagler, P., 2016. NASA Making Earth System Data Records for Use in Research Environments (MEaSUREs) Global Food Security Support Analysis Data (GFSAD) Crop Dominance 2010 Global 1 km V001, distributed by NASA EOSDIS Land Processes Distributed Active Archive Center, https://doi.org/10.5067/MEaSUREs/GFSAD/GFSAD1KCD.001. Accessed 2023-10-17.

Tian, X., Dong, J., Chen, X., Zhou, J., Gao, M., Wei, L., Kang, X., Zhao, D., Zhang, H., Crow, W.T., Huang, R., Shao, W., Zhou, H., 2024. County-Level Evaluation of Large-Scale Gridded Data Sets of Irrigated Area Over China. Journal of Geophysical Research: Atmospheres, 129(5): e2023JD040333.

Worqlul, A.W., Collick, A.S., Rossiter, D.G., Langan, S., Steenhuis, T.S., 2015. Assessment of surface water irrigation potential in the Ethiopian highlands: The Lake Tana Basin. Catena, 129: 76-85.

Worqlul, A.W., Jeong, J., Dile, Y.T., Osorio, J., Schmitter, P., Gerik, T., Srinivasan, R., Clark, N., 2017. Assessing potential land suitable for surface irrigation using

groundwater in Ethiopia. Applied Geography, 85: 1-13.

Wu, B., Tian, F., Nabil, M., Bofana, J., Lu, Y., Elnashar, A., Beyene, A.N., Zhang, M., Zeng, H., Zhu, W., 2023. Mapping global maximum irrigation extent at 30m resolution using the irrigation performances under drought stress. Global Environmental Change, 79: 102652.

Xie, Y., Gibbs, H.K., Lark, T.J., 2021. Landsat-based Irrigation Dataset (LANID): 30-m resolution maps of irrigation distribution, frequency, and change for the U.S., 1997–2017. Earth Syst. Sci. Data, 2021: 1-32.

Xie, Y., Lark, T.J., 2021. Mapping annual irrigation from Landsat imagery and environmental variables across the conterminous United States. Remote Sensing of Environment, 260: 112445.

Xu, X., 2017. Remote sensing-derived crop intensity for China's cropland (in Chinese).

Zhang, C., Dong, J., Ge, Q., 2022a. IrriMap_CN: Annual irrigation maps across China in 2000–2019 based on satellite observations, environmental variables, and machine learning. Remote Sensing of Environment, 280: 113184.

Zhang, L., Ma, Q., Zhao, Y., Chen, H., Hu, Y., Ma, H., 2023. China's strictest water policy: Reversing water use trends and alleviating water stress. Journal of Environmental Management, 345: 118867.

Zhang, L., Wang, W., Ma, Q., Hu, Y., Zhao, Y., 2024. CCropLand30: High-resolution hybrid cropland maps of China created through the synergy of state-of-the-art remote sensing products and the latest national land survey. Computers and Electronics in Agriculture, 218: 108672.

Zhang, L., Zhang, K., Zhu, X., Chen, H., Wang, W., 2022b. Integrating remote sensing, irrigation suitability and statistical data for irrigated cropland mapping over mainland China. Journal of Hydrology, 613: 128413.

Zhu, X., Zhu, W., Zhang, J., Pan, Y., 2014. Mapping Irrigated Areas in China From Remote Sensing and Statistical Data. IEEE Journal of Selected Topics in Applied Earth Observations and Remote Sensing, 7(11): 4490-4504.

Zomer, R.J., Xu, J., Trabucco, A., 2022. Version 3 of the Global Aridity Index and Potential Evapotranspiration Database. Scientific Data, 9(1): 409.

---

## Author Response (AR1)

**Responses to the comments of Referee #1**

**Article ID:** essd-2024-2
**Title:** CIrrMap250: Annual maps of China's irrigated cropland from 2000 to 2020 developed through multisource data integration
**Authors:** Ling Zhang, Yanhua Xie, Xiufang Zhu, Qimin Ma, Luca Brocca

Dear Reviewer,

Thank you very much for the great efforts on our manuscript. Inspired by your valuable comments, we have made a major revision to our manuscript. The key revisions include:

(1) The data description has been carefully rewritten to avoid any potential misinterpretations by users.

(2) Additional experiments have been conducted to provide further explanation of our methodology.

(3) Additional information and discussion regarding our results have been incorporated.

(4) Many paragraphs, sentences, and figures have been revised to improve readability, conciseness, and clarity.

The detailed point-to-point responses are as follows. Texts in red are the reviewer's comments; **those in black** are our responses to the reviewer's comments; and *those in blue and italics* are the revised texts appeared in the revised manuscript.

The manuscript "CIrrMap250: Annual maps of China's irrigated cropland from 2000 to 2020 developed through multisource data integration" applies a random forest algorithm to classify and produce a new irrigation map product (CIrrMap250) over China at 250m resolution. The authors evaluate the new maps quantitatively and qualitatively (using reference data, withdrawal data and other existing irrigation products) over the 2000-2020 period. Generally, the paper is properly structured. It is well suited for this journal. However, the manuscript and supporting document appear rushed with several inconsistencies and mistakes. Some remarks:

• Check (and re-check) all the reported details. I.e., the performance metrics and other variables in the figures, tables and elsewhere in the manuscript (and the supplementary document). Please correct all inconsistencies. More below.

**Response:** Thanks for your detailed and valuable comments. We sincerely apologize for the mistakes we made in the original manuscript. We have thus carefully read through the revised manuscript and supplementary file, including figures, equations, tables, and text. please refer to our point-by-point response below for further details.

• What is the definition of 'irrigated cropland' as used in this study? At first I was rather intrigued when the authors mentioned in the initial sections that their product gives the irrigated cropland (which I interpreted as the fraction of vegetation cover that is actually irrigated). On further reading, however, it seemed the authors were only labeling the pixels as either irrigated [1] or not [0] and then presenting the total fraction vegetation cover (FVC) of [1] as the 'irrigated cropland' …is my understanding correct? If this is the case, what differentiates this product from a binary [1,0] irrigation map that is combined with the many (readily available) FVC products. Actually, one would argue that the latter method is better as it is not prone to misinterpretation by the user. Users are likely to misinterpret the produced CIrrMap250 irrigation maps to mean the ACTUAL irrigated pixel proportion and not the total vegetation cover. Also, how do you address pixels that have possibly been assigned an FVC of ~0 (e.g. at early growth stages) but have an [actual] irrigated area/extent larger than 0?

**Response:** Thanks for the valuable comments. Your understanding is correct. In this study, each 250-meter pixel was categorized as either irrigated or non-irrigated. No further classification was conducted to distinguish between irrigated and non-irrigated cropland at the subpixel level. Therefore, if a pixel was classified as "irrigated", it was assumed that all cropland within that pixel was irrigated.

The binary irrigation maps were spatially filtered and finally multiplied by the corresponding cropland mask layers to produce the annual maps of irrigated cropland

in China (i.e., CIrrMap250). As a result, the pixel value of our product indicates the percentage of a 250 m resolution pixel covered by irrigated croplands (i.e., pixel value = irrigated area / pixel area ×100). Unlike simple binary maps, this post-processing step was implemented to consider the fractional coverage of croplands within coarse-resolution pixels, thereby enhancing the accuracy of irrigated area estimates in China, where farms are typically small and fragmented. For instance, in a binary irrigation map, if 10 grids in a county are classified as "irrigated", the calculated irrigated area would be 250×250×10 = 625,000 m² without considering fractional coverage of cropland. However, if the cropland coverage within each grid in the county is only 50%, then the actual irrigated area should be halved, amounting to 312,500 m².

To mitigate any misinterpretations, we have explicitly clarified our product in the introduction and methodology sections. Additionally, we have removed phrases such as "irrigation cropland proportion", "fraction coverage of irrigated cropland", and "the mixed pixel issue" from our dataset descriptions.

*The newly developed maps (CIrrMap250) feature a spatial resolution of 250 meters at an annual frequency from 2000 to 2020. Our maps show the percentage of each 250 m by 250 m pixel that is covered by irrigated cropland (i.e., pixel value = irrigated area / pixel area ×100).*

*Finally, we multiplied the binary, spatially filtered irrigation maps by their corresponding cropland mask layers to generate annual irrigation maps for China. The final product, CIrrMap250, represents the percentage of a 250 m pixel covered by irrigated croplands (i.e., pixel value = irrigated area / pixel area ×100). Unlike simple binary maps, our product considers the fractional coverage of croplands within coarse-resolution MODIS pixels, thereby enhancing the accuracy of irrigation area estimates in China, where farms are typically small and fragmented.*

Consistent with prior researches (Zhu et al., 2014; Meier et al., 2018; Zhang et al., 2022a; Wu et al., 2023), irrigated cropland in our study is defined as cropland that is subject to irrigation. Consequently, a crucial step of mapping irrigated cropland involved selecting or generating suitable cropland mask layers. The classification of irrigated and non-irrigated cropland was exclusively conducted at the cropland grids (i.e., irrigated cropland was restricted to cropland areas). Thus, each irrigation map corresponds to a specific cropland mask. For example, CIrrMap250 utilized the cropland mask from the high-resolution (30-meter) hybrid cropland product (CCropLand30) (Zhang et al., 2024), while IrriMap_CN employed the cropland mask from the National Land Cover Dataset (NLCD) (Zhang et al., 2022a). Consequently, binary irrigation maps cannot be merged with other cropland masks due to significant disparities in cropland identification by different cropland datasets. For instance, a pixel classified as irrigated cropland in the irrigation map based on cropland mask A may

become non-irrigated if merged with another cropland mask B, as it may be classified as non-cropland in cropland mask B.

The Fraction of Vegetation Cover (FVC) typically represents the percentage of ground covered by green vegetation, ranging from 0% to 100%. However, in our study, fraction coverage of cropland denotes the proportion of cropland area within each 250 m by 250 m pixel. For example, a pixel value of 0.2 in the cropland mask layer indicates that 20% of the 250-meter grid is covered by cropland. The classification of irrigated and non-irrigated cropland was exclusively performed on the cropland grids identified by cropland masks. Cropland proportion in each pixel was assumed to remain unchanged throughout the year in our study and other similar studies (Zhu et al., 2014; Meier et al., 2018; Zhang et al., 2022a; Wu et al., 2023).

- The CIrrMap250 product is limited to China. Have the authors considered applying a similar methodology to other regions, e.g. extend it globally? Obviously, training and test datasets from other global sites would be required, but would it be viable to apply your RF classifier/model (as-is) to other regions beyond China? What would be the limitations?

**Response:** In this study, we developed CIrrMap250 by integrating multisource data through a semi-automatic training approach (Zhang et al., 2022d; Xie et al., 2019). While our irrigation mapping method is applicable to other regions worldwide, we acknowledge that its effectiveness largely depends on the availability and reliability of multisource datasets, particularly those related to irrigation statistics and surveys. This dependency stems from our methodology's framework, which uses a threshold-calibration method to generate training samples for each county in China based on remote sensing data, reported statistics/surveys, and an irrigation suitability maps Consequently, the random forest models trained in this study were customized for China and may not be directly transferable to other regions due to significant variations in irrigation practices, landscapes, and climatic characteristics (Salmon et al., 2015; Zhang et al., 2022d).

Specific comments :
L16: "… and considered the fraction coverage of irrigated cropland (i.e., the mixed pixel issue). In this study, we addressed these important gaps …" - This is somewhat misleading as the mixed pixel issue is not addressed in this manuscript. I was expecting that the authors were referring to 'mixed pixel' in terms of irrigation, i.e. proportion of the fraction vegetation cover (FVC) that is irrigated or not. If not mistaken, the only

consideration here is the total FVC, which is provided within most RS products anyway, and can thus be similarly combined (rather straightforwardly) with any available binary/boolean [1,0] irrigation maps. Also see your comment in L495 : "CIrrMap250 cannot differentiate irrigated and rain-fed croplands at the subpixel scales. There are many small and fragmented croplands in … with complex terrain and diverse vegetation types. CIrrMap250 should be used with caution in these regions due to the wide existence of the mixed pixels"

**Response:** Yes, we agree that this is confusing as the issue of mixed pixels has not been explicitly addressed in our work. We have revised the sentence to mitigate any confusion.

*Accurate maps of irrigation extent and dynamics are crucial for studying food security and its far-reaching impacts on Earth systems and the environment. While several efforts have been made to map irrigated area in China, few have provided multiyear maps, incorporated national land surveys, addressed data discrepancies, and considered the fractional coverage of cropland within coarse-resolution pixels.*

Actually, here, we intend to highlight a gap in previous studies, wherein binary cropland masks were utilized for irrigation mapping. In such masks, each pixel is classified eighter as cropland or non-cropland, disregarding the fractional coverage of cropland within the coarse-resolution pixels. This may lead to overestimations or underestimations of the extent of irrigated cropland due to the following two reasons. First, many studies generated the cropland mask layers by resampling the original 30-meter cropland data to coarse resolution (e.g., 1 km or 500 m). This resampling process could overlook cropland that covers a relatively small proportion of the coarse-resolution grid, while overestimating cropland in grids that are not totally covered by cropland. Secondly, the threshold-splitting method was commonly used in conjunction with irrigated area statistics to depict the spatial distribution of irrigated cropland; and this method relies on the assumption that the spatially allocated irrigated area should be equal to the statistics. If it is assumed that each grid cell is fully covered by cropland, the extent of irrigated cropland may be significantly underestimated. For instance, if the statistical irrigated area of a county is 625,000 $m^2$, and 10 grids (pixel area:250×250 = 62,500 $m^2$) would be classified as irrigated cropland in a binary cropland mask. However, if the cropland proportion within each grid in the county is only 50%, then in reality, 20 grids should be classified as irrigated cropland.

L17: "… named as CIrrMap250 …" – consider describing all abbreviations such as CIrr before use.

**Response:** The abbreviation "CIrrMap250" has been explained.

*Here, we addressed these important gaps and developed new annual maps of China's irrigated cropland from 2000 to 2020, named as CIrrMap250 (China's irrigation map with a 250 m resolution).*

L23: "… accuracy of 0.79-0.88 for years 2000, 2010, and 2020, respectively" - only for years 2000, 2010, 2020? What about the other years in between? Is it because the evaluation data were only available for those 3 years? If so, make it a bit clear here.

**Response:** Yes, the evaluation was conducted only for years 2000, 2010, and 2020, because the reference data were only available for the 3 years.

*Our CIrrMap250 maps demonstrated an overall accuracy of 0.79-0.88 for the years 2000, 2010, and 2020, and outperformed currently available maps..*

L42: its' >> its

**Response:** Sorry for our carelessness. It has been revised.

*Given the vital importance of irrigation, knowing its precise location and dynamics is essential.*

L45-46: "While numerous land use/cover and thematic cropland products have been made available to the public, they often lack information on irrigation status …" - Why would it be important to provide land use land cover (LULC) maps with irrigation status information? Should rain/precipitation or evapotranspiration information be provided within LULC maps/products as well?

**Response:** We agree with you. This sentence has been removed in the revised manuscript.

L51: "…normalized difference water index (NDWI)…" - Note that there is another index that goes by the same name but used to detect floods/open water bodies (NDWI, McFeeters (1996)) – so it could ideally be used to map areas that employ flood irrigation

(rice paddies, for example).

**Response:** Thank you for this reminder. The Normalized Difference Water Index (NDWI) proposed by Gao (1996) is known for its sensitivity to both soil and plant water content, making it a valuable tool for monitoring rice paddy fields (Dong et al., 2016; Singha et al., 2019). Consequently, it was utilized in this study as well as many other studies (Deines et al., 2017; Deines et al., 2019; Xiang et al., 2020; Zhang et al., 2022), for mapping irrigated areas. Regarding the NDWI proposed by McFeeters (1996), we acknowledge its potential utility for mapping areas employing flood irrigation, such as rice paddies. We have incorporated this reference into the revised manuscript.

L57: "…been applied to detected irrigate areas…" >> …to detect irrigated areas

**Response:** Sorry for our carelessness. It has been revised.

*The soil moisture-based approach utilizes remotely sensed soil moisture signals from microwave and optical sensors to detect irrigated areas by using similar techniques like threshold splitting (Yao et al., 2022) and supervised/unsupervised classification (Gao et al., 2018; Dari et al., 2021).*

L74: "China is a big agricultural country with the \*largest irrigated area in the world …" – any reference for this?

**Response:** Yes, we have added the reference.

*China is a big agricultural country with the largest irrigated area in the world (International Commission on Irrigation and Drainage, 2018)*

*International Commission on Irrigation and Drainage: World Irrigated Area-2018, https://www.icid.org/world-irrigated-area.pdf, 1-6, 2018.*

L85: "… in paces …" – do you mean places?

**Response:** Yes, it's a type error. It has been revised.

*As a result, it remains unclear where the changes in irrigation area are water-sustainable (e.g., irrigation expansion in places without water stress) (Mehta et al., 2024).*

L97: "many other studies " – which studies? Add some reference[s] here

**Response:** We have added the related references.

*Finally, it is worth noting that, apart from Zhang et al. (2022a), many studies assessed their irrigation maps with a limited number of reference samples, potentially compromising the reliability of their evaluation results (Zhu et al., 2014; Xiang et al., 2020; Bai et al., 2022; Zhang et al., 2022d).*

L104-105: "CIrrMap250) have a spatial resolution of 250 meters and describe irrigated cropland distribution through fractional coverage" – what of the temporal resolution? Also, as already mentioned above, this statement is misleading as one could assume you are providing the fraction of total FVC that is under irrigation.

**Response:** The newly developed irrigated cropland maps (CIrrMap250) have an annual temporal resolution. The phrase "describe irrigated cropland distribution through fractional coverage" has been removed to prevent potential confusion. We have rewritten the data descriptions in the introduction and methodology sections. Please refer to our response to your first comment.

*The newly developed maps (CIrrMap250) feature a spatial resolution of 250 meters at an annual frequency from 2000 to 2020. Our maps show the percentage of each 250 m by 250 m pixel that is covered by irrigated cropland (i.e., pixel value = irrigated area / pixel area ×100).*

*Finally, we multiplied the binary, spatially filtered irrigation maps by their corresponding cropland mask layers to generate annual irrigation maps for China. The final product, CIrrMap250, represents the percentage of a 250 m pixel covered by irrigated croplands (i.e., pixel value = irrigated area / pixel area ×100). Unlike simple binary maps, our product considers the fractional coverage of croplands within coarse-resolution MODIS pixels, thereby enhancing the accuracy of irrigation area estimates in China, where farms are typically small and fragmented.*

L113-116: "These indices were generated every 16 days with a spatial resolution of 250 meters…" – to be consistent with other descriptions in the section, provide the product number of the vegetation indices product; is it MOD13Q1? " …band 4 …band1" – consider adding the spectral ranges here as well

**Response:** Yes, the product number is MOD13Q1. We have added the product number as well as the spectral ranges for the bands.

*We collected the Terra Moderate Resolution Imaging Spectroradiometer (MODIS) MOD13Q1 vegetation indices, i.e., NDVI and Enhanced Vegetation Index (EVI) (Huete et al., 1997), from the NASA's Earth Science Data Systems (https://www.earthdata.nasa.gov/). These indices are generated every 16 days with a 250 m spatial resolution. Meanwhile, the MODIS band 4 (545-565 nm) surface reflectance from the MOD09A1 product was used and resampled from the original 500 m to 250 m using the nearest neighbor interpolation method (Debeurs and Townsend, 2008). The resampled data were then used together with the 250-m and 8-day band 1 (620-670 nm) surface reflectance from MOD09Q1 to derive the Greenness Index (GI) (Supplementary Table S1).*

L119: "Greenness Index (GI) (Supplementary Table S1)" – in Table.S1 (supplementary document) under GI, you write the 'formula' as GI=NIR/green, and 'MODIS bands' as 'Bands 01, 04'. The sub-caption however reads: 'Red: band 01' and 'Green: band 04' …which is which? Please correct.

**Response:** Apologies for our carelessness. In calculating the GI index, we utilized 'Bands 02, 04' instead of 'Bands 01, 04'. We have revised the table accordingly, as listed below for your reference.

*Table S1. Summary of the MODIS-derived vegetation indices used in this study*

| *Vegetation indices* | *Formula* | *MODIS bands* | *Resolution* |
|---|---|---|---|
| *NDVI* | *(NIR - Red) / (NIR + Red)* | *Band 01 (Red)*
 *Band 02 (NIR)* | *250 m/16 day* |
| *EVI* | *2.5\*(NIR-Red) / (NIR+ 6\*Red–7.5\*Blue+1)* | *Band 01 (Red)*
 *Band 02 (NIR)*
 *Band 03 (Blue)* | *250 m/16 day* |
| *GI* | *NIR/Green* | *Band 02 (NIR)*
 *Band 04 (Green)* | *250 m/8 day* |

*where NIR is the near-infrared band (841-876 nm), and Red (620 – 670 nm), Blue (459-*

*479 nm) and Green (545-565 nm) are the are the visible red band, visible blue band, and visible green band, respectively.*

L121: "… The data for unreliable pixels were reconstructed using a straightforward nearest neighbor interpolation method…" - is this the right way to go about it? For example, for an overcast pixel (which is maybe vegetated), why would you take the remotely sensed spectral signal of the next/closest cloud-free pixel (which is maybe urban/built-up)? Meaning you may end up missing vegetated pixels under irrigation or vice versa. Why not just drop such pixels from your analysis (i.e. at that particular time)?

**Response:** We completely understand your concern. Indeed, directly applying the interpolation method to all MODIS data in China could significantly impact the results. As you pointed out, if the neighboring pixel with reliable data is located in an urban or built-up area, the reconstructed pixel is likely to be erroneously excluded from irrigated cropland due to the low value of vegetation index. However, in this study, we actually extracted MODIS data only for cropland pixels in China. For cropland pixels with unreliable data, their values were interpolated from the nearest neighboring cropland pixels with reliable data. This approach helps to avoid interpolating data for cropland pixels from areas covered by other land use types, such as urban and forest. We have provided a more detailed explanation in the revised manuscript.

*We extracted MODIS data for all cropland pixels in China, using only high-quality data on cloud- and snow/ice-free pixels (Hilker et al., 2012). Low-quality MODIS data were excluded based on the quality band and were interpolated using high-quality data from the nearest neighboring cropland pixels.*

We chose not to exclude pixels with unreliable data at a particular time from our analysis because our mapping process relies heavily on the peak values of MODIS vegetation indices during the growth period. Omitting pixels with unreliable data for a specific time could potentially result in unreliable peak values of vegetation indices, thereby affecting our mapping results.

L157-159: "In years lacking survey data, the harmonized irrigated area was determined using Eq. 2, assuming that the relative changes in statistical irrigated area are reliable" - could you explain the rationale behind Equation (2)? How to interpret it? to me it appears that a year without survey data could end up having a lower assigned/harmonized irrigated area despite having a larger irrigated [statistical] area without land survey (Astatt2). For instance, if we assume: Aharmts=20, Astatts=20,

Astatt2=30, CAsurvts=40 ; then Aharmt2 becomes min(20*(30-20)/20,40)=10?
···the harmonized value (Aharmt2) even becomes negative if we consider Astatt2 to be less than Astatts. What am I missing? Please clarify.

**Response:** We apologize for the typographical error in Equation 2, where the relative changes of the statistical irrigated area should plus one before being multiplied with $A_{harm}^{ts}$. The correct equation should be:

$$A_{harm}^{t2} = min \left( A_{harm}^{ts} \times \left( 1 + \frac{A_{stat}^{t2} - A_{stat}^{ts}}{A_{stat}^{ts}} \right), CA_{surv}^{ts} \right) \tag{2}$$

For your example, Aharmts=20, Astatts=20, Astatt2=30, CAsurvts=40; then Aharmt2 becomes min (20*(1+(30-20)/20),40) =30. In this case, the relative change of the statical irrigated area is (30-20)/20*100=50%. Consequently, the harmonized data in the survey year should be adjusted by increasing 50%, i.e., 20*(1+0.5) =30. This ensures that the relative changes between Aharmt and Aharmt2, and between Astatts and Astatts2 remain consistent. This process preserves the interannual changes observed in the statistical irrigated area while enhancing data consistency across years. For instance, in a span of five years lacking survey data, the recorded statistical irrigated areas are 11, 12, 13, 14, and 15 hectares respectively, whereas the reconciled irrigated areas in adjacent years with survey data might amount to 101, 102, 103, 104, and 105 hectares. Without the aforementioned adjustment, notable data inconsistencies would arise. In the revised manuscript, we corrected Equation 2, and meanwhile, double-checked all other equations to ensure their correct formulation.

*For years without survey data, the irrigation area was estimated by adjusting the harmonized data from adjacent survey years using relative change information derived from the irrigation statistics (Eq. 2). This method preserved the interannual changes observed in statistical irrigation area while enhancing data consistency across years.*

$$A_{harm}^{t2} = min \left( A_{harm}^{ts} \times \left( 1 + \frac{A_{stat}^{t2} - A_{stat}^{ts}}{A_{stat}^{ts}} \right), CA_{surv}^{ts} \right) \tag{2}$$

*where $A_{harm}$, $A_{stas}$ and $A_{surv}$ represent the harmonized, statistical and surveyed irrigation area, respectively; $CA$ is surveyed area of cropland; and $ts$ and $t2$ indicate the year with and without land surveys, respectively.*

L189: "… used in combination with the MCD43A3 albedo product" - this is a daily product. Did the authors calculate the daily PET? how did you reconcile this with the other 8/16-day products?

**Response:** Yes, the MCD43A3 albedo is a daily product, so we computed the daily PET accordingly. These daily PET values were summed to annual values spanning from 2000 to 2020. In the revised manuscript, we have clarified this point.

*These datasets were combined with the MCD43A3 albedo product to compute daily potential evapotranspiration (PET) using the Priestley-Taylor method (Priestley and Taylor, 1972). The daily PET values were aggregated to annual values for the period from 2000 to 2020, which were then used to derive the aridity index, defined as the ratio of precipitation to PET.*

In terms of the other 8/16-day products, such as NDVI, EVI, and GI, we utilized their annual peak values in this study, rather than directly employing their original values. Consequently, both the estimated PET and other MODIS products were utilized at annual scales.

L211: "… were then then" >> were then

**Response:** Revised.

*The resulting county-level maps were then mosaicked and post-processed to produce the annual maps of irrigated cropland in China, referred to as CIrrMap250.*

L222: "*A static irrigation suitability map *were constructed based on …, and aridity index of cropland" - was this one map or several ('*A static' then '*were'). If one, why was the temporal variation of the aridity index not considered?

**Response:** In this study, we utilized a single and static irrigation suitability map. The concept of integrating irrigation suitability into the mapping process was inspired by previous researches assessing land potential for irrigation (Worqlul et al., 2015; Worqlul et al., 2017; Li and Chen, 2020; Zhang et al., 2022b). The aridity index served as a metric reflecting climate suitability for irrigation, where lower values indicate a higher deficit of available water for crops, thereby suggesting a greater need for irrigation. However, aridity is typically measured by comparing long-term average water supply (precipitation) to long-term average water demand (evapotranspiration) (Zomer et al., 2022), rather than through the lens of a single year's data. Consequently, in our study, we utilized a 21-year average aridity index (covering the period 2000-2020) in conjunction with elevation and slope data to produce a static irrigation map for China.

Regarding your concerns, we conducted an additional experiment to evaluate the influence of temporal variations in the aridity index on irrigation mapping results.

Specifically, we computed the aridity index for each year and applied it to derive annual irrigation suitability maps spanning from 2000 to 2020. The resulting 21 suitability maps were then applied to each corresponding year to generate irrigated cropland maps using the methodology outlined in our study. We kept other factors the same to ensure that any disparities in the mapping results from our original ones are solely attributed to the use of annual irrigation suitability maps. As shown in Figure R1, the incorporation of annual irrigation suitability maps has a negligible impact on the accuracy of irrigation maps.

[Figure]

Figure R1. Accuracy of irrigation maps derived from the experiments with static and varying irrigation suitability maps, respectively. The static irrigation map was derived from elevation, slope, and the 21-year (i.e., 2000-2020) averaged aridity index, while the varying irrigation suitability maps from elevation, slope, and annual aridity index.

L230: "… (Supplementary Table S2)" – in Table S2 (supplementary document), why do you have the same 'Suitability value' for the lowest suitability classes S3 and S4. I.e., for the 'elevation' and 'slope' irrigation suitability factors, S3=2 and S4=2.

**Response:** It appears to be an unintentional error in the original table. For the irrigation suitability factors 'elevation' and 'slope', the correct values should be S3=2 and S4=1. The suitability values for each factor should follow a monotonic pattern, with higher values indicating greater suitability for irrigation. We have corrected this unintentional error and have also double-checked all the other tables in our manuscript to avoid similar typo errors.

*Table S3. Suitability values for the influencing factors of irrigation suitability*

| *Influencing factors* | *Reclassification* | *Suitability value* |
| --- | --- | --- |

| | | |
|---|---|---|
| *elevation* | *S1: < min+100* | *S1=4* |
| | *S2: [min+100, min+300]* | *S2=3* |
| | *S3: [min+300, min+500]* | *S3=2* |
| | *S4: > min+500* | *S4=1* |
| *slope* | *S1: <2%* | *S1=4* |
| | *S2: [2%, 4%]* | *S2=3* |
| | *S3: [4%, 8%]* | *S3=2* |
| | *S4: > 8%* | *S4=1* |
| *aridity index* | *S1: <0.1* | *S1=10* |
| | *S2: [0.1, 0.2]* | *S2=9* |
| | *S3: [0.2, 0.3]* | *S3=8* |
| | *S4: [0.3, 0.4]* | *S4=7* |
| | *S5: [0.4, 0.5]* | *S5=6* |
| | *S6: [0.5, 0.6]* | *S6=5* |
| | *S7: [0.6, 0.7]* | *S7=4* |
| | *S8: [0.7, 0.8]* | *S8=3* |
| | *S9: [0.8, 0.9]* | *S9=2* |
| | *S10: >0.9* | *S10=1* |

*Note: min is minimum elevation of the mapping unit*

L258: "and time-invariant environmental variables (i.e., latitude, longitude, crop intensity" – why is the crop intensity considered time-invariant?

**Response:** In this study, we utilized a stable (time-invariant) cropping intensity dataset (Xu, 2017) due to the lack of publicly available annual cropping intensity dataset when our work was conducted. While an annual dynamic global cropping intensity dataset is available for the years covering 2001-2019 (Liu et al., 2021), employing this dynamic dataset did not yield improvements in mapping accuracy; in fact, it resulted in a slightly decreased accuracy, potentially attributable to its relatively lower precision in China. The test results, as depicted in Figure R2, are provided for your reference. Consequently, we opted to utilize the available time-invariant crop intensity data for this study.

[Figure]

Figure R2. Comparison of irrigation mapping accuracy for the year 2000, 2010, and 2020 in the experiments of using stable and dynamic cropping intensity datasets, respectively.

L261: "To enhance the accuracy of these maps, a spatial filter (a 7x7 window)…" - clarify what you mean by this. Why 7x7? …'constituting <5% of the window area' is ambiguous. Is the 250 m resolution retained after this?

**Response:** After classification, we employed a spatial filtering to remove isolated irrigation pixels and identify potentially omitted irrigated croplands. Specifically, we first calculated the irrigation proportion within a 7×7-pixel window for each preliminary irrigation pixel. Then, all cropland pixels within the moving window were assigned as "non-irrigated" if the calculated ratio fell below 5%. Conversely, if the ratio exceeded 95%, we assumed all cropland pixels within the moving window to be irrigated. The spatial filtering operation preserved the original spatial resolution of the maps (250 m). The size of the moving window was determined to be 7×7 pixels through a trial-and-error process. We tested three different window sizes (5×5, 7×7, and 9×9) in the post-processing step and found that the 7×7 window size yielded the highest mapping accuracy.

In the revised manuscript, we have clarified the spatial filtering process.

*We then employed a spatial filtering to remove isolated irrigation pixels and identify potentially omitted irrigated croplands. Specifically, we first calculated the irrigation proportion within a 7×7-pixel window for each preliminary irrigation pixel.*

*Then, all cropland pixels within the moving window were assigned as "non-irrigated" if the calculated ratio fell below 5%. Conversely, if the ratio exceeded 95%, we assumed all cropland pixels within the moving window to be irrigated. The spatial filtering operation preserved the original spatial resolution of the maps (250 m).*

L276: "were acquire from …" >> were acquired

**Response:** Revised.

*The second validation dataset, for the year 2020 (Figure 2c), was acquired from Chen et al. (2023) that showed the global location of center pivot irrigation systems (CPIS).*

L282-: "Due to the lack of georeferencing information, we georeferenced these land use maps using the georeferencing tool in ArcGIS in conjunction with high-resolution images " – the authors do not talk about the data that were used to serve as ground control points for the georeferencing (e.g. How many GCPs, their spatial distribution, …?)

**Response:** In total, we selected 234 control points nationwide, primarily distributed along provincial boundaries. In the revised manuscript, we added the information on the georeferencing points, and provided the spatial distribution map of these points (see below) in the supplementary file.

*We georeferenced these land use maps using the georeferencing tool in ArcGIS. A total of 234 control points were selected from high-resolution images and provincial administrative boundaries for the georeferencing process (Supplementary Figure S1). The irrigation samples were randomly extracted from irrigated lands and paddy fields, while non-irrigated samples were taken from dryland patches.*

[Figure]

*Figure S1. Spatial distribution of the identified reference points used for georeferencing the land-use maps of China's second National Land Survey*

L294-: "It's noteworthy that this percentage represents the proportion of cropland within the 250 …, not the proportion of irrigated cropland to total cropland" ; L362: "irrigated cropland in CIrrMap250". As already mentioned, giving the irrigated cropland as a percentage is very likely to mislead users into assuming that your irrigation product provides the proportion of the total fraction of vegetation cover (FVC/cropland) that is irrigated. If feasible, wouldn't it be more useful to have both products, i.e. the total fraction cover product and the proportion of that that is deemed irrigated? The authors also acknowledge in L492 that "…cirrmap250 has a relatively coarse resolution". You may still argue that at the relatively higher spatial resolution of 250m, one could assume the whole cropland (total FVC) to be equivalent to the irrigated area. This might be true but still needs validation to avoid being misleading.

**Response:** To prevent misinterpretation, we have clarified our product in in the introduction and methodology sections.

*The newly developed maps (CIrrMap250) feature a spatial resolution of 250 meters at an annual frequency from 2000 to 2020. Our maps show the percentage of each 250 m by 250 m pixel that is covered by irrigated cropland (i.e., pixel value = irrigated area*

*/ pixel area ×100).*

*Finally, we multiplied the binary, spatially filtered irrigation maps by their corresponding cropland mask layers to generate annual irrigation maps for China. The final product, CIrrMap250, represents the percentage of a 250 m pixel covered by irrigated croplands (i.e., pixel value = irrigated area / pixel area ×100). Unlike simple binary maps, our product considers the fractional coverage of croplands within coarse-resolution MODIS pixels, thereby enhancing the accuracy of irrigation area estimates in China, where farms are typically small and fragmented.*

As you pointed out, one potential approach to avoid the misinterpretation is to provide both binary irrigated cropland maps and cropland mask layers (representing fractional coverage of cropland) to users. While we acknowledge the merits of this method, we opted not to implement it for the following reasons. Firstly, data users may misinterpret that pixels with value equals to 1 are fully irrigated, and may directly utilize the binary irrigation maps for their research, such as estimating irrigation water use or assessing the hydroclimatic impact of irrigation. However, relying solely on these maps could bring significant biases into their results. Secondly, users would need to combine the binary irrigation maps with the cropland mask layers, a process that may introduce errors and increase the risk of generating irrigation maps divergent from those we have released. Lastly, providing only irrigation maps, instead of both binary irrigation maps and cropland mask layers, aligns with the practices of other similar studies (Zhu et al., 2014; Meier et al., 2018; Xie and Lark, 2021; Zhang et al., 2022a; Wu et al., 2023).

L340: "… under severe to extreme…" - In the previous sentence (L339), only low, moderate, high and severe WSI ranges are described. What is the extreme WSI range? Is extreme synonymous to severe here?

**Response:** The levels of water stress should be categorized as: low (WSI ≤ 0.2), moderate (0.2 < WSI ≤ 0.4), severe (0.4 < WSI ≤ 1.0), and extreme (WSI > 1), in line with our previous study (Zhang et al., 2023). It has been revised in the new manuscript.

*WSI denotes the fraction of available water resources appropriated by humans and is employed to categorize water stress into four levels: low (WSI ≤ 0.2), moderate (0.2 < WSI ≤ 0.4), severe (0.4 < WSI ≤ 1.0), and extreme (WSI > 1) (Zhang et al., 2023b). Irrigation expansion under severe to extreme water stress was designated as "unsustainable" due to the potential of exacerbating depletion of surface water and groundwater (Mehta et al., 2024). Conversely, expansion of irrigation under low to moderate water stress or shrinkage of irrigation under severe to extreme stress was deemed "sustainable".*

L352: "CIrrMap250 and IrriMap_CN performs similarly in user's accuracy…" – TableS5 (supplementary document) shows a user accuracy (UA) of 1 (error of commission=0). Can this perfect UA be explained? From Fig3c/TableS2 (year 2020), IrriMap_CN has a producer accuracy (PA) of 0.2, why this huge discrepancy between the [perfect] irrigated.UA (1) and the [rather poor] irrigated.PA (0.2)?

**Response:** In 2010, the reference points were extracted from the Center Pivot Irrigation Systems (CPIS) map developed by Chen et al. (2023). All of these reference points represent irrigated samples, as shown in the newly added confusion matrix (see below). Consequently, both CIrrMap250 and IrriMap_CN achieved a perfect user's accuracy for the irrigation class. However, IrriMap_CN exhibited a low producer's accuracy of 0.2, as only 20% of the irrigated samples were correctly identified. We have clarified it in the revised manuscript.

*For the year 2020, CIrrMap250 detects 88% of center pivot irrigated fields, while IrriMap_CN identifies only 20% (Figure 3c and Supplementary Figure S2). Note that both CIrrMap250 and IrriMap_CN achieves a perfect user's accuracy for the irrigation class in 2020 because all the reference points are irrigated samples (Section 3.31 and Supplementary Table S7).*

Since you use ~20,000 samples in your classification exercise (into irrigated and non-irrigated), could you provide (in supplementary doc) the CIrrMap/IrriMap confusion matrices for 2000, 2010, 2020 to aid with interpretation (i.e. how many of the reference samples are irrigated or not? How do you split these into training and test sets? …more details on how the RF classifier used in CIrrMap250 performs, …)

**Response:** Thanks for your suggestion. As per your advice, we have incorporated the Confusion Matrix for CIrrMap250 and existing maps (IrriMap_CN, IAAA, GFSAD) in 2000, 2010, and 2020, respectively. The Confusion Matrix presents the numbers of correctly and erroneously classified irrigated and non-irrigated samples by different products, thereby facilitating a more comprehensive understanding of our results. It's important to note that, in this study, the training samples were generated using a threshold-calibration method (refer to Section 3.1), rather than obtained from the reference points. All reference samples were independent of the training data and were utilized for evaluating irrigation maps.

*Table S6. Confusion matrix for CIrrMap250 and existing maps (IrriMap_CN, IAAA, GFSAD) in 2000, 2010, and 2020, respectively*

| | Products | Classified | Reference | |
|---|---|---|---|---|
| | | | Irrigated | Non-irrigated |
| 2000 | CIrrMap250 | Irrigated | 271 | 75 |
| | | Non-irrigated | 66 | 246 |
| | IrriMap_CN | Irrigated | 172 | 43 |
| | | Non-irrigated | 165 | 278 |
| | IAAA | Irrigated | 221 | 177 |
| | | Non-irrigated | 116 | 144 |

| | Products | Classified | Reference | |
|---|---|---|---|---|
| | | | Irrigated | Non-irrigated |
| 2010 | CIrrMap250 | Irrigated | 6818 | 1385 |
| | | Non-irrigated | 1365 | 3325 |
| | IrriMap_CN | Irrigated | 5003 | 1167 |
| | | Non-irrigated | 3180 | 3543 |
| | IAAA | Irrigated | 5274 | 2183 |
| | | Non-irrigated | 2909 | 2527 |
| | GFSAD | Irrigated | 4939 | 1995 |
| | | Non-irrigated | 3244 | 2715 |

| | Products | Classified | Reference | |
|---|---|---|---|---|
| | | | Irrigated | Non-irrigated |
| 2020 | CIrrMap250 | Irrigated | 6340 | 0 |
| | | Non-irrigated | 849 | 0 |
| | IrriMap_CN | Irrigated | 1426 | 0 |
| | | Non-irrigated | 5763 | 0 |

L366-367: "CIrrMap250 yields irrigation ratios (i.e., the ratio of irrigated area to the total cropland area) of…" – this sentence contradicts L294 (i.e., "… this percentage represents the proportion of cropland within the 250 …, not the proportion of irrigated cropland to total cropland"), and many other statements in this report (e.g. L495 "cirrmap250 cannot differentiate irrigated and rainfed croplads at the subpixel scales"). Such inconsistencies make it somewhat difficult to follow and interpret your results/analyses.

**Response:** In this stud, we classified each 250-meter grid cell as either irrigated or non-irrigated. The binary irrigation maps were finally multiplied by the corresponding cropland mask layers to produce the annual maps of irrigated cropland in China (i.e., CIrrMap250). As a result, the pixel value of our product indicates the percentage of each 250-meter resolution pixel covered by irrigated croplands (i.e., pixel value =

irrigated area / pixel area ×100).  While our product does not provide the proportion of irrigated cropland area to total cropland area at the pixel scale, it can be utilized to determine irrigation ratio at the regional scale. Specifically, for a target region, we first calculate the irrigated cropland area and the total cropland area, respectively. Then, the area of irrigated cropland is divided by the total area of cropland to estimate the irrigation ratio for this region. We have revised the related sentence in the new manuscript.

*IrriMap_CN estimates irrigation proportion (i.e., the ratio of irrigated cropland area to total cropland area) to be 0.47, 0.37, and 0.61 for China, Northern China, and Xinjiang Uygur Autonomous Region, respectively (Supplementary Figure S4). In comparison, the values derived from CIrrMap250 are 0.58, 0.70, and 0.96, respectively, which align more closely with the official reports (https://gtdc.mnr.gov.cn/).*

L370-374: "However, CirrMap250 tends … southern part of South China (SC)" – why? Could you discuss this section a little bit more. Readers may not go back to the literature on the other products to find out by themselves. Also, what does '*southern part of *South China' mean?

**Response:** Thanks for the suggestion. We have provided more explanation and discussion on those results. South China is a subregion of China. We have revised the phrase "south part of South China" to simply "South China" to prevent any potential confusion.

*Nevertheless, CIrrMap250 tends to yield lower estimates of irrigation area in Northeast China (NEC) when compared to IrriMap_CN, possibly due to inaccurate statistical and survey data in this region. In contrast to CIrrMap250 and IrriMap_CN, IAAA notably underestimates irrigated croplands in Northwest China (NWC) and North China (NC), but overestimates in NEC and Southwest China (SWC). This could be explained by the fact that IAAA was developed using unsupervised classification (Siddiqui et al., 2016), limiting its ability to characterize the spatial heterogeneity of irrigation in China (Tian et al., 2024). GFSAD shows overestimations of irrigated area in the Dujiangyan district and the North China Plain but exhibits evident omission errors in sparsely distributed irrigation regions like NWC and South China (SC). The large bias of GFSAD is understandable, as it is not an irrigation-specific product and only covers five irrigated crops (Thenkabail et al., 2016; Xie et al., 2021).*

L388: Fig5 – This figure needs improvement. How come no irrigated pixels in zone B

**Response:** In the revised manuscript, we have improved the figure by thickening the borders of the subplots. Meanwhile, we have provided the legend for different products (as shown below). Regarding your comment on lacking irrigated pixels of GFSAD in Zone B, we have carefully reviewed the map and indeed found that it did not identify any irrigated cropland. This is because GFSAD notably underestimates irrigated cropland in Zone B and its surrounding regions. Furthermore, we cross-checked the comparison results of the irrigated cropland map with GFSAD in the study by Zhang et al. (2022a), they also reported significant underestimation of irrigated cropland by GFSAD, particularly in Southern China. The underestimation of irrigated cropland by GFSAD may be attributed to the fact that it is not an irrigation-specific product and only considers five irrigated crops (Thenkabail et al., 2016; Xie et al., 2021).

[Figure]

*Figure 5. Visual comparison of CIrrMap250 with existing maps. The five rows from top*

*to bottom correspond to the Google map, CIrrMap250, IrriMap_CN, IAAA and GFSAD, respectively. Locations of the four selected zones are presented in Figure 4a.*

L391: "Figure 6 …CIrrMap250 exhibits a robust agreement with OPTRAM3" - This is not clear from the figure. Qualitatively, Figure 6a may even be interpreted differently unless the authors have overlain CIrrMap250 over OPTRAM30. If that is the case, please find a better way to illustrate/present the map inter-comparisons.

**Response:** Thanks for the comment. In the previous version, Figure 6a presented overlays of CIrrMap250 on the OPTRAM30 map. We agree it is not easy to interpret. We have thus revised the figure. Specifically, we have depicted the irrigated cropland distribution in CIrrMap250, IAAA, IrriMap_CN, GFSAD, and OPTRAM30, respectively. Additionally, we have further compared CIrrMap250 and IrriMap_CN with OPTRAM30 in two local zones to better illustrate the differences between CIrrMap250 and IrriMap_CN. We present the revised figure alongside the related descriptions below for your reference.

*When examining in the Hexi Corridor (Figure 6), CIrrMap250 exhibits a high agreement with OPTRAM30. While IrriMap_CN captures the general patterns, it tends to underestimate the overall irrigation extent, as demonstrated in zones I and II of the region (Figure 6d). The IAAA product struggles to identify irrigated cropland in this area, displaying significant omission and commission errors. Similarly, GFSAD has a limited ability to accurately depict irrigated areas in the Hexi Corridor.*

[Figure]

*Figure 6. Comparison of large-scale irrigation maps with the field-scale remote sensing irrigation map (OPTRAM30) in the Hexi Corridor of Northwest China. Panels a, b, c, e, and f depict the distribution of irrigated cropland in OPTRAM30, CIrrMap250, IAAA,*

*IrriMap_CN, and GFSAD, respectively. Panel d shows the comparisons of CIrrMap250 and IrriMap_CN with OPTRAM30 in two local zones.*

Figure6 – the [0-100] color scale as provided in Figure4 is missing. In supplementary, Figure S2 (b, c, d) – the magenta color scale (for IrriMap, IAAA, GFSAD) is missing. Additionally, why was year 2019 selected in Figure6a,b (CIrrMap250/IrriMap_CN) for the comparisons with the 2014-2020 OPTRAM30 product?

**Response:** The color scale has been provided in the revised figure 6 (see above). The OPTRAM30 product was derived by counting the detected irrigation events over an extended period (2014-2020) to complement for any missed detections. We opted to compare CIrrMap250 and IrriMap_CN with OPTRAM30 using data from the year 2019 for two reasons. Firstly, the authors of OPTRAM30 have utilized images from 2019 and 2020 to assess the spatial pattern of their irrigation maps. Second, IrriMap_CN covers the period from 2000 to 2019, while our product CIrrMap250 spans from 2000 to 2020. To maintain consistency between IrriMap_CN and CIrrMap250, we selected the year 2019 for data comparison.

Figure S2 becomes Figure S3 now, and it has been replotted in the new manuscript. We presented it here for your reference.

[Figure]

*Figure S3. Comparison of irrigated cropland distribution from CIrrMap250 with existing products (IrriMap_CN, IAAA, GFSAD) for the year 2010. Panel a shows the spatial distribution of irrigated cropland from CIrrMap250, while panels b, c, and d overlay the existing binary maps IrriMap_CN, IAAA, and GFSAD on CIrrMap250, respectively.*

L406-407: "…, namely 2010 and 2020. The estimates of irrigated areas from the other two maps, namely IAAA and GFSAD, are able to explain only a small proportion of the variances in irrigation water withdrawals (i.e., 0.12 and 0.20) …" – Please clarify. According to Figure 7c,e, these (IAAS and GFSAD) metrics only apply to year 2010 NOT 2020.

**Response:** Yes, these metrics of IAAA and GFSAD only apply to the year 2010. We have clarified it in the revised manuscript.

*As shown in Figures 7c and f, the irrigated area estimates from the other two maps (i.e., IAAA and GFSAD) demonstrate limited explanatory power, explaining only 12% and 20% of the variation in irrigation withdrawals for the year 2010.*

L410: "…irrigated area estimates against irrigation water withdrawals…" – maybe you mean 'irrigated water withdrawals against irrigation area estimates…'? Y against X.

**Response:** We agree, and it has been revised.

*Figure 7. Scatterplots of irrigation water withdrawals against irrigated area estimates from different products for the years circa 2010 and 2020. The data are presented in logarithmic units to reflect both small and large values.*

L426: "As shown in Figure 9, all subregions exhibit an increasing trend in irrigated area from 2000 to 2020" - is this conclusion based on CIrrMap250 or some other [reference] data?

**Response:** Yes, the conclusion is based on our new product CIrrMap250. We have revised the sentence.

*As shown in Figure 9, our annual irrigation maps indicated that all subregions exhibited an increasing trend in irrigated area from 2000 to 2020, with NEC expanding significantly faster than the other subregions.*

L435-: Figure 9d - some of the percentage entries in the concentric pie charts are likely incorrect. I.e. percentages for years 2000 and 2020 add up to 101% (11+7+17+30+24+12) and 98% (11+11+16+26+22+12), respectively.

**Response:** Thanks for the reminder. We have checked the results carefully and revised the figure accordingly. We present the revised figure here for your reference.

[Figure]

*Figure 9. Changes in irrigated area across the six subregions of China during 2000-2020. a, Relative changes in irrigated area. b, Changes in China's total irrigated area, with the contribution of different subregions depicted in the inserted pie chart. c, Relative changes in the proportion of irrigated area. d, Proportion of irrigated area for the years 2000, 2010 and 2020.*

L445: "… The net expansion of irrigated area is about 180,000 …" but L427 reads "The irrigated area of China increases from 750,000 to 950,000…", which is ~200,000. Both for the 2000-2020 period. Please be consistent with the presented numbers.

**Response:** Thank you for the comment. We have double-checked our results and have confirmed that the irrigated area of China has increased from about 760,000 to 940,000 km$^2$ over 2000-2020, with a net increase of about 180,000 km$^2$. We have revised this sentence.

*More specifically, China's irrigation aera increased from about 760,000 to 940,000 km2 at an annual rate of 10,000 km2 (or 1.29%/year).*

L465: "…leading to a decrease in irrigation mapping accuracy by 8%-26% (Supplementary Figure S4)." – do these numbers refer to supplementary Figures S3? They do not appear in Figure S4.

**Response:** Apologies for our carelessness. We have placed the figure in the wrong place. This sentence should be referred to Figure S6 in the revised manuscript. We present the figure below for your reference. Furthermore, we have carefully reviewed the figures in Supplementary file to avoid similar mistakes.

[Figure]

*Figure S6. Comparison of the performance of irrigation maps constrained by different irrigated area data. "without adjustment" means the use of the original irrigation statistics, while "with adjustment" indicates the use of the harmonized and reconciled irrigated areas (this study).*

The caption of Figure S3 reads "Comparison of irrigated ratio estimates of CIrrMap250 and IrrMap_CN in China, Northern China, Xinjiang Uygur Autonomous Region" … does this mean that this conclusion only applies to that specific part of China?

**Response:** Figure S3 has been erroneously presented in the original manuscript. We present the correct figure below for your reference. The sentence in the main texts that refers to the figure are as follows (see Section 4.1.2):

*IrriMap_CN estimates irrigation proportion (i.e., the ratio of irrigated cropland area to total cropland area) to be 0.47, 0.37, and 0.61 for China, Northern China, and Xinjiang Uygur Autonomous Region, respectively (Supplementary Figure S4). In*

*comparison, the values derived from CIrrMap250 are 0.58, 0.70, and 0.96, respectively, which align more closely with the official reports (https://gtdc.mnr.gov.cn/).*

[Figure]

*Figure S4. Comparison of irrigated ratio estimates of CIrrMap250 and IrriMap_CN in China, Northern China, Xinjiang Uygur Autonomous Region*

L474-475: "... The accuracy of the irrigated cropland map would decrease by approximately 5%-6% (Supplementary Figure S6) " - Figure S6 (in the supplementary document) contradicts this statement. From the figure, it appears that "considering FC of cropland" (blue bars according to the plot legend, and "this study" according to the caption) yields worse overall accuracies (OA) than "Neglecting FC of cropland" (green bars). This is the case for all three (2000, 2010, 2020) years. Is the plot legend correct?

**Response:** Thanks for the kind reminder. We have mistakenly presented the legend of the figure. As depicted in the revised figure (Figure S8, see below), the accuracy of the irrigated cropland map would decrease by approximately 5%-6% if we disregard the fractional coverage of cropland. Yes, this decrease in mapping accuracy can be observed across the three years (i.e., 2000, 2010, and 2020).

[Figure]

*Figure S8. Comparison of performance of irrigation maps in the scenarios of considering fractional coverage (FC) of cropland (this study) and neglecting FC of cropland*

The detailed point-to-point responses are as follows. Texts in red are the reviewer's comments; **those in black** are our responses to the reviewer's comments; and *those in blue and italics* are the revised texts appeared in the revised manuscript.

General summary:

This study presents the development of a multi-year (2000-2021) irrigated cropland map for China, named CIrrMap250. The authors employ a semi-automatic training approach integrating remote sensing data (vegetation indices, hybrid cropland products, and paddy field maps), county-level irrigation statistics and surveys, and an irrigation suitability map. Utilizing a threshold-calibration method and the random forest algorithm, the CIrrMap250 map is evaluated against reference sites and other large-scale irrigation maps, demonstrating superior accuracy. The study reveals a consistent net expansion of irrigated croplands in Northeast and Northwest China, with over 60% deemed unsustainable due to severe water stress. The CIrrMap250 map holds significant application potential for water resource management and food security. I have some comments below and please address them before this article can be published.

Thanks for the positive comments.

Major comments:

1 L74-81: China's vast agricultural landscape comprises diverse cropping systems and associated irrigation methods, such as rice paddies in the South and Northeast, and corn/wheat rotations in the North China Plain and Northwest. The study does not adequately address this diversity. It would be beneficial for the CIrrMap250 to provide detailed mappings for irrigation methods for associated crop types, if possible. Moreover, the integration of county-level yearbook data on irrigated crop types and rotations could enhance the map's specificity and utility. Clarifying how different cropping systems and crop types are distinguished would significantly improve the comprehensiveness of the methodology. For example, L123 - Mapping 30-m CCropLand30 cropland layer (available every 5-year) - does this dataset also tell you which crop type is associated with each pixel?

**Response:** Thank you for the insightful comments. It would indeed be ideal for irrigation mapping to include the full thematic detail required for agricultural monitoring, such as irrigation methods and crop types. However, there are two major challenges in achieving this.

First, to our knowledge, existing cropland data in China, including CCropLand30, only provide the spatial distribution of cropland without crop type information due to the diversity and complexity of agricultural systems (Zhang et al., 2022a; Van Tricht et al., 2023). While numerous studies have mapped some crops (e.g., wheat, rice and maize) across China, none of them have included all crop types and accounted for

mixed or sequential cropping practices (Dong et al., 2020; You et al., 2021; Shen et al., 2022; Mei et al., 2023; Shen et al., 2023; Zhang et al., 2024a). The mixed and sequential cropping practices are crucial because when a specific crop is mapped in a grid for a given month, the remaining crop types must be allocated to the rest of the available cropland area within that grid. This lack of cropland products that distinguish crop types limits the classification of irrigated versus rainfed crop types.

Second, although statistical yearbooks provide planted area for different crop types, they do not offer information on the irrigated versus rainfed area for each crop types, nor do they detail crop rotations. This also hinders the identification of irrigated versus non-irrigated crop types. Addressing these challenges and mapping the distribution of irrigated and rainfed crops is beyond the scope of this research but will be considered in our future work.

Regarding irrigation methods, there are indeed some statistical data on the areas of different irrigation methods (i.e., flood, drip, and sprinkler irrigation). However, spatially explicit allocation of irrigated areas by different methods is a significant challenge because irrigation methods cannot be easily distinguished by remote sensing data, except for certain systems like center pivot irrigation. This limitation is especially pronounced with coarse-resolution imagery such as MODIS.

2 Although the CIrrMap250 is purportedly an annual dataset, the primary analyses are based on three specific years (2000, 2010, and 2020). Although Figure 9 presents a 20-year timeseries of irrigated croplands in different regions, it is crucial to present the interannual variability of irrigation areas. Analyzing annual data across the entire 20-year period can reveal the influence of climatic factors, such as temperature and precipitation, on irrigation trends. Additionally, showcasing irrigation transitions in various regions, beyond the highlighted area between CSC and NC, would provide a more comprehensive view of national trends.

**Response:** The spatiotemporal changes in irrigated areas were analyzed based on the annual data of CIrrMap250, rather than three specific years (2000, 2010, and 2020), as shown in Figure 8 in the manuscript. Figure 8 shows the interannual trend of irrigated areas from 2000 to 2020 at the pixel scale (see below). Pixels with significant temporal changes (increasing or decreasing trend) in irrigated area (p<0.05) are marked as "expansion" or "reduction," while those with insignificant changes are marked as "stable." The inset panel at the top of the figure depicts the center-of-gravity movement (i.e., spatial trend) of China's irrigated areas at the national scale. Each circle in the inset panel corresponds to the gravity center of China's irrigated area for a specific year (ranging from 2000 to 2020).

We further analyzed the spatial trends in irrigated areas from 2000 to 2020 in each subregion of China. As shown in Supplementary Figure S5 (see below), the gravity center of irrigated areas showed clear trends in NWC, NEC, and NC but was insignificant in the remaining subregions. In NWC, the irrigated area significantly shifted to the northwest, while in NEC, it significantly shifted to the northeast. Meanwhile, there was a northward spatial trend in irrigated areas in NC.

The related results have been added in the revised manuscript

*The gravity center showed clear trends in NWC, NEC, and NC but was insignificant in the remaining subregions (Supplementary Figure S5). In NWC, irrigation significantly shifted to the northwest, while in NEC, it significantly shifted to the northeast. Meanwhile, there was a northward spatial trend in irrigation in NC.*

[Figure]

***Figure 8. Spatiotemporal changes in irrigated area from 2000 to 2020.*** *Pixels exhibiting significant interannual trends (p < 0.05) in irrigated area were labelled as "expansion" or "reduction", while those with insignificant changes are denoted as "stable". Pixels with less than 5% irrigated croplands were excluded from the map. The inset panel on the top of the figure depicts the center-of-gravity movement (spatial trend) of China's irrigated areas at the national scale.*

[Figure]

***Figure S5. Spatial trends in irrigated areas from 2000 to 2020 in the six subregions of China.*** *The top panel shows the interannual trend in irrigated area at the pixel scale (same as Figure 8 in the main text) and illustrates the locations of the gravity centers of irrigated areas for each subregion. Panels a-d depict the center-of-gravity movement of irrigated areas from 2000 to 2020 in each subregion.*

3. Since this data product is a fusion of data from multiple sources, would it be good and necessary to quantify uncertainties of different sources? Such as the assumption outlined in 2.2.1, L151-155, and also in 2.2.2 L164-184. Section 5.2 L486-502 discussed the uncertainties and limitations. What are the limitations associated with integrating multiple sources? For example, for the same region, how do remote-sensing indices, statistics, and survey data differ from each other (or not)? In which regions does each index perform better? Please give some discussion as it may be useful to evaluate the CIrrMap250 product and provide future user information.

**Response:** Thanks for the comment. We agree that it is crucial to convey the underlying uncertainties of data from different sources and the final product. However, it is challenging for us to quantify uncertainties of each data source given the unavailability of ground reference data. Instead, we evaluated our final product CIrrMap250 and discussed possible product uncertainties in relation to data sources. To do so, we have conducted additional analyses and discussions on the uncertainties associated with CIrrMap250 (see below).

[revised manuscript text omitted]

**Figure S10. Uncertainty analysis of the CIrrMap250 product. a**. *Comparison of statistics and surveys of irrigated area across different subregions.* **b**. *Proportion of croplands consistently identified by five state-of-the-art remote sensing land use/cover products, including GlobeLand30 (Chen et al., 2015), GLAD (Potapov et al., 2021), CLUD (Liu et al., 2014), CLCD (Yang and Huang, 2021), and CACD (Yu et al., 2021).* **c**. *Comparison of the accuracy of the hybrid cropland product CCropLand30 across different subregions.* **d**. *Comparison of the accuracy of CIrrMap250 (for the year 2010) across different subregions.*

4. The study employs numerous data products and indices, yet lacks clear definitions and descriptions. A detailed table in the main text or supplementary material, listing all data products used and indices defined, could enhance clarity. Including equations for calculating indices such as PET, aridity index, Water Scarcity Index (WSI), NDVI, EVI, and GI is essential for transparency. Defining WSI and its components, including whether groundwater pumping in North China is considered, would further elucidate the methodology. Providing detailed and comprehensive information, similar to the content in Tables S1-S3, would greatly benefit readers.

**Response:** Thank you for the suggestion. We have compiled a comprehensive list of

the products and variables utilized in this study, which can be found in Supplementary Table S2. Each entry in the table includes a detailed description, formula, and the respective data source. Please refer to the table below for details.

*Table S2. Summary of the products and variables used in this study*

| Product /variable | Description | Formula | Source |
|---|---|---|---|
| CCropLand30 | Hybrid cropland product for China | - | Zhang et al. (2024) |
| CLUD | China's Land-use/cover dataset | - | Liu et al. (2014) |
| NDVI/EVI/GI | Normalized Vegetation Index / Enhanced Vegetation Index / Greenness Index | See Table S1 | MODIS[a] |
| Irrigation suitability | Suitability of cropland for irrigation | Equation 4 in the main text | This study[b] |
| SVI | Irrigation suitability-adjusted peak vegetation index | Equation 5 in the main text | This study[b] |
| Precipitation | Annual precipitation | $\sum_{i=1}^{Ydays} PCP_i$ | NMIC[c] |
| Temperature | Mean annual temperature | $\frac{1}{Ydays}\sum_{i=1}^{Ydays} TMP_i$ | NMIC[c] |
| PET | Annual evapotranspiration | $\sum_{i=1}^{Ydays} PET_i$ | This study[b] |
| Aridity index | Degree of dryness of the climate | $MA\_PCP/MA\_PET$ | This study[b] |
| Irrigation water withdrawal | Total amount of water withdrawals used for crop irrigation | - | PWRD[d] |
| WSI | Water scarcity index | $TWU/WA$ | Zhang et al. (2023) |
| Cropping intensity | Number of crops grown on the same field in a given agricultural year | - | Xu et al. (2017) |
| Soil type | Genetic soil classification system in China | - | RESDC[e] |
| Elevation | Mean elevation | - | SRTM[f] |

| Slope | Mean slope | - | This study[b] |
| Distances to water bodies | Euclidean distance to rivers, lakes, reservoirs, canals, and ponds | | This study[b] |

Note. [a]indicates variables derived from Moderate Resolution Imaging Spectroradiometer (MODIS) data (https://modis.gsfc.nasa.gov/). [b]indicates variables generated in this study. [c]indicates the National Meteorological Information Center (http://data.cma.cn/). [d]indicates the provincial water resources departments. [e]indicates the Resource and Environment Science and Data Center (https://www.resdc.cn/Default.aspx). [f]indicates the Shuttle Radar Topography Mission (https://www.earthdata.nasa.gov/sensors/srtm). Ydays represents the number of days in a given year; $PCP_i$ denotes the amount of precipitation at the $i^{th}$ day; $TMP_i$ indicates mean air temperature at the $i^{th}$ day; $PET_i$ represents evapotranspiration estimated using the Priestley-Taylor method (Priestley and Taylor, 1972); MA_PCP and MA_PET denote mean annual precipitation and PET, respectively; TWU represents total water use, including both groundwater and surface water withdrawals for irrigation, industry, domestic purposes, forestry, livestock, and fishery; WA represents water availability and refers to the total surface water and groundwater generated by precipitation.

5. 3.1 L219 - Can you also give a brief description of the threshold-calibration method, instead of stating "following the previous studies" - letting the readers of this paper understand your method is important. Particularly Equation 5 - how do you determine the threshold? Since you have 20-year data, is this threshold constant? Or does it change year-by-year? Please elaborate.

**Response:** We have provided a brief description of the threshold-calibration method, following your suggestion.

We applied a threshold-calibration method to automatically generate the training pool, following previous studies by Xie et al. (2019; 2021) and Zhang et al. (2022d). With this method, cropland pixels with annual peak vegetation greenness exceeding an optimized threshold were classified as "irrigated". The threshold was individually calibrated for each county and year using available irrigation statistics and surveys. Based on the calculated optimized thresholds, intermediate irrigation maps were generated at the county level. Pixels consistently classified as "irrigated" in all intermediate maps were identified as irrigation candidates, while those classified as "non-irrigated" were considered potential non-irrigated samples.

6. L223-226, please elaborate on this statement - "Cropland with lower elevation, gentler slope, and higher aridity index was hypothesized to have higher irrigation

suitability and potential" - is this statement a hypothesis? Or has it already been demonstrated in Liu et al.? It is not clear by now.

**Response:** Liu et al. (2022) did not directly validate this hypothesis but highlighted the significant role of geographical factors such as elevation, slope, and precipitation in shaping the spatial distribution of irrigated cropland in China using a Select K Best algorithm. Hence, it is a hypothesis that lower elevation, gentler slopes, and higher aridity indices characterize cropland areas with greater irrigation suitability and potential. This hypothesis aligns with previous studies (Worqlul et al., 2015; Worqlul et al., 2017; Li and Chen, 2020; Zhang et al., 2022b), and is proposed for following reasons.

In regions with higher elevations, accessing water resources becomes more challenging, reducing the likelihood of irrigation. For instance, our field observations on the Loess Plateau's high-elevation areas revealed that residents rely solely on deep wells for domestic water, with crop growth entirely dependent on rainfall. Moreover, agricultural productivity at higher altitudes in China is hindered by the absence of irrigation infrastructure and increased costs associated with transportation and labor. Meanwhile, areas with steeper slopes generally have lower water holding capacities and are less suitable for irrigation systems. Typically, slopes exceeding 8% are considered impractical for surface irrigation systems. Therefore, areas with gentler slopes are more conducive to the presence of irrigated cropland. Lastly, croplands with higher aridity indices, characterized by lower precipitation but higher potential evapotranspiration (PET), are also more likely to require irrigation due to greater water demand.

We have clarified the hypothesis in the new manuscript

*A static irrigation suitability map was created based on elevation, slope, and aridity index of cropland. These factors play a crucial role in shaping the spatial distribution of irrigated cropland in China, as demonstrated by Liu et al. (2022). Cropland areas characterized by lower elevation, gentler slopes, and higher aridity indices were hypothesized to exhibit greater irrigation suitability and potential, in line with previous studies (Worqlul et al., 2015; Worqlul et al., 2017; Li and Chen, 2020; Zhang et al., 2022d).*

7. L275-280 the description of Figure 2c and the figure caption don't match - Figure 2c shows 2010, but the text indicates 2020. Please check your text and figure captions. Also, since you can identify the center pivot irrigation system, can you also provide an irrigation method map, distinguishing between sprinkler irrigation (mostly in North China) and flood irrigation (more common in South China)?

**Response:** As shown in Figure 2 (see below), panel c shows the spatial distribution of the third-party samples in 2020. We have double-checked all the texts and figure captions carefully to ensure consistency.

Center pivot irrigation systems are identifiable in remote sensing imagery due to their distinctive circular irrigation pattern centered on pivots, which creates a unique visual signature on crops (Chen et al., 2023). However, other irrigation methods such as flood irrigation, drip irrigation, and sprinkler irrigation are not easily distinguishable using remote sensing data, especially using coarse-resolution datasets like MODIS.

[Figure]

*Figure 2. Spatial distribution of validation samples. **a**, Spatial distribution of the third-party samples in 2000. **b**, Spatial distribution of the samples in 2010 retrieved from provincial land-use maps of China's second National Land Survey. **c**, Spatial distribution of the third-party samples in 2020. **d**, Numbers of irrigated and non-irrigated samples for different years.*

8. L337-340, please define WSI. What are the major water resources used in WSI? How about considering pumping groundwater for irrigation in North China? Is it a part of the WSI calculation and evaluation of the irrigation map?

**Response:** The Water Scarcity Index (WSI) is defined as the ratio of total water use (TWU) to water availability (WA), i.e., WSI = TWU/WA. This index quantifies the fraction of available water resources appropriated by humans. TWU encompasses both groundwater and surface water withdrawals for irrigation, industry, domestic purposes, forestry, livestock, and fishery. WA refers to the total surface water and groundwater generated by precipitation. The definition of WSI is provided in the main text of the new manuscript as well as in Table S2 of the Supplementary file.

*The prefecture-level data on water scarcity index (WSI) for 2010-2020 were extracted from our previous study (Zhang et al., 2023b). WSI is defined as the ratio of total water use to water availability, as shown in Supplementary Table S2. Total water use encompasses both groundwater and surface water withdrawals for irrigation, industry, domestic purposes, forestry, livestock, and fishery. Water availability refers to the total surface water and groundwater generated by precipitation.*

9. Figure 5, please give a scale legend for region A, B, C, and D. i.e., how big are these four regions?

**Response:** Thank you for your suggestion. In the revised figure, scale bars have been added for regions A, B, C, and D, as shown below.

[Figure]

**Figure 5. Visual comparison of CIrrMap250 with existing maps.** *The five rows from top to bottom correspond to the Google map, CIrrMap250, IrriMap_CN, IAAA and GFSAD, respectively. Locations of the four selected zones are presented in Figure 4a.*

10. Figure 8. The transition only shows three years, 2000, 2010, and 2020. What about the interannual variability, since you have 20-year annual data? It would be good to show an interannual timeseries of irrigation transitions across China. Figure 8 highlights an area between CSC and NC; how about other regions in China? What are the transitions across regions over the 20-year period?

**Response:** The spatiotemporal changes in irrigated areas were analyzed based on the annual data of CIrrMap250, rather than three specific years (2000, 2010, and 2020). Figure 8 shows the interannual trend of irrigated areas from 2000 to 2020 at the pixel scale. Pixels with significant temporal changes (increasing or decreasing trend) in

irrigated area (p<0.05) are marked as "expansion" or "reduction," while those with insignificant changes are marked as "stable." The inset panel at the top of Figure 8 depicts the center-of-gravity movement (i.e., spatial trend) of China's irrigated areas at the national scale. Each circle in the inset panel corresponds to the gravity center of China's irrigated area for a specific year (ranging from 2000 to 2020). The spatial trend in irrigated area at the subregional scale has also been provided in Supplementary Figure S5. Please refer to our response to your second comment.

11. Please also discuss the potential use of CIrrMap250, who will be interested in using this data? Science communities, Hydrologic models? Climate models? Or water resource managers?

**Response:** Following your suggestion, we briefly discussed the potential use of CIrrMap250 in the revised manuscript. However, more specific uses are dependent on users and actual applications.

*Despite these limitations, CIrrMap250 makes a valuable contribution to the field of irrigation mapping and is poised to significantly support agricultural, hydrological, and climate studies, as well as water resource management in China. Ongoing efforts to address these limitations and explore potential enhancements will undoubtedly improve the accuracy and utility of our irrigation maps in the future. One of the major applications of CIrrMap250 will be estimating irrigation water use or requirements, considering that irrigated area is a dominate driver of irrigation water withdrawal (Ozdogan and Gutman, 2008; Puy et al., 2021). Secondly, the spatial detail provided by CIrrMap250 can be integrated into crop, hydrological, and climate models to improve the simulations of water uses and land-atmosphere interactions (Uniyal and Dietrich, 2021; Mcdermid et al., 2023; Yang et al., 2023). This integration will advance our understanding of how irrigation practices influence crop yield, and hydrological and climatic processes from local to nationwide scales. Lastly, CIrrMap250 provides insights into irrigation changes and can assist in optimizing the spatial distribution of irrigated croplands (Rosa et al., 2020a; Rosa et al., 2020b), thereby supporting more informed decisions for sustainable water and land use.*

Minor comments:
1: L17-20: "...reconciled them with remote sensing data … integrated with multiple remote sensing data …" These two sentences seem redundant.

**Response:** These sentences describe the data sources utilized in developing CIrrMap250, as well as the primary method employed for data integration. This information is crucial and has been retained.

*We harmonized irrigation statistics and surveys and reconciled them with remote sensing data. The refined estimates of irrigated area were then integrated with multiple remote sensing data (i.e., vegetation indices, hybrid cropland product, and paddy field maps) and an irrigation suitability map through a semi-automatic training approach.*

2: L94: "both cropland and other land use"

**Response:** Revised.

*This leads to the widespread presence of mixed pixels where cropland and other land use/cover types coexist.*

3: L231: "with the assumption that"

**Response:** Revised.

*The peak vegetation index was subsequently adjusted by irrigation suitability (Eq. 5), with the assumption that irrigated cropland, being greener and more productive, is also more suitable for irrigation compared to rainfed cropland.*

4: 3.3 L265-270 and 3.3.3 L310-323, these texts seem similar and redundant.

**Response:** Thanks for the suggestion. The content in "3.3 L265-270" has been removed.

---

## Author Response (AR2)

**Responses to the comments of Referee #1**

**Article ID:** essd-2024-2
**Title:** CIrrMap250: Annual maps of China's irrigated cropland from 2000 to 2020 developed through multisource data integration
**Authors:** Ling Zhang, Yanhua Xie, Xiufang Zhu, Qimin Ma, Luca Brocca

Dear Editor,

Thank you very much for the great efforts on our manuscript. Below, you will find the original comments from the reviewer along with our point-to-point responses.

Texts in red are the reviewer's comments; **those in black** are our responses to the reviewer's comments; and *those in blue and italics* are the revised texts appeared in the revised manuscript.

The authors have greatly improved the manuscript. I am generally satisfied with their responses + the corrections made in the revised manuscript/supplement. As such, I only have a few minor comments.
Thank you for the positive comment.

- While they have somewhat explained why they use the total FVC and not the binary [1 0] in their irrigation maps, I still believe that could lead to misinterpretation by users, e.g. a pixel could have 50% vegetation coverage but only 50% of that is irrigated (thus only 25% of the pixel is under irrigation).

**Response:** Thank you for the valuable comment. As mentioned in our earlier response, binary irrigation maps can lead to misinterpretations, where users may assume pixels with a value of 1 are fully covered by irrigated areas. This can result in inaccurate estimates of irrigated area, especially in regions with small and fragmented farms. In future work, we plan to use higher resolution satellite data (e.g., Landsat and Sentinel images) to better address this issue more effectively.

To further clarify and avoid potential misinterpretations, we have provided a clear detailed explanation of our product in the data availability section (Page 25 Lines 532-536).

*Our maps show the percentage of each 250 m pixel covered by irrigated cropland (i.e., pixel value = irrigated area / pixel area ×100). Note that our product accounts for*

*the fractional coverage of croplands within coarse-resolution MODIS pixels but does not differentiate between irrigated and rainfed croplands at subpixel scales. For example, if a pixel has 50% cropland coverage, all cropland within that pixel would be classified as either "irrigated" or "non-irrigated".*

- "(CIrrMap250) feature a spatial resolution of 250 meters and an annual temporal resolution": should that be interpreted to mean that you assume no short or mixed periods with/without irrigation during the year? Maybe add a comment why you did not consider a product based on growth seasons?

**Response:** Yes, our product was created at an annual frequency from 2000 to 2020, and thus does not provide irrigation information at monthly or seasonal scales. As you noted, there may be short or mixed periods of irrigation throughout the year. In this study, we define cropland in a pixel as "irrigated" if it has been irrigated at any point within the year, regardless of the specific month or season. This assumption aligns with several previous studies (Zhu et al., 2014; Meier et al., 2018; Xie and Lark, 2021; Zhang et al., 2022; Wu et al., 2023).

Using satellite observations to identify seasonal irrigation practices is possible. However, there are several challenges, particularly for nationwide mapping. First, developing such a product requires a high-resolution cropland dataset that can accurately map cropland areas, distinguish among various crop types (e.g., rice, wheat, maize), and account for mixed and sequential cropping practices. To our knowledge, no existing cropland data in China meets these criteria. Second, creating a growth-season-based product requires high-quality training samples with high temporal resolution and detailed information on crop types. Meeting these requirements poses significant challenges.

In the revised manuscript, we have pointed out the above-mentioned limitation of our product in the discussion section (Page 25 Lines 512-516).

*Additionally, CIrrMap250 was created at an annual frequency and therefore does not provide monthly and seasonal irrigation information. A pixel is classified as "irrigated" if it has been irrigated at any point during the year, regardless of the specific month or season. While a growth-season-based irrigation product could be more desirable, it faces significant challenges, such as the lack of high-resolution crop type information (e.g., rice, wheat, maize) and high-quality training samples with sufficient temporal resolution.*

- As it is a separate document, ensure the corrected supplementary document is the one available online (at the moment, only the old uncorrected version is accessible online).

**Response:** Thank you for the reminder. We have double-checked our submitted files in the last round of revisions and confirmed that the revised supplementary document had been correctly uploaded. However, only the previous, uncorrected supplementary document is accessible online (we are unsure why). We anticipate that the revised manuscript and its supplementary materials will be updated after formal publication.

**References**

Meier, J., Zabel, F., Mauser, W., 2018. A global approach to estimate irrigated areas – a comparison between different data and statistics. Hydrology and Earth System Sciences, 22(2): 1119-1133.

Wu, B., Tian, F., Nabil, M., Bofana, J., Lu, Y., Elnashar, A., Beyene, A.N., Zhang, M., Zeng, H., Zhu, W., 2023. Mapping global maximum irrigation extent at 30m resolution using the irrigation performances under drought stress. Global Environmental Change, 79: 102652.

Xie, Y., Lark, T.J., 2021. Mapping annual irrigation from Landsat imagery and environmental variables across the conterminous United States. Remote Sensing of Environment, 260: 112445.

Zhang, C., Dong, J., Ge, Q., 2022. IrriMap_CN: Annual irrigation maps across China in 2000–2019 based on satellite observations, environmental variables, and machine learning. Remote Sensing of Environment, 280: 113184.

Zhu, X., Zhu, W., Zhang, J., Pan, Y., 2014. Mapping Irrigated Areas in China From Remote Sensing and Statistical Data. IEEE Journal of Selected Topics in Applied Earth Observations and Remote Sensing, 7(11): 4490-4504.